# Contact Wasserstein Geodesics for Non-Conservative Schrödinger Bridges

**Andrea Testa**[1,2]**, Søren Hauberg**[3]**, Tamim Asfour**[2]**, Leonel Rozo**[1,4]
[1] Bosch Center for Artificial Intelligence (BCAI), [2] Karlsruhe Institute of Technology (KIT),
[3] Technical University of Denmark (DTU), [4] Italian Institute of Artificial Intelligence (AI4I)
Emails: andrea.testa@de.bosch.com, sohau@dtu.dk, asfour@kit.edu, leonel.rozo@ai4i.it

## Abstract

The Schrödinger Bridge provides a principled framework for modeling stochastic processes between distributions; however, existing methods are limited by energy-conservation assumptions, which constrains the bridge's shape preventing it from model varying-energy phenomena. To overcome this, we introduce the *non-conservative generalized Schrödinger bridge* (NCGSB), a novel, energy-varying reformulation based on contact Hamiltonian mechanics. By allowing energy to change over time, the NCGSB provides a broader class of real-world stochastic processes, capturing richer and more faithful intermediate dynamics. By parameterizing the Wasserstein manifold, we lift the bridge problem to a tractable geodesic computation in a finite-dimensional space. Unlike computationally expensive iterative solutions, our *contact Wasserstein geodesic* (CWG) is naturally implemented via a ResNet architecture and relies on a non-iterative solver with near-linear complexity. Furthermore, CWG supports guided generation by modulating a task-specific distance metric. We validate our framework on tasks including manifold navigation, molecular dynamics predictions, and image generation, demonstrating its practical benefits and versatility.
Project website: https://sites.google.com/view/c-w-g

## 1 Introduction

Inferring the stochastic process that most likely generates a set of sparse observations is a fundamental challenge, e.g., in cellular dynamics (Yeo et al., 2021; Zhang et al., 2024; Moon et al., 2019), meteorology (Franzke et al., 2015), and economics (Kazakevičius et al., 2021; Huang et al., 2024). Here, the target is not merely the distributions of observed data, but rather the underlying dynamics of cell populations, weather patterns, or economic phenomena, enabling reconstruction of missing intermediate states and predicting the systems' future evolution.

The *Schrödinger Bridge* (SB, Schrödinger (1931)) is a powerful mathematical framework to address this. SB seeks the most likely stochastic path between marginals (i.e., observations), while being close to a reference process, typically Brownian motion. This offers a general stochastic optimal control perspective that encompasses

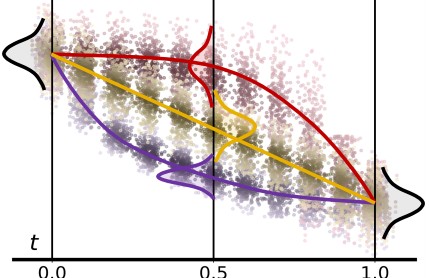

Figure 1: Probability paths under energy-conserving (——), energy-decreasing (——), and energy-increasing conditions (——) (details in App. F.2). Our NCGSB enables energy variation, increasing modeling flexibility in applications where distributions at intermediate time steps are of interest.

both *Optimal Transport* (OT, Vargas et al. (2021)) and generative approaches such as diffusion models (Ho et al., 2020; Chen et al., 2024), which can be interpreted as optimal bridges with a Gaussian initial marginal (Bortoli et al., 2021). Unfortunately, current SB solvers operate on an infinite probability space and rely on iterative forward–backward stochastic simulations or progressive refinement of the reference dynamics. This leads to complex and costly optimizations, limiting adoption.

Solutions provided by the SB preserve the distribution's energy throughout the full stochastic path. Here, energy is understood as a combination of *kinetic energy*, which reflects how fast samples move across the probability manifold, and *potential energy* from the underlying landscape. This energy-preservation principle constrains the shape of the bridge and excludes stochastic paths with

varying energy profiles, such as dissipative behaviors commonly encountered in real-world physical systems, e.g., storms gradually losing intensity in weather forecasting.

**This paper** overcomes both the energy-preserving limitation and the iterative optimization schemes of existing SB approaches by formulating a novel geometric generalization of the SB to model non-conservative systems and by proposing a near-linear time solver. Specifically, we build on the geometric perspective of the SB, which casts it as a flow governed by Hamiltonian dynamics on the Wasserstein probability space (Sec. 3). Extending the Hamiltonian system to a contact Hamiltonian (Zadra, 2023), we propose a more general energy-varying formulation of the SB problem: **The Non-Conservative Generalized SB** (NCGSB, Sec. 4). To make computations tractable, we introduce the **Contact Wasserstein Geodesic** (CWG), which solves the NCGSB problem by casting it as a geodesic computation, reshaping its cost functional into a Riemannian metric whose induced distance is minimized. Discretizing the geodesic leads to geodesic segments that match standard residual blocks. We show how this leads to an efficient solver that avoids outer iteration loops and achieves near-linear complexity in both dimensionality and batch size. Additionally, our approach allows for guided generation, steering the learned process via a task-dependent loss, which in our geometric framework corresponds to modulating the Riemannian metric (Sec. 5). We demonstrate our approach on benchmarks and tasks such as LiDAR manifold navigation, molecular dynamics predictions, unpaired images matching, image-based reconstruction of systems such as sea-surface temperature and robotic pick-and-place tasks (Sec. 6).

In summary, **we contribute**: (1) A novel non-conservative formulation of the Schrödinger Bridge problem that models a wider range of real-world physical stochastic processes by building on geometric flows governed by contact Hamiltonian dynamics; (2) The introduction of the Contact Wasserstein Geodesic (CWG) framework, a general geometric solver compatible with all Schrödinger Bridge variants, enabling fast and scalable computation; (3) a guided generation method for steering the bridge learned by CWG according to new task specifications.

## 2 RELATED WORK

**Schrödinger bridges.** The SB problem imposes no constraints on the probabilistic path beyond matching the endpoint marginals, limiting its applicability when intermediate observations exist. It also does not permit including known physical laws governing the system's dynamics. The *multi-marginal Schrödinger Bridge* (mmSB) treats intermediate observations (Theodoropoulos et al., 2025) as constraints, which enables reconstruction of continuous dynamics without piecewise approximations. In contrast, the *Generalized Schrödinger Bridge* (GSB) adds a state cost, allowing potential energy functionals to be minimized along the probability path. This allows to model mean-field interactions (Gaitonde et al., 2021; Ruthotto et al., 2020), conservative forces (Philippidis et al., 1979; Noé et al., 2020), or geometric priors (Chen & Lipman, 2024; Liu et al., 2018).

**Non-conservative Schrödinger bridge formulations.** The SB problem assumes constant energy, preventing it from modeling non-conservative systems. The *momentum SB* augments the state space with a velocity term (Theodoropoulos et al., 2025; Chen et al., 2023), allowing damping to be incorporated (Blessing et al., 2025; Sterling et al., 2025). However, this doubles the state space and increases computational cost. Other extensions replace the Brownian reference process with the *Ornstein–Uhlenbeck (OU)* process (Orland, 2025; Zhang & Stumpf, 2025), introducing a non-conservative prior for the target dynamics. The OU process defines a mean distribution with a curl-free component that drives convergence and a divergence-free component that induces rotation. Yet, this formulation lacks a mechanism for energy dissipation in the rotational dynamics. **We introduce** a more general energy-varying framework in which dissipation naturally emerges across all components of the dynamics, while only requiring a scalar augmentation of the state space.

**Schrödinger bridge solvers.** Matching-based iterative approaches (Bortoli et al., 2024; Shi et al., 2023; Gushchin et al., 2024) have gained popularity by improving the scalability and robustness of traditional *Iterative Proportional Fitting (IPF)* algorithms (Kullback, 1968; Léonard, 2013) through Markovian projections, thereby avoiding the need for full trajectory storage. However, for GSB, restrictive assumptions like Gaussian probability paths limit expressivity (Liu et al., 2024; Tang et al., 2025), while for mmSB, global trajectory consistency is compromised due to the piecewise nature of the approach and its sensitivity to the initial choice of the reference process (Shen et al., 2025). To overcome these limitations, the stochastic dynamics can be learned indirectly by leveraging the ana-

lytical optimality conditions of the SB problem (Vargas et al., 2021; Chen et al., 2022). This has been shown to scale more effectively to both the GSB (Liu et al., 2022; Buzun et al., 2025) and the mmSB problem (Theodoropoulos et al., 2025; Hong et al., 2025; Chen et al., 2023), due to the symmetries inherent in these optimality conditions (App. A). However, these methods are limited by a classical SB solver bottleneck: the computational overhead of their iterative nature, which alternates between forward–backward passes or repeated dynamics integration and reference updates. **We propose** a cheaper non-iterative solver that scales nearly linearly with both dimensionality and sample size.

**Schrödinger bridge guided generation.** The SB framework naturally extends to conditional generation, where the marginals and the transition path depend on additional parameters or objectives (Shi et al., 2022). Model guidance techniques (Song et al., 2023; Guo et al., 2024) introduce a guidance term into the stochastic process, typically derived from the gradient of a loss function. This approach steers the flow locally and resembles a gradient-based form of optimal control. Alternatively, Raja et al. (2025) employs a global optimal control perspective. Their approach generates full trajectories and chooses the one minimizing a task-specific action functional. However, their method is deterministic and yields a single optimal path rather than a posterior distribution over paths. Unlike previous guidance methods, **we propose** a hybrid approach to guided generation within the NCGSB framework. By embedding a task-specific loss into the potential energy of the NCGSB, we reshape its Riemannian metric so that the resulting geodesic reflects the guidance objective, allowing the learned dynamics to align with the desired outcome.

## 3 PRELIMINARIES

We introduce the necessary notation throughout the following sections, and summarize it in App. B.

**The Wasserstein manifold.** Before formally introducing the SB problem, we define its domain. Let $\mathcal{P}^+(\mathcal{M})$ denote the space of smooth, positive density functions supported on a manifold $\mathcal{M}$. Each element $\rho \in \mathcal{P}^+(\mathcal{M})$ is a function $\rho(x) : \mathcal{M} \to \mathbb{R}^+$ satisfying $\int_{\mathcal{M}} \rho(x)\, dx = 1$. The density dynamics is represented by a time-dependent family $\{\rho^t\}_{t \in \mathbb{R}^+} \subset \mathcal{P}^+(\mathcal{M})$. The infinitesimal variation of the density at time $t$ is the time derivative $\partial_t \rho^t(x)$, which lies in the tangent space $\mathcal{T}_{\rho^t} \mathcal{P}^+(\mathcal{M})$. The collection of all such tangent spaces forms the tangent bundle $\mathcal{T}\mathcal{P}^+(\mathcal{M})$. When equipped with the Wasserstein metric, $\mathcal{P}^+(\mathcal{M})$ becomes a Riemannian manifold (Ambrosio et al., 2005). The corresponding metric tensor is defined as,

$$g^{\mathcal{W}_2}(\partial_t \rho^t, \partial_t \rho^t) = \int_{\mathcal{M}} \partial_t \rho^t(x)(-\Delta_{\rho^t})^\dagger \partial_t \rho^t(x)\, \rho^t(x)\, dx, \tag{1}$$

where $(-\Delta_{\rho^t})^\dagger$ is the inverse of the weighted Laplacian operator $\Delta_{\rho^t} = -\nabla_x \cdot (\rho^t \nabla_x)$ (Chow et al., 2020), inducing an inner product on $\mathcal{T}\mathcal{P}^+(\mathcal{M})$ and a distance $d_{\mathcal{W}_2}(\rho_a, \rho_b)$ for $\rho_a, \rho_b \in \mathcal{P}^+(\mathcal{M})$. The minimum-length curve $\rho^t$ connecting the two distributions $\rho_a, \rho_b$ is called a *geodesic*.

**Multi-marginal generalized Schrödinger bridge (mmGSB).** Given two endpoint densities $\rho_a, \rho_b \in \mathcal{P}^+(\mathcal{M})$, the SB problem (Schrödinger, 1931; Schrödinger, 1932) seeks the most probable interpolating density path $\rho^t$. This minimizes the Kullback–Leibler divergence w.r.t. a reference process $\rho_{\text{ref}}^t$, typically Brownian motion. The SB problem is equivalent to a stochastic optimal control setting (Dai Pra, 1991), which minimizes the cost required to transport a set of diffusing particles from an initial distribution $\rho_a$ to a target distribution $\rho_b$. This dynamic reformulation of the OT problem (Benamou & Brenier, 2000; Chen et al., 2014) has solutions corresponding to geodesics on the Wasserstein manifold $\mathcal{P}^+(\mathcal{M})$. These trajectories are straight, since the classical SB problem assumes that particles dynamics are unaffected by external potential functions. This assumption, however, limits our ability to model complex real-world physical systems.

Intermediate observations represented by marginal distributions $\{\rho_m\}_{m=1}^M$ at specific time steps $\{t_m\}_{m=1}^M$, can also be incorporated as additional constraints (Chen et al., 2023; Tang et al., 2025). This leads to the mmGSB problem,

$$\min_{v^t} J(v^t, \rho^t) = \int_0^1 \int_{\mathcal{M}} \left( \frac{1}{2} \|v^t(x)\|^2 + U(x) \right) \rho^t(x)\, dx\, dt; \tag{2a}$$

$$\text{s.t. } \partial_t \rho^t(x) + \nabla_x \cdot \left( \rho^t(x)\, v^t(x) \right) = \varepsilon \Delta_x \rho^t(x); \tag{2b}$$

$$\rho^0 = \rho_a,\ \rho^1 = \rho_b,\ \rho^{t_m} = \rho_m,\ \forall m \in 1, \dots, M. \tag{2c}$$

Here, the density evolution $\rho^t$ is governed by the Fokker–Planck equation (2b), which generalizes Brownian motion by incorporating a deterministic drift term $v^t$ alongside a stochastic diffusion term scaled by $\varepsilon$. This drift $v^t$ acts as the control variable and ensures that the probability path satisfies the boundary conditions in equation (2c). The objective functional includes the kinetic energy associated with the drift to quantifies the deviation from the (uncontrolled) reference stochastic process and the potential energy $U$ to favor low-potential pathways.

**Wasserstein Hamiltonian flows and geodesics.** A convenient solution to the mmGSB problem (2) is to specify analytical optimality conditions (Sec. 2). These take the form of a Wasserstein Hamiltonian Flow (Chow et al., 2020), which describes a probability distribution evolving according to Hamiltonian dynamics. This evolution lies on planes tangent to $\mathcal{P}^+(\mathcal{M})$, specifically on the cotangent bundle $\mathcal{T}^*\mathcal{P}^+(\mathcal{M})$, the dual of $\mathcal{T}\mathcal{P}^+(\mathcal{M})$, and it is governed by the derivatives of a scalar Hamiltonian function $H$. However, their integration remains computationally expensive (Buzun et al., 2025; Wu et al., 2025), so we propose a geometric reformulation that results in a significant simplification. To this end, we introduce Proposition 1, a standard result from differential geometry (App. C.2), which is instrumental in lifting these equations to geodesics on $\mathcal{P}^+(\mathcal{M})$.

> **Proposition 1.** *Let the optimality conditions of the mmGSB problem (2) be expressed in Hamiltonian form, yielding the optimal bridge $\rho^t(x)$. Then, $\rho^t(x)$ can be viewed as a geodesic connecting the marginals in equation 2c w.r.t. the modified Riemannian metric $g_J$, known as the Jacobi metric (Abraham & Marsden, 2008).*

To access the Jacobi metric and determine the corresponding geodesic, we first derive the Hamiltonian optimality conditions of the mmGSB problem (2) using Lagrange multipliers (Cui et al., 2024). This introduces a potential function $S^t(x)$, whose gradient defines the drift via $v^t(x) = \nabla_x S^t(x)$. The potential enforces the dynamic constraint (2b) within the cost functional (2a), whose first variation yields the Hamiltonian optimality conditions,

$$\partial_t \rho^t(x) = \partial_S H(\cdot) = -\nabla_x \cdot \left( \rho^t(x) \nabla_x S^t(x) \right); \tag{3a}$$

$$\partial_t S^t(x) = -\partial_\rho H(\cdot) = -\tfrac{1}{2}\|\nabla_x S^t(x)\|^2 + \tfrac{1}{2}\varepsilon^2 \partial_\rho I(\rho^t(x)) + U(x), \tag{3b}$$

with the corresponding Hamiltonian, $H(\rho^t, S^t) = \mathcal{K}(\rho^t, S^t) + \mathcal{F}(\rho^t)$, defined as the sum of a kinetic energy $\mathcal{K}$ and a potential energy $\mathcal{F} = -U - I$, dependent on the potential function $U$ and the Fisher information $I$. A detailed derivation of these dynamics and the full Hamiltonian function are given in App. D.1. To handle boundary conditions (2c), the Hamiltonian dynamics (3) are typically integrated backward in time, where the solution at each intermediate point $(\rho^{t_m}, S^{t_m})$ serves as the initial condition for the next segment (Theodoropoulos et al., 2025). Equation (3a) links the potential function $S^t(x)$ to the infinitesimal density variation $\partial_t \rho^t \in \mathcal{T}\mathcal{P}^+(\mathcal{M})$ via the weighted Laplacian operator $\Delta_{\rho^t}$, introduced through the Wasserstein metric (1). This connection establishes a correspondence between the tangent bundle $\mathcal{T}\mathcal{P}^+(\mathcal{M})$ and the cotangent bundle $\mathcal{T}^*\mathcal{P}^+(\mathcal{M})$, where $S^t(x)$ naturally resides, and where the Hamiltonian dynamics of $(\rho^t, S^t)$ unfold.

By Proposition 1, the Hamiltonian dynamics (3) corresponds to a geodesic flow on the underlying Wasserstein manifold $\mathcal{P}^+(\mathcal{M})$, which minimizes the Jacobi metric $g_J = (H - \mathcal{F}) g^{\mathcal{W}_2}$. The original metric $g^{\mathcal{W}_2}$ (1) accounts only for the kinetic energy of the transport map $\mathcal{K}$ by measuring distances between distributions. In contrast, the Jacobi metric $g_J$ also includes the potential energy $\mathcal{F}$, which is maximized to attain values $\mathcal{F} \approx H$. Consequently, computing the geodesic between marginals under this metric is equivalent to solving the mmGSB problem (2).

## 4 THE NON-CONSERVATIVE GENERALIZED SCHRÖDINGER BRIDGE

**Non-conservative formulation.** The solution to the GSB problem (2) assumes a constant energy function $H$, and restricts the drift $v^t$ to depend solely on the potential energy $\mathcal{F}$. This limits the model's flexibility in representing dynamics that cannot be described by a conservative potential, which reduces its ability to capture real-world processes involving energy dissipation and external interactions. To overcome this, we introduce the *non-conservative generalized Schrödinger bridge* (NCGSB), which allows for time-varying energy systems. To do so, we reformulate the cost functional $J$ as the time integral of a new scalar state $z^t$, representing the *Lagrangian action*, whose evolution depends recursively on itself. The NCGSB problem is formulated as follows,

$$\min_{v^t} J(v^t, \rho^t) = \int_0^1 \partial_t z^t \, dt; \tag{4a}$$

$$\text{s.t. } \partial_t z^t = \int_{\mathcal{M}} \left( \frac{1}{2} \|v^t(x)\|^2 + U(x) \right) \rho^t(x)\, dx - \gamma z^t; \tag{4b}$$

$$\partial_t \rho^t(x) + \nabla_x \cdot \left( \rho^t(x)\, v^t(x) \right) = \varepsilon \Delta_x \rho^t(x); \tag{4c}$$

$$\rho^0 = \rho_a,\ \rho^1 = \rho_b,\ \rho^{t_m} = \rho_m,\ \forall m \in 1, \ldots, M, \tag{4d}$$

where $\gamma \in \mathbb{R}$ is a damping factor. The objective in equation 4a is no longer to minimize a static quantity, but rather a time-varying state $z^t$. Its dynamics (4b) depend explicitly on its current value. This recursive structure endows the system with a form of memory, as its evolution is influenced by the entire trajectory, implicitly encoded in $z^t$. Because non-conservative forces are path-dependent, augmenting the system's state space with the scalar $z^t$ allows their effects to be modeled, enabling the system's energy to vary over time. The sign and magnitude of $\gamma$ determine the direction and rate of this variation. By relaxing the implicit energy-conservation constraint of the GSB problem, our approach enhances the model's flexibility and improves the quality of the resulting optimal solution.

**Guided NCGSB.** NCGSB (4) can be extended to the guided generation setting by introducing a guiding function $f$, which steers the generative process toward desired conditions at any chosen time (Song et al., 2023; Guo et al., 2024). For a given time $t_s$, the guidance is expressed as $y = f(x^{t_s})$, with $x^{t_s} \sim \rho^{t_s}$. To enforce this form, the bridge $\rho^t$ is steered according to $\rho^t(x|y) = \frac{1}{Z} \rho^t(x)\, e^{-\|y - f(x^{t_s})\|^2}$, where $Z$ is a normalization constant and $x^{t_s}$ denotes a sample from the predicted guided distribution $\rho^{t_s}(x|y)$. By Bayes' rule, the dynamics of the guided bridge $\rho^t(x|y)$ acquire an additional guidance term via the drift $v^t$, determined by $\|y - f(x^{t_s})\|^2$. To perform a guided generation that enforces the constraint $y = f(x^{t_s})$ while preserving the underlying data manifold, we incorporate $y$ into the Lagrangian action constraint (4b) as (see App. D.3 for details),

$$\partial_t z^t = \int_{\mathcal{M}} \left( \frac{1}{2}\|v^t(x)\|^2 + U(x) + \|y - f(x^{t_s})\|^2 \right) \rho^t(x)\, dx\ -\ \gamma z^t. \tag{5}$$

**Wasserstein contact Hamiltonian flows and geodesics.** Analogously to mmGSB (2), understanding the dynamics of the optimality conditions in NCGSB (4) is essential for reformulating it as a geodesic computation. As detailed in App. D.2, we propose to leverage the contact Hamiltonian formalism (Kholodenko, 2013), an extension of classical Hamiltonian mechanics to non-conservative systems (App. C.1), to model the dynamics of the NCGSB optimality conditions as Wasserstein contact Hamiltonian flows. This generalizes Prop. 1, since the contact Hamiltonian dynamics defines a geodesic but on the extended space $\mathcal{P}^+(\mathcal{M}) \times \mathbb{R}$ (Udrişte, 2000; Testa et al., 2025). The contact Hamiltonian optimality conditions are,

$$\partial_t \rho^t(x) = \partial_S H(\cdot) = -\nabla_x \cdot (\rho^t(x)\nabla_x S^t(x)), \tag{6a}$$

$$\partial_t S^t(x) = -\partial_\rho H(\cdot) - S^t(x)\partial_z H(\cdot) = -\frac{1}{2}\|\nabla_x S^t(x)\|^2 + \frac{1}{2}\varepsilon^2 \partial_\rho I(\rho^t(x)) + U(x)$$
$$- \gamma S^t(x) - \varepsilon\gamma \log \rho^t(x), \tag{6b}$$

$$\partial_t z^t = S^t(x)\partial_S H(\cdot) - H(\cdot) = \int_{\mathcal{M}} \left( \frac{1}{2}\|\nabla_x S^t(x)\|^2 + U(x) \right) \rho^t(x)\, dx + \frac{1}{2}\varepsilon^2 I(\rho^t)$$
$$- \int_{\mathcal{M}} \varepsilon\gamma(\log \rho^t(x) - 1)\rho^t(x)\, dx - \gamma z^t, \tag{6c}$$

The corresponding contact Hamiltonian function is defined as, $H(\rho^t, S^t, z^t) = \mathcal{K}(\rho^t, S^t) + \mathcal{F}(\rho^t) + \mathcal{B}(\rho^t) + \gamma z^t$. This differs from its conservative counterpart in two ways. First, its explicit dependence on $z^t$ allows the total energy to vary over time. Second, the potential energy is augmented by an entropy term, $\mathcal{B}(\rho^t) = \int_{\mathcal{M}} \varepsilon(\log \rho^t(x) - 1)\rho^t(x)\, dx$, producing an additional diffusion in the dynamics. As previously mentioned, for guided generation, an additional potential energy term $\|y - f(x^{t_s})\|^2$ can be introduced here to steer the flow. Geometrically, the dynamics of $(\rho^t, S^t, z^t)$ can be interpreted as a flow on the cotangent bundle of the Wasserstein manifold, augmented by the scalar state $z^t$. That is, the dynamics unfold on the space $\mathcal{T}^*\mathcal{P}^+(\mathcal{M}) \times \mathbb{R}$.

The contact Hamiltonian flow evolving on the extended phase space $\mathcal{T}^*\mathcal{P}^+(\mathcal{M}) \times \mathbb{R}$, and interpolating between the marginal densities, induces a geodesic on the augmented manifold $\mathcal{P}^+(\mathcal{M}) \times \mathbb{R}$. This geodesic minimizes a Jacobi metric $\tilde{g}_J = (H - \mathcal{F} - \mathcal{B})\, g^{\mathcal{W}_2}$, which generalizes the classical Wasserstein metric by incorporating the potential energy of the contact Hamiltonian function. Com-

puting the geodesic under the Jacobi metric $\tilde{g}_J$ corresponds to the NCGSB problem (4). Unlike the conservative case, the contact Hamiltonian $H$ is no longer constant along the flow, allowing the total energy to vary over time. This introduces an additional degree of freedom that can be leveraged to shape the system's energy along the path over $\mathcal{P}^+(\mathcal{M})$. This is the reason why the geodesic $(\rho^t, H^t)$ is defined on the extended space $\mathcal{P}^+(\mathcal{M}) \times \mathbb{R}$.

## 5 CONTACT WASSERSTEIN GEODESICS (CWG)

**ResNet resembles a discrete geodesic.** Our objective is to compute a geodesic $\rho^t$ on $\mathcal{P}^+(\mathcal{M})$, induced by the contact Hamiltonian dynamics, that is constrained to pass through a set of observed marginals $\{\rho_a, \rho_m, \rho_b\}$ (i.e., discretized distributions along the probability path). These constraints naturally lead to a discretized parameterization of $\rho^t$, where the overall density transformation is modeled as a composition of maps, each connecting a pair of consecutive observations. A ResNet is ideally suited for this problem, as its sequential block structure directly mirrors this piecewise, compositional nature of the approximated geodesic. Let $\lambda$ be a fixed reference measure on $\mathcal{P}^+(\mathcal{M})$ (e.g., a standard Gaussian or uniform distribution). We define a $(K+1)$-block ResNet as follows,

$$T_{\{\theta^k\}_{k=0}^K} = T_{\theta^K} \circ \cdots \circ T_{\theta^1} \circ T_{\theta^0}, \tag{7}$$

with parameters $\{\theta^k\}_{k=0}^K \in \Theta$. The process begins by sampling an initial batch of points $x^s \sim \lambda$, that is pushed forward through the first block to obtain $x^{t_0} = T_{\theta^0}(x^s)$. Then, the parameters $\theta^0$ are optimized such that the resulting pushforward reference measure approximates the initial marginal $\rho_\theta^{t_0} \approx \rho_a$. Thereafter, each subsequent block $k$ pushes forward the sample via $x^{t_{k+1}} = T_{\theta^{k+1}}(x^{t_k})$, $x^{t_k} \sim \rho_\theta^{t_k}$. The full pushforward map induces,

$$\rho_\theta^{t_{k+1}} = (T_{\theta^{k+1}})_\# \rho_\theta^{t_k} = \rho_\theta^{t_k}\big(T_{\theta^{k+1}}^{-1}(x^{k+1})\big) \det\big[\nabla_x T_{\theta^{k+1}}^{-1}(x^{k+1})\big]. \tag{8}$$

Starting from the reference measure $\lambda$, the ResNet parameters $\{\theta^k\}_{k=0}^K$ define a sequence of discrete probability transitions $\{\partial_t \rho_\theta^{t_k}\}_{k=0}^K$, which in turn specify the discrete family of densities $\{\rho_\theta^{t_k}\}_{k=0}^K$. Geometrically, the discretizations $\{\partial_t \rho_\theta^{t_k}, \rho_\theta^{t_k}\}_{k=0}^K$, provided by the ResNet, approximate $(\rho^t, \partial_t \rho^t) \in \mathcal{T}\mathcal{P}^+(\mathcal{M})$, which can be seen as inducing a mapping from $\mathcal{T}\mathcal{P}^+(\mathcal{M})$ onto the parameter space $\Theta$ (Fig. 5). As stated in Proposition 2, the existence of such a map endows the finite-dimensional space $\Sigma$, where the parameterized densities $\rho_\theta^{t_k}$ reside, with the geometric properties of $\mathcal{P}^+(\mathcal{M})$. This lifting enables faster and tractable computations for SB problems.

> **Proposition 2.** *Approximate the evolution of the density $\rho^t \in \mathcal{P}^+(\mathcal{M})$ by a series of $K$ smooth parametrized pushforwards $T_{\theta^k}$, with $\theta^k$ belonging to a finite-dimensional space $\Theta$. If each pushforward $T_{\theta^k}$ is an immersion $T_{\theta^k} : \Theta \to \mathcal{T}\mathcal{P}^+(\mathcal{M})$, then the parameter space $\Theta$ can be endowed with a Riemannian structure via the pullback of the Wasserstein metric $g^{\mathcal{W}_2}$. Consequently, the contact Hamiltonian dynamics on $\mathcal{T}^*\mathcal{P}^+(\mathcal{M}) \times \mathbb{R}$ can be represented in the reduced phase space $\Theta^* \times \mathbb{R}$, with the associated geodesic on $\mathcal{P}^+(\mathcal{M}) \times \mathbb{R}$ projected onto $\Sigma \times \mathbb{R}$ (see App. E.1).*

Proposition 2 allows us to transform the geodesic computation from the infinite-dimensional $\mathcal{P}^+(\mathcal{M})$ to a geodesic on a finite-dimensional parameterized space $\Sigma$, such that the resulting geodesic flow on $\Sigma \times \mathbb{R}$ evolves under the pullback of the Jacobi metric $T_\theta^* \tilde{g}_J = \Phi^{t_k} T_\theta^* g^{\mathcal{W}_2}$, where the scalar factor $\Phi^{t_k} = H(\rho^{t_k}, S^{t_k}, z^{t_k}) - \mathcal{F}(\rho^{t_k}) - \mathcal{B}(\rho^{t_k})$, encodes the kinetic energy. Specifying the time evolution of $H^{t_k}$ determines a unique parameterized bridge on $\Sigma$. This formulation enables a tractable computation of geodesic flows to solve the NCGSB problem. Although different parameterizations $\{\theta^k\}_{k=0}^K$ may define distinct coordinate systems on $\Sigma$, the geodesics solutions remain equivalent and share the same energy (Syrota et al., 2025).

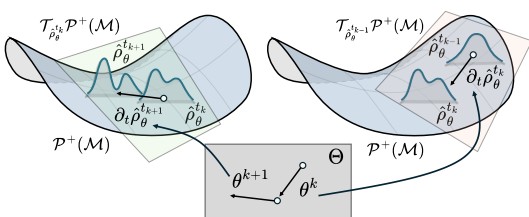

Figure 2: Visualization of the ResNet transformation. Two successive pushforwards $\rho_\theta^{t_{k-1}} \to \rho_\theta^{t_k} \to \rho_\theta^{t_{k+1}}$ on $\mathcal{P}^+(\mathcal{M})$ are shown as local updates $\partial_t \rho_\theta^{t_k}$, $\partial_t \rho_\theta^{t_{k+1}}$ on tangent spaces. Each update is parameterized by $\theta^k, \theta^{k+1} \in \Theta$, defining local coordinates on $\mathcal{T}\mathcal{P}^+(\mathcal{M})$. This coordinate system is not unique.

**Geodesic computation.** The *contact Wasserstein geodesic* (CWG) corresponds to the discrete path $\{\rho_\theta^{t_k}\}_{k=0}^K$, that approximates the NCGSB solution. This is trained to reconstruct the available marginals while minimizing the geodesic energy under the pullback Jacobi metric $T_\theta^* \tilde{g}_J = \Phi^{t_k} T_\theta^* g^{\mathcal{W}_2}$. The initial and final marginals, $\rho_a$ and $\rho_b$, are enforced at the path endpoints, corresponding to the ResNet outputs at times $t_0$ and $t_K$. When available, intermediate marginals $\rho^m$ must appear at time points matching the ResNet discretization for the condition to be enforced.

CWG training happens in two stages (Alg. 1 in App. E.2): (1) we optimize the first ResNet block to match the initial marginal $\rho_a$, and (2) we find the optimal path by minimizing the loss,

$$\ell = \underbrace{d_{\mathcal{W}_2}^2(\rho_\theta^{t_K}, \rho_b)}_{\text{Terminal marginal}} + \underbrace{\sum_{m=1}^M d_{\mathcal{W}_2}^2(\rho_\theta^{t_{k_m}}, \rho_m)}_{\text{Intermediate marginals}} + \underbrace{\sum_{k=1}^K \Phi^{t_k} d_{\mathcal{W}_2}^2(\rho_\theta^{t_k}, \rho_\theta^{t_{k-1}})}_{\text{Energy minimization}}. \tag{9}$$

Here $d_{\mathcal{W}_2}$ denotes the Wasserstein-2 distance between probability distributions. In practice, this distance is approximated using empirical estimators based on samples drawn from the distributions. Convergence of the algorithm is guaranteed in App. E.3. Its complexity is $\mathcal{O}\big(NK(T_{\text{sh}} + D(LW + \log N))\big)$, scaling linearly in dimension $D$ and nearly linearly in batch size $N$ (see App. E.5), rather than exponentially or quadratically (Hong et al., 2025). Unlike Chen et al. (2023); Shen et al. (2025), our CWG avoids costly iteration loops and is only weakly affected by the number of marginals.

**Guided contact Wasserstein geodesics.** In the conditional setting, the Lagrangian action dynamics (4b) in NCGSB (4) is augmented as in equation 5. Here, the scaling factor $\Phi^{t_k} = H(\rho^{t_k}, S^{t_k}, z^{t_k}) - \mathcal{F}(\rho^{t_k}) - \mathcal{B}(\rho^{t_k})$ of the pullback Jacobi metric is augmented with the guidance term $\|y - f(x^{t_s})\|^2$, to enforce the constraint $y = f(x^{t_s})$ at time $t_s$ of the generative process. Under the ResNet parameterization, the desired distribution is approximated by $x^{t_{k_s}} \approx x^{t_s}$ at time step $t_{k_s}$, and the Jacobi metric is modified as $\tilde{g}'_J = \big(\Phi^{t_k} + \|y - f(x^{t_s})\|^2\big) g^{\mathcal{W}_2}$, with $x^{t_{k_s}} \sim \rho_\theta^{t_{k_s}}$. This penalizes geodesics crossing undesired regions at $t_{k_s}$. The loss for the guided optimization is

$$\ell = d_{\mathcal{W}_2}^2(\rho_\theta^{t_K}, \rho_b) + \sum_{m=1}^M d_{\mathcal{W}_2}^2(\rho_\theta^{t_{k_m}}, \rho_m) + \sum_{k=1}^K \big(\Phi^{t_k} + \|y - f(x^{t_s})\|^2\big) d_{\mathcal{W}_2}^2(\rho_\theta^{t_k}, \rho_\theta^{t_{k-1}}) + d_{\mathcal{W}_2}'^2(\rho_\theta^{t_{t_s}}, \rho_s) \tag{10}$$

where the modified distance $d_{\mathcal{W}_2}'^2$ measures deviations between the generated distribution $\rho_\theta^{t_{t_s}}$ and the intermediate marginal $\rho_s$ at $t_s$, while incorporating the penalty for samples $x_s$ that violate the guidance constraint $y = f(x_s)$, c.f. App. F.1. In practice, this loss is optimized through a fine-tuning procedure applied to a model initially trained without any guidance.

**Proof of concept.** We demonstrate our framework on a 2D distribution-matching task and guided generation setting using the Two-Moons and Checkerboard benchmarks (Holderrieth & Erives, 2025). These lack intermediate marginals $\{\rho_m\}_{m=1}^M$, and the initial distribution $\rho_a$ coincides with the reference distribution $\lambda$. Hence, only the second step of Alg. 1 is needed. Figure 3 shows that our method successfully generates the target distributions, and steers the generation to samples confined to a subset of the target space (here, the upper half). This guided behavior is achieved via the term $\|y - f(x^{t_s})\|^2$, with $t_s = 1$, $f$ measuring 2D sample positions, and $y$ defining the admissible region.

## 6 RESULTS

We benchmark our approach against four established baselines summarized in Table 1. Further details of the experimental setups are provided in App. G.1.

| Method | GSB | mmSB | Energy variation | Image Gen. | Guided Gen. |
|---|---|---|---|---|---|
| DSBM (Shi et al., 2023) | ✗ | ✗ | ✗ | ✓ | ✗ |
| SB-Flow (Bortoli et al., 2024) | ✗ | ✗ | ✗ | ✓ | ✗ |
| GSBM (Liu et al., 2024) | ✓ | ✗ | ✗ | ✓ | ✗ |
| SBIRR (Shen et al., 2025) | ✗ | ✓ | ✗ | ✗ | ✗ |
| DM-SB (Chen et al., 2023) | ✗ | ✓ | ✗ | ✗ | ✗ |
| CWG (ours) | ✓ | ✓ | ✓ | ✓ | ✓ |

Table 1: Comparison of our CWG with baselines designed to address various SB variants and types of problems.

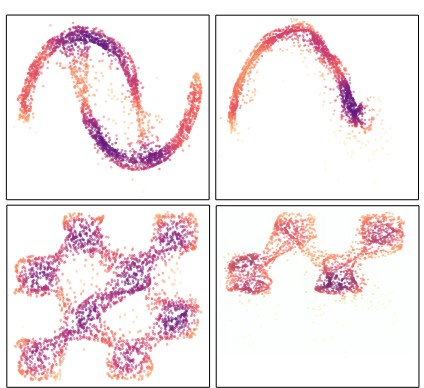

Figure 3: Two-Moons (top) and Checkerboard (bottom) benchmarks with guided variants (right).

Figure 4: LiDAR Manifold Navigation: CWG before and after guidance (top), CWG vs GSBM (bottom).

**LiDAR manifold navigation.** First, we tackle a standard GSB task: computing a bridge evolving on a geometric manifold. We use the LiDAR scan of Mount Rainier (OpenTopography, 2025) as the reference surface, and we aim to connect two marginals while remaining on the manifold and favoring low-altitude regions. These conditions are encoded into the potential function $U$ (see App. G.2). In this experiment, we do not model a physical system but instead compute an optimal transport map between the marginals under a conservative setting. Unlike our approach, DSBM and GSBM iteratively fit a deterministic path between the marginals, falling short on representing a posterior distribution. As a result, there is no guarantee that a Gaussian path remains on the manifold (Fig. 4). This leads to substantially higher-energy paths, as reflected by the Optimality metric in Table 2, and to inaccurate modeling of the target distribution, as indicated by the Feasibility metric (both metrics as defined in the baseline Liu et al. (2024)). In contrast, *our approach finds lower-energy solutions that respect the marginal constraints and converges significantly faster*. Furthermore, our method uniquely supports guided generation, illustrated here by steering the probabilistic path to the right side of the mount (Fig. 4; see App. G.2 for quantitative results).

Table 2: Optimality (67a) (↓), Feasibility (67b) (↓), and training time (tt) (↓) in LiDAR Navigation.

| Metric | CWG (ours) | GSBM | DSBM |
|---|---|---|---|
| Optimality | $\mathbf{1.40}_{\pm\mathbf{0.02}}$ | $2.18_{\pm0.02}$ | $4.16_{\pm0.01}$ |
| Feasibility | $\mathbf{0.06}_{\pm\mathbf{0.01}}$ | $0.83_{\pm0.01}$ | $0.97_{\pm0.01}$ |
| tt (s) | $\mathbf{280}_{\pm\mathbf{20}}$ | $1570_{\pm50}$ | $1340_{\pm50}$ |

Table 3: Wasserstein error at validation (↓) and training time (tt) (↓) in Single Cell Sequencing.

| Metric | CWG (ours) | DM-SB | SBIRR |
|---|---|---|---|
| $d_{\mathcal{W}_2}(x^{t_1})$ | $\mathbf{1.11}_{\pm\mathbf{0.06}}$ | $2.25_{\pm0.01}$ | $1.92_{\pm0.02}$ |
| $d_{\mathcal{W}_2}(x^{t_3})$ | $\mathbf{0.33}_{\pm\mathbf{0.02}}$ | $1.64_{\pm0.03}$ | $1.86_{\pm0.02}$ |
| tt (s) | $\mathbf{710}_{\pm\mathbf{30}}$ | $38120_{\pm1100}$ | $1740_{\pm40}$ |

**Single cell sequencing.** Next, we reconstruct stem cell differentiation dynamics from a series of isolated cellular snapshots. We use the Embryoid Body (EB) dataset from Moon et al. (2019), which tracks cell state progression across five developmental stages $[t_0, t_1, t_2, t_3, t_4]$. Cell differentiation is fundamentally a non-conservative biological process (Zeevaert et al., 2020; Kinney et al., 2014) and the ability to model energy-varying bridges is essential. To evaluate generalization in regions with no available data, we split the dataset into a training set $[t_0, t_2, t_4]$ and a validation set $[t_1, t_3]$. Accordingly, the former contains the distributions $\{\rho_a, \rho_{m_2}, \rho_b\}$, while the latter contains $\{\rho_{m_1}, \rho_{m_3}\}$. The geometry of the training distributions is encoded in the potential function $U$, which penalizes paths that stray from the observed data manifold. Minimizing $U$ ensures the learned bridge remains close to the data manifold, enabling effective generalization. The combination of the data manifold guidance and an energy-varying bridge allows our approach to outperform other mmSB baselines in both reconstruction accuracy and computation time. Quantitative results are reported in Table 3, with additional details and an ablation study on the importance of energy variation provided in App. G.3.

**Image generation.** We also demonstrate our framework's applicability to image generation tasks. Given sequences of images capturing the time evolution of physical phenomena, the model aims, from a single input image, to predict the most likely terminal state of the system, along with realistic intermediate frames at unseen time steps. Specifically, we use the NOAA OISST v2 High Resolution Dataset (Huang et al., 2021), which provides daily sea surface temperature averages over multiple years, and the BridgeData V2 (Walke et al., 2023) dataset, containing image snapshots of robotic manipulation tasks. Our NCGSB problem can also be applied to unpaired image-matching tasks



Figure 5: Predictions generated by CWG (ours, top), GSBM (second), DSBM (third), and SB-Flow (fourth). The red row displays the corresponding reference samples.

Table 4: FID scores at validation steps ($\downarrow$) and training time (tt) ($\downarrow$) in Sea Prediction (2020-2024).

| Metric | CWG (ours) | GSBM | DSBM | SB-Flow |
|---|---|---|---|---|
| FID($x^{t_1}$) | $\mathbf{121_{\pm 6}}$ | $161_{\pm 5}$ | $242_{\pm 10}$ | $177_{\pm 4}$ |
| FID($x^{t_3}$) | $\mathbf{160_{\pm 7}}$ | $186_{\pm 7}$ | $236_{\pm 10}$ | $190_{\pm 7}$ |
| tt (60s) | $\mathbf{17_{\pm 1}}$ | $1227_{\pm 54}$ | $318_{\pm 15}$ | $83_{\pm 5}$ |

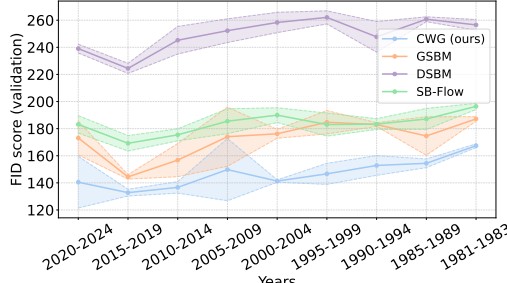

Figure 6: FID scores at validation time steps for four methods, evaluated over 9 tests (1981–2024). Our CWG scores are significantly lower than baselines.

using general image datasets that do not describe dynamical systems, such as MNIST, EMNIST, and Flickr-Faces-HQ (FFHQ) (Karras et al., 2019). For the high-resolution unpaired image-matching task on the FFHQ dataset, the SB is computed in a latent space after encoding the images with the pre-trained ALAE encoder (Pidhorskyi et al., 2020). For all the other datasets, we compute the SB directly in the ambient image space. To ensure that the generated frames remain faithful to the underlying data distribution, we introduce a potential function $U$, that penalizes deviations from the learned data manifold. Following the baseline (Liu et al., 2024), for samples $x^t \sim \rho^t$, $U(x^t)$ is defined as the reconstruction error, obtained via a VAE (Song & Itti, 2025). Details on the energy behavior and extended results are provided in Apps. G.4, G.5, G.6, and G.7.

For the sea temperature prediction task, we cluster data from 1981–2024 into five-year intervals. Using heatmaps from January, May, and September (i.e., $[t_0, t_2, t_4]$), our method predicts the temperature profiles of March and July (i.e., $[t_1, t_3]$). Our CWG produces cleaner, more accurate predictions than the baselines (Fig. 5). Since our framework operates efficiently in probability space and is not constrained by energy conservation, it achieves these results with an order of magnitude less computation (Table 4 for 2020-2024; Fig. 6 for all years). In the robotic task reconstruction, our model



Figure 7: Reconstructions from CWG (top), GSBM (middle), and DSBM (bottom). Red row shows the reference.

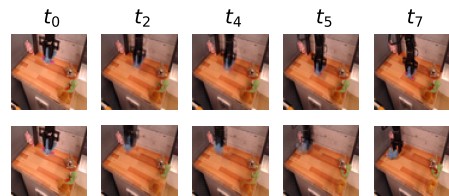

Figure 8: CWG outputs before (top) vs. after guidance (place the item left).

| Metric | CWG(ours) | GSBM | DSBM | SB-Flow |
|---|---|---|---|---|
| FID | $\mathbf{19_{\pm 1}}$ | $40_{\pm 2}$ | $150_{\pm 1}$ | $73_{\pm 1}$ |
| tt (h) | $\mathbf{0.5_{\pm 0}}$ | $25.3_{\pm 2.2}$ | $7.6_{\pm 0.7}$ | $1.4_{\pm 0.1}$ |

Table 5: FID score ($\downarrow$) and training time (tt) ($\downarrow$) in the robotic task reconstruction.

| Metric | Standard | Guidance |
|---|---|---|
| Centroid | $35.8_{\pm 11.1}$ | $22.3_{\pm 2.9}$ |
| FID | $19.52_{\pm 0.78}$ | $23.77_{\pm 1.94}$ |

Table 6: Item centroid position (px) and FID before vs. after guidance.

generates realistic intermediate frames connecting the initial and final states of a robot's reaching motion (Fig. 7), and demonstrates consistently robust performance, outperforming baselines in image quality (Table 8). Also, Fig. 8 shows guided generation, where our model successfully steers the

placing motion task toward a target location on the left side of the table. This is achieved with only a minimal drop in image quality, maintaining a clear advantage over competing methods (Table 6).

Regarding the unpaired image transfer experiments, we first evaluate the MNIST-to-EMNIST transfer (Table 38, App. G.7) and validate the guided-bridge capabilities through a Gaussian deblurring task. We then present the FFHQ transfer experiment, in which the SB maps images from a starting *adult* distribution $\rho_a$ to its closest analogue in a target *children* distribution $\rho_b$. As shown in Table 7, while GSBM and DSBM produce images visually similar to the input (Optimality), they fail to satisfy the boundary conditions of (2) and achieve poor Feasibility. In contrast, our CWG model consistently generates images within the children distribution $\rho_b$, with average predicted ages below 18, as illustrated in Fig. 10.

Table 7: Metrics for the FFHQ transfer experiment: Training time (tt) ($\downarrow$), Optimality (72a) ($\downarrow$), measuring the geodesic distance to assess the transport cost between the two marginals ($\rho_a,\rho_b$), and Feasibility (72b) ($\downarrow$), indicating how well the marginals are preserved, i.e., how closely the bridge endpoint aligns with $\rho_b$. Metric < 18 ($\downarrow$) indicates confidence that the final predicted images satisfy the boundary constraints.

| Metric | CWG | GSBM | DSBM | SB-Flow |
|---|---|---|---|---|
| Optimality | $218.1_{\pm2.6}$ | $206.2_{\pm2.9}$ | $\mathbf{198.0_{\pm2.7}}$ | $237.4_{\pm2.2}$ |
| Feasibility | $\mathbf{4.332_{\pm0.526}}$ | $6.839_{\pm0.753}$ | $7.780_{\pm0.801}$ | $21.748_{\pm0.498}$ |
| < 18 (p-value) | $\mathbf{2.1\times10^{-9}}$ | $6.6\times10^{-1}$ | $6.1\times10^{-1}$ | $1.1\times10^{-2}$ |
| tt (s) | $\mathbf{930_{\pm30}}$ | $2650_{\pm30}$ | $2530_{\pm30}$ | $1490_{\pm30}$ |

Input

CWG(ours)

GSBM

DSBM

SB − Flow

Figure 9: Adult → Child image generation on the FFHQ transfer experiment.

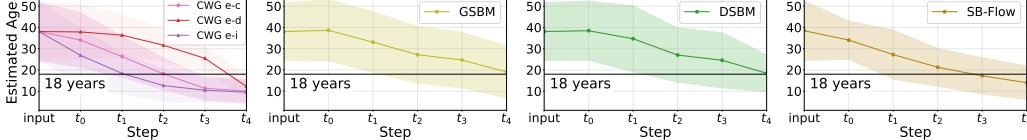

Figure 10: Average age predictions at each generation time step for the images shown in Figure 9, comparing the baseline models with three CWG variants: energy-conserving, energy-increasing, and energy-decreasing.

# 7 CONCLUSION

Our work is motivated by the need to model intermediate time steps of Schrödinger bridges (SBs), arising from the underlying dynamics of the observed physical system. As standard SBs conserve energy across time, they cannot meaningfully encode such dynamics. To overcome this, we introduced the *non-conservative generalized Schrödinger bridge* (NCGSB), which extends the usual Hamiltonian to its non-conservative counterpart, the *contact Hamiltonian*, allowing energy to vary. Through a geometric perspective, we show that the NCGSB is equivalent to geodesics on *contact Wasserstein manifolds*. This link leads to a non-iterative and near-linear time algorithm for computing the non-conservative bridge, which can practically be realized by a ResNet-like construction, easing its implementation. We show that these theoretical contributions lead to a SB framework that is not only more expressive but also significantly faster than existing approaches, as validated by the significant improvements achieved across a range of diverse tasks.

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

## A    EXTENDED STATE-OF-THE-ART ON SCHRÖDINGER BRIDGE SOLVERS

Existing methodologies for addressing the Schrödinger Bridge problem can be broadly divided into two main categories, depending on their solution strategy: those that directly fit the stochastic dynamics on the probabilistic manifold, and those that leverage the analytic optimality conditions of the problem to solve it. We review them in the sequel.

**Schrödinger Bridge Solvers via Dynamics Parametrization.**    Traditional SB solvers often use Iterative Proportional Fitting (IPF) (Kullback, 1968; Léonard, 2013), which alternates forward and backward updates to successively match the initial and terminal marginals. However, IPF is computationally expensive, as it stores full trajectories, and it suffers from error accumulation, numerical instability, and reliance on strong priors (Vargas et al., 2021; Gushchin et al., 2024). Recent matching-based approaches (Bortoli et al., 2024; Shi et al., 2023; Gushchin et al., 2024; Peluchetti, 2023) improve scalability and robustness by learning time-reversed drifts via Markovian projections, circumventing the need for full trajectory storage and mitigating the IPF discretization errors. While Liu et al. (2024) extended this idea to the GSB setting, their assumption of Gaussian probability paths limits the model's expressivity. To overcome this, Tang et al. (2025) proposed branched dynamics using Gaussian mixtures. This enables more flexible path structures but at the expense of higher computational cost. For mmSB, iterative reference refinement with piecewise SB interpolation (Shen et al., 2025) suffers from inconsistencies in global trajectory construction due to its piecewise nature and shows high sensitivity to the choice of initial reference process. Alternatively, Tong et al. (2020) proposed a continuous normalizing flow for deterministic interpolation, removing noise from the reference process but preventing the construction of a true probabilistic bridge.

**Schrödinger Bridge Solvers via Optimality Conditions.**    The optimality conditions of the SB problem take the form of dynamical equations on a Hamiltonian phase space, driven by dual potential functions (Chow et al., 2020). These conditions allow the exact dynamics to be recovered via integration and provide a flexible framework for generalization through modifications of the Hamiltonian. Furthermore, the state-dependent nature of the Hamiltonian framework offers a natural way to obtain Markovian approximations of a stochastic process. Vargas et al. (2021) and Chen et al. (2022) leveraged this view and solved the SB problem using control- and likelihood-based approaches, both employing iterative forward-backward updates on the Hamiltonian dynamics. Liu et al. (2022) extended this idea to the GSB setting, although without convergence guarantees. However, a critical bottleneck of these and derived methods is that a tractable integration of optimality conditions relies on iterative updates of a reference process. For example, Buzun et al. (2025) improved stability by directly modeling the dual potential and minimizing residuals of the Hamiltonian conditions, yet iterative updates incur significant computational overhead and may destabilize training due to the dependence on self-generated samples (Bertrand et al., 2024). This issue persists in mmSB settings (Theodoropoulos et al., 2025; Hong et al., 2025). Even when belief propagation is used to reduce time complexity (Chen et al., 2023), scaling to high dimensions remains poor. Therefore, while leveraging the optimality conditions offers clear advantages, it remains essential to develop computationally efficient, non-iterative algorithms with favorable scaling properties.

# B  NOTATION

Table 8: Summary of mathematical notation

| Notation | Description | Context |
|---|---|---|
| **Probability Distributions and Manifolds** | | |
| $\mathcal{M}$ | Underlying data manifold. | Sec. 3 |
| $\mathcal{P}^+(\mathcal{M})$ | The space of smooth, positive probability density functions supported on $\mathcal{M}$ (the Wasserstein manifold). | Sec. 3 |
| $\rho(x), \rho^t(x)$ | A probability density function and its time-dependent path. | Sec. 3 |
| $\rho_a, \rho_b$ | The initial and final endpoint marginal distributions. | Sec. 3 |
| $\{\rho_m\}_{m=1}^M$ | A set of intermediate marginal constraints at times $\{t_m\}_{m=1}^M$. | Eq. (2) |
| $\lambda$ | A fixed reference measure (e.g., standard Gaussian). | Sec. 5 |
| **Transport Dynamics** | | |
| $v^t(x)$ | The drift term (control variable) in the Fokker-Planck equation. | Eq. (2) |
| $U(x)$ | The external potential function (state cost). | Eq. (2) |
| $\epsilon$ | Scaling factor for the stochastic diffusion term in the Fokker-Planck equation (related to entropy regularization). | Eq. (2) |
| $J$ | The objective functional (cost) to be minimized both in the mmGSB problem and in the NCGSB problem. | Eq. (2) |
| $S^t(x)$ | The potential function/Lagrange multiplier that acts as a canonical momentum, with $\nabla_x S^t(x) = v^t(x)$. | Eq. (3) |
| $z^t$ | The augmented scalar state representing the Lagrangian action. | Eq. (5) |
| $\gamma$ | The damping factor in the Non-Conservative GSB formulation. | Eq. (5) |
| **Hamiltonian variables** | | |
| $H(\rho^t, S^t)$ | The Hamiltonian function, defined as the sum of kinetic ($\mathcal{K}$) and potential ($\mathcal{F}$) energy: $H(\rho^t, S^t) = \mathcal{K}(\rho^t, S^t) + \mathcal{F}(\rho^t)$. | Eq. (3) |
| $H(\rho^t, S^t, z^t)$ | The contact Hamiltonian function, defined as $H(\rho^t, S^t, z^t) = \mathcal{K}(\rho^t, S^t) + \mathcal{F}(\rho^t) + \mathcal{B}(\rho^t) + \gamma z^t$. | Eq. (6) |
| $\mathcal{K}$ | Kinetic energy of the transport map. | Sec. 3 |
| $\mathcal{F}$ | Potential energy, $\mathcal{F} = -U - I$. | Sec. 3 |
| $I$ | Fisher information term arising from entropy regularization, defined as $I = \int_{\mathcal{M}} \|\nabla_x \log \rho^t(x)\|^2 \rho^t(x)\, dx$. | Sec. 3 |
| $\mathcal{B}$ | Entropy term, defined as $\mathcal{B} = \int_{\mathcal{M}} \varepsilon(\log \rho^t(x) - 1)\rho^t(x)\, dx$, producing an additional diffusion in the contact Hamiltonian dynamics. | Sec. 4 |
| **Guidance Terms** | | |
| $\|y - f(x^{t_s})\|^2$ | Guidance loss terms, penalizing the difference between the detected features and the desired values $y$ | Eq. (5) |
| $f(x^{t_s})$ | Feature function applied to the sample $x^{t_s}$ at time $t_s$. It can indicate a class, describe a property of the sample, or quantify the sample's value depending on the task. | Eq. (5) |
| **Geometric Terms** | | |
| $g^{\mathcal{W}_2}(\cdot, \cdot)$ | The Wasserstein metric tensor on $\mathcal{P}^+(\mathcal{M})$. | Eq. (1) |
| $d_{\mathcal{W}_2}(\rho_a, \rho_b)$ | The Wasserstein distance between $\rho_a$ and $\rho_b$. | Sec. 3 |
| $\Delta_{\rho^t}$ | The weighted Laplacian operator, $\Delta_{\rho^t} = -\nabla_x \cdot (\rho^t \nabla_x)$. | Eq. (1) |
| $g_J$ | The Jacobi metric, $g_J = (H - \mathcal{F})g^{\mathcal{W}_2}$, which is minimized by the optimal bridge geodesic for the mmGSB problem. | Sec. 3 |
| $\tilde{g}_J$ | The Jacobi metric, $\tilde{g}_J = (H - \mathcal{F} - \mathcal{B})g^{\mathcal{W}_2}$, which is minimized by the optimal bridge geodesic for the NCGSB problem. | Sec. 5 |
| **ResNet parameterization** | | |
| $T_{\{\theta^k\}_{k=0}^K}$ | ResNet model. | Eq. (7) |
| $T_{\theta^k}$ | A parametrized pushforward map (one block of the ResNet). | Eq. (7) |
| $\theta^k$ | The parameters associated with the pushforward map $T_{\theta^k}$. | Eq. (7) |
| $\Phi^k$ | Scaling term for the Wasserstein distance, defined as $\Phi^{t_k} = H(\rho^{t_k}, S^{t_k}, z^{t_k}) - \mathcal{F}(\rho^{t_k}) - \mathcal{B}(\rho^{t_k})$. | Sec. 5 |

## C  EXTENDED PRELIMINARIES ON DIFFERENTIAL GEOMETRY

### C.1  HAMILTONIAN AND CONTACT HAMILTONIAN DYNAMICS

Hamiltonian and contact Hamiltonian dynamics are governed by specific energy constraints that can be analyzed via differential geometry as flows on specialized manifolds. Hamiltonian dynamics is energy-conserving and evolves on a symplectic manifold. Contact Hamiltonian dynamics is more general, allowing for variable energy levels, and takes place on a contact manifold. Their formal definitions and key differences are discussed next.

**Symplectic and Contact Structures.** Let $\mathcal{M}$ be a smooth compact manifold, and let $\mathcal{T}_x\mathcal{M}$ denote the tangent space at $x \in \mathcal{M}$. The collection of all the tangent spaces identifies the tangent bundle $\mathcal{T}\mathcal{M} = \cup_{x \in \mathcal{M}}\mathcal{T}_x\mathcal{M}$. A vector field $X : \mathcal{M} \to \mathcal{T}\mathcal{M}$ assigns a tangent vector $v$ to each point $x \in \mathcal{M}$. The set of all the vector fields over $\mathcal{T}\mathcal{M}$ is denoted as $\Gamma(\mathcal{T}\mathcal{M})$. A differential 1-form $\alpha : \mathcal{T}\mathcal{M} \to \mathbb{R}$ is a smooth map field acting on vectors of the tangent bundle. For a smooth function $f : \mathcal{M} \to \mathbb{R}$, the 1-form $\alpha = df$ generalizes the gradient from Euclidean spaces. Specifically, $df$ measures the variation of $f$ under an infinitesimal displacement on $\mathcal{M}$. This displacement is locally described by a starting point $x$ and a direction $v$, such that $(x, v) \in \mathcal{T}\mathcal{M}$. Alternatively, it can be globally expressed by a vector field $X$. The variation of $f$ along the vector field $X$ is given by $df(X)$. This variation is independent of the choice of reference frame. To preserve this invariance, $df$ must transform covariantly with $X$. Consequently, the 1-form $\alpha = df$ resides in the cotangent bundle $\mathcal{T}^*\mathcal{M}$, the dual space to $\mathcal{T}\mathcal{M}$. The symplectic and contact structures provide two distinct mechanisms for associating a 1-form to a vector field, thereby establishing connections between the tangent and cotangent bundles. By considering the dynamics governed by the vector field and the scalar function defining the 1-form, a relationship between these elements emerges, as illustrated in Fig. 11.

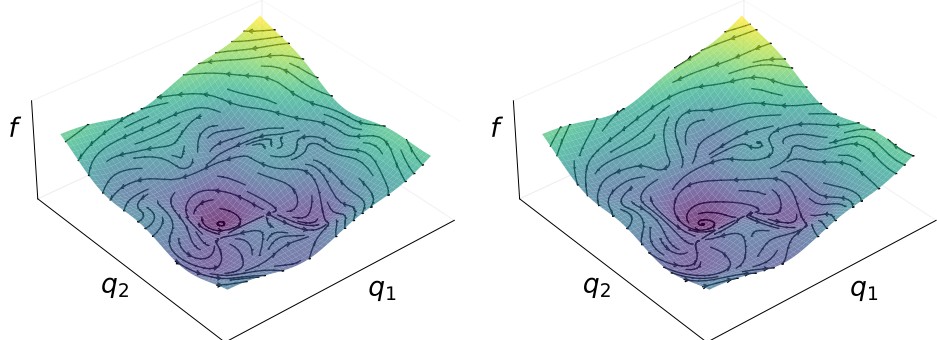

Figure 11: The same scalar function $f$, associated with the 1-form $\alpha = df$, gives rise to two distinct vector fields under the symplectic (left) and contact (right) geometric structures. The streamlines of these vector fields are illustrated on a representation of the state manifold. In symplectic geometry, the streamlines are tangent to the level curves of $f$, representing isoenergetic trajectories where $f$ remains constant, thus describing the dynamics of conservative systems. In contrast, in contact geometry, a single flow line can traverse different energy levels.

**Symplectic Geometry.** A differential 2-form $\omega : \mathcal{T}\mathcal{M} \times \mathcal{T}\mathcal{M} \to \mathbb{R}$ is a skew-symmetric, bilinear, and smooth field of maps acting on pairs of tangent vectors. A 2-form is called symplectic if it is both closed ($d\omega = 0$) and non-degenerate. The symplectic form lacks the properties required to define an inner product. However, it still establishes a fundamental relation between differential 1-forms and vector fields: Given a 1-form $df$, the symplectic form $\omega$ uniquely determines a vector field $X_f$ that is tangent to the level sets of $f$. This relation is defined by,

$$df(X) = \omega(X_f, X), \quad \forall X \in \Gamma(\mathcal{T}\mathcal{M}). \tag{11}$$

By definition, $f$ remains constant along the flow of $X_f$, which in turn preserves the symplectic form $\omega$, i.e., $\mathcal{L}_{X_f}\omega = 0$ where $\mathcal{L}_{X_f}$ denotes the Lie derivative (Silva, 2001). In this framework, the function $f$ is interpreted as a conserved energy, or equivalently, as a Hamiltonian $H$. The symplectic structure thereby endows $\mathcal{M}$ with a natural geometric framework for formulating Hamiltonian dynamics (Tokasi & Pickl, 2022). The pair $(\mathcal{M}, \omega)$ is referred to as a symplectic manifold. Notably, the non-degeneracy of $\omega$ implies that $\mathcal{M}$ must be even-dimensional.

**Contact Geometry.** While symplectic manifolds provide a geometric framework for modeling the dynamics of conservative systems in classical mechanics, a more general approach is required to describe non-conservative systems. This is addressed by contact manifolds, the odd-dimensional counterparts of symplectic manifolds (Geiges, 2001; Bravetti et al., 2017). A contact manifold is defined as $(\mathcal{M}, \eta)$, where $\mathcal{M}$ is an odd-dimensional smooth manifold, and $\eta$ is a non-degenerate 1-form known as the contact form (Geiges, 2008). The contact form satisfies the maximal non-integrability condition, meaning that the top-degree differential form $\eta \wedge (d\eta)^d \neq 0$ is nowhere vanishing on $\mathcal{M}$. This form is constructed by taking the exterior product of $\eta$ with the $d$-fold wedge product of its exterior derivative $d\eta$, i.e.,

$$(d\eta)^d = \underbrace{d\eta \wedge \cdots \wedge d\eta}_{d \text{ times}}. \tag{12}$$

The $(2d+1)$-form defines a volume form on $\mathcal{M}$, ensuring that the hyperplanes $\ker(\eta) \subset \mathcal{TM}$, constraining the dynamics on the contact manifold, do not form a foliation, i.e., they do not partition the manifold into lower-dimensional submanifolds (Geiges, 2001; 2008). Geometrically, this means that the contact distribution imposes non-holonomic constraints: it restricts the admissible directions of motion at each point without confining the dynamics to a fixed submanifold or energy level. This property is crucial for modeling systems where energy can change over time, enabling constraints on energy behavior without enforcing conservation.

Like symplectic geometry, contact geometry connects scalar functions to vector fields, enabling the description of dynamical systems (Zadra, 2023). Given an energy function $H : \mathcal{M} \to \mathbb{R}$, the dynamics on a contact manifold are defined by a contact Hamiltonian vector field $X_H$, as follows,

$$dH(X) = d\eta(X_H, X) - \mathcal{L}_{X_H} \eta(X), \ \forall X \in \Gamma(\mathcal{TM}). \tag{13}$$

Unlike symplectic geometry, where dynamics are confined to energy-preserving flows along the level sets of the Hamiltonian, contact geometry allows for an additional component of motion. Specifically, the dynamics on a contact manifold are not restricted to the term $d\eta(X_H, X)$, which lies tangent to the level sets of $H$, but also include a transverse component $\mathcal{L}_{X_H}\eta(X)$, arising from the non-degeneracy of the contact form. Consequently, while in symplectic geometry the symplectic form $\omega$ is strictly preserved, contact geometry allows the contact form $\eta$ to be preserved only up to a scaling factor $a \in \mathbb{R}$ (Bravetti et al., 2017).

## C.2 RIEMANNIAN AND JACOBI METRICS

The Jacobi metric $g_{\mathrm{J}}$ is a rescaled version of a Riemannian metric $g$ that allows Hamiltonian dynamics on the cotangent bundle $\mathcal{T}^*\mathcal{M}$ to be represented as geodesics on the Riemannian manifold $(\mathcal{M}, g)$. The construction is detailed below.

**The Riemannian Metric.** Let $\mathcal{M}$ be a smooth compact manifold. A Riemannian metric $g : \mathcal{TM} \times \mathcal{TM} \to \mathbb{R}$ is a smooth, symmetric, and positive-definite bilinear field of maps defined on pairs of vectors in the tangent bundle. This enables the introduction of an inner product on the tangent spaces of the manifold, allowing us to measure distances and curve lengths. For a smooth curve $x(t) : [t_0, t_1] \to \mathcal{M}$, the length $l$ w.r.t. the metric $g$ is $l = \int_{t_0}^{t_1} \sqrt{g(\dot{x}(t), \dot{x}(t))} dt$, where $\dot{x}(t) \in \mathcal{T}_{x(t)}\mathcal{M}$ is the vector tangent to the curve at $x(t)$. The curve minimizing this length between two points $x(t_0)$ and $x(t_1)$ on $\mathcal{M}$ is called a geodesic. Geodesics generalize straight lines in Euclidean space to curved spaces, representing the shortest paths in the geometry induced by $g$.

**The Jacobi Metric.** The geodesic flow $x(t)$ on a Riemannian manifold $(\mathcal{M}, g)$ lifts to the joint evolution of coordinates $(x(t), \alpha(x(t), \dot{x}(t))$ on the cotangent bundle $\mathcal{T}^*\mathcal{M}$ (Abraham & Marsden, 2008). This extended dynamics is governed by an energy function $H(x, \alpha) : \mathcal{T}^*\mathcal{M} \to \mathbb{R} = g^{-1}(\alpha, \alpha)$, which remains constant along the flow. A reparameterization $ds = \sqrt{H} dt$ links the trajectory of the integrated dynamical system at time $t$ on $\mathcal{T}^*\mathcal{M}$ with the length of the corresponding geodesic on $\mathcal{M}$. This framework reveals a fundamental connection between geodesic flows and Hamiltonian dynamics in the special case where the Hamiltonian consists solely of a kinetic energy term. The cotangent bundle $\mathcal{T}^*\mathcal{M}$ is naturally equipped with a symplectic structure, making it a symplectic manifold $(\mathcal{T}^*\mathcal{R}, \omega)$.

This formulation can be further generalized by introducing a potential energy function into the Hamiltonian, given by $H(x, \alpha) = g^{-1}(\alpha, \alpha) + \mathcal{F}(x)$. In this setting, the geodesic structure un-

derlying the Hamiltonian flow is determined by the Jacobi metric,

$$g_J = \big(H - \mathcal{F}(x)\big)\, g, \tag{14}$$

which rescales the original metric $g$ by a position-dependent conformal factor (Abraham & Marsden, 2008). The corresponding time reparameterization takes the form $ds = \sqrt{H - \mathcal{F}(x)}\, dt$, restoring the interpretation of the trajectory as a geodesic with respect to the metric $g_J$ (Udriște, 2000).

## D    INSIGHTS ON THE SCHRÖDINGER BRIDGE

### D.1    HAMILTONIAN STRUCTURE OF THE GENERALIZED SCHRÖDINGER BRIDGE

This part presents the derivation of the Hamiltonian structure of the mmGSB problem (2), introduced in section 3, obtained via the method of Lagrange multipliers. To transform the constrained optimization problem into an unconstrained one, we incorporate the Fokker–Planck equation, scaled by the Lagrange multiplier $S^t$, into the original running cost $\mathcal{L}$ (i.e., the Lagrangian), as follows,

$$J(v^t, \rho^t, S^t) = \int_0^1 \mathcal{L}(v^t, \rho^t, S^t)\, dt; \tag{15}$$

$$\mathcal{L}(v^t, \rho^t, S^t) = \int_{\mathcal{M}} \left( \frac{1}{2}\|v^t(x)\|^2 + U(x) \right) \rho^t(x)\, dx$$

$$+ \int_{\mathcal{M}} S^t(x) \underbrace{\big(\partial_t \rho^t(x) + \nabla_x \cdot (\rho^t(x)v^t(x)) - \varepsilon \Delta \rho^t(x)\big)}_{\text{Fokker-Planck equation}}\, dx. \tag{16}$$

The optimality conditions resulting from the extremization of the cost functional $J$ in equation (15) follow from the Euler–Lagrange equations, generalized to the setting of classical field theory (Blohmann, 2024). In this framework, the arguments of the Lagrangian $\mathcal{L}$, in equation (16), are viewed as smooth fields defined over space and time. By setting to zero the variations of $\mathcal{L}$ with respect to these fields, we obtain the stationarity conditions for $J$. For a generic field $\psi^t(x)$, the corresponding Euler–Lagrange equation takes the form,

$$d_\psi \mathcal{L} = \partial_\psi \mathcal{L} + \partial_t(\partial_{\partial_t \psi} \mathcal{L}) + \nabla_x \cdot (\partial_{\nabla_x \psi} \mathcal{L}) + \Delta_x(\partial_{\Delta_x} \mathcal{L}) = 0. \tag{17}$$

Applying equation (17) to equation (16) for the fields $v^t$, $\rho^t$, and $S^t$, we obtain the following system of optimality conditions,

$$d_v \mathcal{L} = v^t(x)\rho^t(x) - \rho^t(x)\nabla_x S^t(x) = 0 \implies v^t(x) = \nabla_x S^t(x), \tag{18a}$$

$$d_\rho \mathcal{L} = \frac{1}{2}\|v^t(x)\|^2 - \partial_t S^t(x) - \nabla_x S^t(x) \cdot v^t(x) - \varepsilon \Delta_x S^t(x) + U(x) = 0, \tag{18b}$$

$$d_S \mathcal{L} = \partial_t \rho^t(x) + \nabla_x \cdot (\rho^t(x)v(x)) - \varepsilon \Delta_x \rho^t(x) = 0. \tag{18c}$$

Substituting the expression for the optimal velocity from equation (18a) into equations (18b) and (18c), we obtain the following Hamiltonian system,

$$\partial_t \rho^t(x) = \partial_S H(\cdot) = -\nabla_x \cdot (\rho^t(x)\nabla_x S^t(x)) + \varepsilon \Delta \rho^t(x), \tag{19a}$$

$$\partial_t S^t(x) = -\partial_\rho H(\cdot) = -\frac{1}{2}\|\nabla_x S^t(x)\|^2 - \varepsilon \Delta_x S^t(x) + U(x), \tag{19b}$$

with the corresponding Hamiltonian function,

$$H(\rho^t, S^t) = \frac{1}{2}\int_{\mathcal{M}} \|\nabla_x S^t(x)\|^2 \rho^t(x)\, dx - \int_{\mathcal{M}} U(x)\rho^t(x)\, dx + \varepsilon \int_{\mathcal{M}} S^t(x)\Delta_x \rho^t(x)\, dx. \tag{20}$$

The Hamiltonian system (19) can be reformulated in a linear and decoupled form via the coordinate transformation from the original variables $(\rho^t, S^t)$ to an alternative canonical set $(\hat{\rho}^t, \hat{S}^t)$. This transformation, known as Hopf–Cole coordinate transformation (Léger & Li, 2021; Chow et al., 2020), is derived from the generating function $F$,

$$F(\rho, \hat{S}) = \rho(x)\hat{S} + \varepsilon \rho(x)(\log \rho(x) - 1), \tag{21}$$

and preserve the symplectic form, i.e., $dF = Sd\rho - \hat{S}d\hat{\rho}$. Following the standard generating-function rules (Chapter 18.3 Johns (2005)),

$$\hat{\rho}^t(x) = \partial_{\hat{S}}F(\cdot), \quad S^t(x) = \partial_\rho F(\cdot), \tag{22}$$

the new coordinates are explicitly given by,

$$\hat{\rho}^t(x) = \rho^t(x), \tag{23a}$$

$$\hat{S}^t(x) = S^t(x) - \varepsilon \log \rho^t(x). \tag{23b}$$

The Hopf–Cole transformation is well established in the literature, not only for simplifying the mathematical form of the equations, but also for enabling the design of efficient numerical integration schemes (Léger & Li, 2021). In our context, it is particularly advantageous for obtaining a Hamiltonian with separable kinetic and potential energy components. The Hamiltonian system (19) then becomes,

$$\partial_t \hat{\rho}^t(x) + \nabla_x \cdot (\hat{\rho}^t(x)\nabla_x \hat{S}^t(x)) = 0, \tag{24a}$$

$$\partial_t \hat{S}^t(x) + \varepsilon \frac{1}{\hat{\rho}^t(x)}\partial_t \hat{\rho}^t(x) = -\frac{1}{2}\left\|\nabla_x \hat{S}^t(x) + \varepsilon\nabla_x \log \hat{\rho}^t(x)\right\|^2$$
$$- \varepsilon\Delta_x \hat{S}^t(x) - \varepsilon^2\Delta_x \log \hat{\rho}^t(x) + U(x). \tag{24b}$$

Expanding the squared norm in equation (24b) and substituting $\partial_t \rho^t(x)$ from equation (24a) yields,

$$\partial_t \hat{S}^t(x) - \varepsilon\frac{1}{\hat{\rho}^t(x)}\nabla_x \cdot (\hat{\rho}^t(x)\nabla_x \hat{S}^t(x)) = -\frac{1}{2}\|\nabla_x \hat{S}^t(x)\|^2 - \frac{1}{2}\varepsilon^2\|\nabla_x \log, \hat{\rho}^t(x)\|^2$$
$$- \varepsilon\nabla_x \hat{S}^t(x)\nabla_x \log \hat{\rho}^t(x) - \varepsilon\Delta_x \hat{S}^t(x)$$
$$- \varepsilon^2\Delta_x \log \hat{\rho}^t(x) + U(x). \tag{25}$$

Using the identity,

$$\varepsilon\frac{1}{\hat{\rho}^t(x)}\nabla_x \cdot (\hat{\rho}^t(x)\nabla_x \hat{S}^t(x)) = \varepsilon\Delta_x \hat{S}^t(x) + \varepsilon\nabla_x \hat{S}^t(x)\nabla_x \log \hat{\rho}^t(x), \tag{26}$$

equation (25) simplifies to,

$$\partial_t \hat{S}^t(x) = -\frac{1}{2}\|\nabla_x \hat{S}^t(x)\|^2 - \varepsilon^2\frac{1}{2}\|\nabla_x \log \hat{\rho}^t(x)\|^2 - \varepsilon^2\Delta_x \log \hat{\rho}^t(x) + U(x). \tag{27}$$

This form reveals the emergence of the Fisher information, defined as,

$$I(\hat{\rho}^t(x)) = \int_{\mathcal{M}} \|\nabla_x \log \hat{\rho}^t(x)\|^2 \hat{\rho}^t(x)\, dx, \tag{28a}$$

$$\partial_{\hat{\rho}}I(\hat{\rho}^t(x)) = -2\Delta_x \log \hat{\rho}^t(x) - \|\nabla_x \log \hat{\rho}^t(x)\|^2. \tag{28b}$$

Thus, equations (24a) and (27) admit the linear decoupled Hamiltonian formulation described in equation (3),

$$\partial_t \hat{\rho}^t(x) = \partial_{\hat{S}}H(\cdot) = -\nabla_x \cdot (\hat{\rho}^t(x)\nabla_x \hat{S}_t(x));$$

$$\partial_t \hat{S}^t(x) = -\partial_{\hat{\rho}}H(\cdot) = -\frac{1}{2}\|\nabla_x \hat{S}^t(x)\|^2 + \frac{1}{2}\varepsilon^2\partial_{\hat{\rho}}I(\hat{\rho}^t(x)) + U(x),$$

governed by the Hamiltonian function,

$$H(\hat{\rho}^t, \hat{S}^t) = \underbrace{\frac{1}{2}\int_{\mathcal{M}} \|\nabla_x \hat{S}^t(x)\|^2 \hat{\rho}^t(x)\, dx}_{\text{Kinetic energy } \mathcal{K}} - \underbrace{\int_{\mathcal{M}} U(x)\hat{\rho}^t(x)\, dx - \frac{1}{2}\varepsilon^2 I(\hat{\rho}^t(x))}_{\text{Potential energy } \mathcal{F}} \tag{29}$$

In this formulation, the Fisher information contributes to the potential energy and encodes the effect of stochastic diffusion. Minimizing the Fisher information term promotes smoothness in the density and steers the Hamiltonian flow toward the target distribution $\rho^1$, providing greater robustness

due to the regularizing effect of diffusion. This mechanism has been studied in the literature and employed in control applications for its regularization properties (Chen et al., 2025).

## D.2 CONTACT HAMILTONIAN STRUCTURE OF THE NON-CONSERVATIVE GSB

Here we derive the contact Hamiltonian formulation of the NCGSB problem, introduced in equation (4) and discussed in section 4. The structure of the derivation closely mirrors that of the GSB in Appendix D.1, with one key distinction: the Lagrangian $\mathcal{L}$ now depends explicitly on the accumulated action $z^t$. Specifically, the augmented cost functional and Lagrangian are given by,

$$J(v^t, \rho^t, S^t, z^t) = \int_0^1 \mathcal{L}(v^t, \rho^t, S^t, z^t)\, dt; \tag{30}$$

$$\mathcal{L}(v^t, \rho^t, S^t, z^t) = \int_{\mathcal{M}} \left( \frac{1}{2}\|v^t(x)\|^2 + U(x) \right) \rho^t(x)\, dx - \gamma z^t$$
$$+ \int_{\mathcal{M}} S^t(x) \left( \partial_t \rho^t(x) + \nabla_x \cdot (\rho^t(x)v^t(x)) - \varepsilon \Delta \rho^t(x) \right)\, dx, \tag{31}$$

where the scalar $\gamma$ acts as a damping factor. Since $\mathcal{L}$ depends explicitly on the evolving action $z^t$, the problem lies outside the scope of classical variational calculus. Instead, it fits within the framework of non-conservative variational principles, where the cost functional $J$ evolves dynamically with the system. This is naturally addressed by the Herglotz variational principle, which extends the Euler–Lagrange equations to systems with dissipative effects. The optimality conditions obtained from the variations of $\mathcal{L}$, namely, the Herglotz-type Euler–Lagrange equations, for a generic field argument $\psi^t(x)$ in this case take the form,

$$d_\psi \mathcal{L} = \partial_\psi \mathcal{L} + \partial_t(\partial_{\partial_t \psi} \mathcal{L}) + \nabla_x \cdot (\partial_{\nabla_x \psi} \mathcal{L}) + \Delta_x(\partial_{\Delta_x} \mathcal{L}) - \partial_z \mathcal{L}\, \partial_{\partial_t \psi}\mathcal{L} = 0. \tag{32}$$

Applying equation (32) for the fields $v^t$, $\rho^t$, and $S^t$, and recovering the dynamics of $z^t$ from the optimization problem (4), we obtain the following system of optimality conditions,

$$d_v \mathcal{L} = v^t(x)\rho^t(x) - \rho^t(x)\nabla_x S^t(x) = 0 \implies v^t(x) = \nabla_x S^t(x), \tag{33a}$$

$$d_\rho \mathcal{L} = \frac{1}{2}\|v^t(x)\|^2 - \partial_t S^t(x) - \nabla_x S^t(x) \cdot v^t(x) - \varepsilon \Delta_x S^t(x) + U(x) - \gamma S^t(x) = 0, \tag{33b}$$

$$d_S \mathcal{L} = \partial_t \rho^t(x) + \nabla_x \cdot (\rho^t(x)v(x)) - \varepsilon \Delta_x \rho^t(x) = 0, \tag{33c}$$

$$\partial_t z^t - \int_{\mathcal{M}} \left( \frac{1}{2}\|v^t(x)\|^2 + U(x) \right) \rho^t(x)\, dx - \gamma z^t = 0. \tag{33d}$$

Compared to the optimality conditions for the GSB problem presented in equations (18), the set (33) includes an additional term, $-\gamma S^t(x)$, in equation (33b), which accounts for the dissipation term. By substituting the expression for the optimal velocity from equation (33a) into equations (33c), (33b), and equation (33d), we obtain the system of contact Hamiltonian dynamics,

$$\partial_t \rho^t(x) = \partial_S H(\cdot) = -\nabla_x \cdot (\rho^t(x)\nabla_x S^t(x)) + \varepsilon \Delta_x \rho^t(x), \tag{34a}$$

$$\partial_t S^t(x) = -\partial_\rho H(\cdot) - S^t(x)\partial_z H(\cdot) = -\frac{1}{2}\|\nabla_x S^t(x)\|^2 - \varepsilon \Delta_x S^t(x) + U(x) - \gamma S^t(x), \tag{34b}$$

$$\partial_t z^t = S^t(x)\partial_S H(\cdot) - H(\cdot) = \int_{\mathcal{M}} \left( \frac{1}{2}\|\nabla_x S^t(x)\|^2 + U(x) \right) \rho^t(x)\, dx + \gamma z^t, \tag{34c}$$

with the associated contact Hamiltonian function given by,

$$H(\rho^t, S^t, z^t) = \frac{1}{2}\int_{\mathcal{M}} \|\nabla_x S^t(x)\|^2 \rho^t(x)\, dx - \int_{\mathcal{M}} U(x)\rho^t(x)\, dx + \varepsilon \int_{\mathcal{M}} S^t(x)\Delta_x \rho^t(x)\, dx - z^t. \tag{35}$$

In this case as well, it is beneficial to derive a decoupled and linearized representation of the dynamics. To this end, we perform a coordinate transformation from the original variables $(\rho^t, S^t, z^t)$ to an alternative canonical set $(\hat{\rho}^t, \hat{S}^t, \hat{z}^t)$, while preserving the contact structure,

$$dz - S d\rho = d\hat{z} - \hat{S} d\hat{\rho}. \tag{36}$$

This is achieved via a contact transformation generated by a generating function $F$, defined as,

$$F(\rho, \hat{S}, \hat{z}^t) = \rho(x)\hat{S} + \varepsilon\rho(x)(\log\rho(x) - 1) + \hat{z}^t, \tag{37}$$

which induces the following coordinate change according to the contact generating rules in Struckmeier & Redelbach (2008),

$$\hat{\rho}^t(x) = \partial_{\hat{S}}F(\cdot), \tag{38a}$$

$$S^t(x) = \partial_\rho F(\cdot), \tag{38b}$$

$$\hat{z}^t = F(\cdot) - \hat{S}\hat{\rho}. \tag{38c}$$

As a result, the transformed variables are given by

$$\hat{\rho}^t(x) = \rho^t(x), \tag{39a}$$

$$\hat{S}^t(x) = S^t(x) - \varepsilon\log\rho^t(x), \tag{39b}$$

$$\hat{z}^t = z^t + \varepsilon\rho^t(x)\left(\log\rho^t(x) - 1\right). \tag{39c}$$

By substituting the new coordinates (39) into the dynamical system (34), we obtain the following transformed contact Hamiltonian dynamics,

$$\partial_t\hat{\rho}^t(x) = \partial_{\hat{S}}H(\cdot) = -\nabla_x \cdot (\hat{\rho}^t(x)\nabla_x\hat{S}^t(x)),$$

$$\partial_t\hat{S}^t(x) = -\partial_{\hat{\rho}}H(\cdot) - \hat{S}^t(x)\partial_{\hat{z}}H(\cdot) = -\frac{1}{2}\|\nabla_x\hat{S}^t(x)\|^2 + \frac{1}{2}\varepsilon^2\partial_{\hat{\rho}}I(\hat{\rho}^t(x)) + U(x)$$
$$- \gamma\hat{S}^t(x) - \varepsilon\gamma\log\hat{\rho}^t(x),$$

$$\partial_t\hat{z}^t = \hat{S}^t(x)\partial_{\hat{S}}H(\cdot) - H(\cdot) = \int_{\mathcal{M}}\left(\frac{1}{2}\|\nabla_x\hat{S}^t(x)\|^2 + U(x)\right)\hat{\rho}^t(x)\,dx + \frac{1}{2}\varepsilon^2 I(\hat{\rho}^t)$$
$$- \int_{\mathcal{M}}\varepsilon\gamma(\log\hat{\rho}^t(x) - 1)\hat{\rho}^t(x)\,dx - \gamma\hat{z}^t,$$

with the associated contact Hamiltonian function given by,

$$H(\hat{\rho}^t, \hat{S}^t, \hat{z}^t) = \underbrace{\frac{1}{2}\int_{\mathcal{M}}\|\nabla_x\hat{S}^t(x)\|^2\hat{\rho}^t(x)\,dx}_{\text{Kinetic energy }\mathcal{K}} + \underbrace{\gamma\hat{z}^t}_{\text{Non-conservative potential}}$$

$$\underbrace{-\int_{\mathcal{M}}U(x)\hat{\rho}^t(x)\,dx - \frac{1}{2}\varepsilon^2 I(\hat{\rho}^t)}_{\text{Potential energy }\mathcal{F}} + \underbrace{\int_{\mathcal{M}}\varepsilon\gamma\left(\log\hat{\rho}^t(x) - 1\right)\hat{\rho}^t(x)\,dx}_{\text{Entropy }\mathcal{B}}. \tag{40}$$

### D.3 GUIDED SCHRÖDINGER BRIDGE

We consider the bridge $\rho^t$, computed between the terminal marginals $\rho_a$ and $\rho_b$, using any variant of the SB problem (e.g., GSB, mmSB, NCGSB). This process can be modified to enforce desired conditions $y$ at any chosen time $t_s$ defined as,

$$y = f(x^{t_s}), \quad x^{t_s} \sim \rho^{t_s}, \tag{41}$$

while preserving the underlying data manifold. Conditioning in this way modifies the probability flow $\rho^t$ (Guo et al., 2024) as,

$$\rho^t(x \mid y) = \frac{1}{Z} \rho^t(x) \, e^{-\|y - f(x^{t_s})\|^2}, \tag{42}$$

where $Z$ is a normalization constant, and $x^{t_s}$ denotes a sample from the predicted prescribed distribution $\rho^{t_s}(x—y)$. This weight biases the generation toward samples that satisfy the desired property $y$. In a dynamical setting, we perform this conditioning by incorporating a control term $G^t$ into the Fokker–Planck dynamics,

$$\partial_t \rho^t(x) + \nabla_x \cdot \left[\rho^t(x) \left(v^t(x) + G^t(x)\right)\right] = \varepsilon \Delta_x \rho^t(x). \tag{43}$$

A naive choice such as $G^t(x) \propto \nabla_{x^t} f(x^t)$ often drives the dynamics off the data manifold, producing unrealistic samples far from it and the target distribution $\rho_b$. Instead, Bayes' rule provides the correct structure of the guidance term. The gradient of the conditional log-likelihood decomposes as,

$$\nabla_{x^t} \log \rho^t(x^t \mid y) = \nabla_{x^t} \log \rho^t(x^t) + \underbrace{\nabla_{x^t} \log \rho^t(y \mid x^t)}_{\text{estimated by } G^t}. \tag{44}$$

Substituting the conditional form from equation (42) yields,

$$G^t(x) = \nabla_{x^t} \log e^{-\|y - f(x^{t_s})\|^2} = -\nabla_{x^t} \|y - f(x^{t_s})\|^2. \tag{45}$$

Specifically, in the NCGSB framework (4), this guidance is incorporated directly into the drift $v^t$ by adding the penalty $\|y - f(x^{t_s})\|^2$ to the Lagrangian in the action dynamics constraint (4b),

$$
\begin{aligned}
\min_{v^t} \quad & J(v^t, \rho^t) = \int_0^1 \partial_t z^t \, dt, \\
\text{s.t.} \quad & \partial_t z^t = \int_{\mathcal{M}} \left(\frac{1}{2}\|v^t(x)\|^2 + U(x) + \|y - f(x^{t_s})\|^2\right) \rho^t(x) \, dx - z^t, \\
& \partial_t \rho^t(x) + \nabla_x \cdot \left(\rho^t(x)\, v^t(x)\right) = \varepsilon \Delta_x \rho^t(x), \\
& \rho^0 = \rho_a, \quad \rho^1 = \rho_b, \quad \rho^{t_m} = \rho_m, \quad \forall\, m \in \{1, \ldots, M\}.
\end{aligned}
\tag{46}
$$

Here, the inclusion of $\|y - f(x^{t_s})\|^2$ in the Lagrangian produces the desired $-\nabla_{x^t} \|y - f(x^{t_s})\|^2$ correction in the drift, while preserving the Schrödinger bridge structure and constraints.

## E RESNET PARAMETERIZATION FOR DISCRETE GEODESICS

### E.1 GEOMETRIC INTERPRETATION

As stated in the main paper, our objective is to compute a geodesic $\rho^t$ on $\mathcal{P}^+(\mathcal{M})$, induced by the contact Hamiltonian dynamics, that is constrained to pass through a set of observed marginals $\{\rho_a, \rho_m, \rho_b\}$ (i.e., discretized distributions along the probability path). These constraints naturally lead to a discretized parameterization of $\rho^t$, where the overall density transformation is modeled as a composition of maps, each connecting a pair of consecutive observations. In this context, a ResNet architecture is ideally suited for this problem, as its sequential block structure directly mirrors this piecewise, compositional nature of the approximated geodesic.

Geometrically, each parameterized pushforward defines a vector $\partial_t \rho_\theta^{t_k}$ in the tangent space of $\rho_\theta^{t_{k-1}}$, representing its change rate. The pair $(\rho_\theta^{t_{k-1}}, \partial_t \rho_\theta^{t_k})$ thus corresponds to a point on the tangent bundle $\mathcal{T}\mathcal{P}^+(\mathcal{M})$, with parameters $\theta^k \in \Theta$ representing one of the possible coordinate charts for

this update. As such, the parameter space $\Theta$ forms a finite–dimensional subspace of $\mathcal{TP}^+(\mathcal{M})$ (see Fig. 5). The block transformation defines a smooth immersion $T_{\theta^k} : \Theta \to \mathcal{TP}^+(\mathcal{M})$ with full-rank Jacobian $\nabla_x T_{\theta^k}$, ensuring the pullback of the Wasserstein metric $g^{\mathcal{W}_2}$ to $\Theta$, denoted $T_\theta^* g^{\mathcal{W}_2}$, is well-defined and induces a Riemannian structure. This Riemannian metric identifies $\Theta$ with its dual $\Theta^*$ via the standard tangent–cotangent isomorphism (do Carmo, 1992). Consequently, the contact Hamiltonian dynamics on $\mathcal{T}^*\mathcal{P}^+(\mathcal{M}) \times \mathbb{R}$ can be equivalently expressed in the reduced phase space $\Theta^* \times \mathbb{R}$ (Wu et al., 2025). At the same time, the Wasserstein manifold is approximated by the finite-dimensional submanifold $\mathcal{P}_\theta^+(\mathcal{M})$, whose tangent space is $\mathcal{TP}_\theta^+(\mathcal{M}) = \Theta$.

### E.2 Training Algorithm

---

**Algorithm 1** Training the Contact Wasserstein Geodesic (CWG) Framework

---

**Input:** Dataset: samples from marginals $x_{a,\{i,j\}} \sim \rho_a$, $x_{b,\{i,j\}} \sim \rho_b$, $x_{m,\{i,j\}} \sim \{\rho_m\}_{m=1}^N$.
**Output:** A trained ResNet $T_{\{\theta^k\}_{k=0}^K}$.

    **Part I: Initialization to Match the Initial Marginal**
1: **for** $i = 1$ to $E$ **do**                                                ▷ Epoch loop
2:     **for** $j = 1$ to $B$ **do**                                        ▷ Batch loop
3:         $x^s \sim \lambda$                         ▷ Sample from reference distribution
4:         $x^{t_0} = T_{\theta^0}(x^s)$                        ▷ Apply initial block
5:         $\min_{\theta^0} \ d_{\mathcal{W}_2}^2(x^{t_0}, x_{a,\{i,j\}})$        ▷ Match initial marginal
6:     **end for**
7: **end for**

    **Part II: Geodesic Optimization**
8: **for** $i = 1$ to $E$ **do**
9:     **for** $j = 1$ to $B$ **do**
10:         $x^s \sim \lambda$                     ▷ Sample from reference distribution
11:         $\{x^{t_0}, x^{t_1}, \ldots, x^{t_K}\} = T_{\{\theta^k\}_{k=0}^K}(x^s)$       ▷ Full ResNet transformation
12:         $\min_{\theta \backslash \theta^0} \ \ell(\{x^{t_k}\}_{k=0}^K, x_{b,\{i,j\}}, x_{m,\{i,j\}})$    ▷ Minimize geodesic loss (equation (9))
13:     **end for**
14: **end for**

---

### E.3 Stability Considerations

The proposed solver discretizes the geodesic flow on the Wasserstein manifold and can be interpreted as a single-shooting direct method for solving the optimal control problem of determining the optimal set of parameters $\{\theta^{k*}\}_{k=0}^K$. Specifically, the ResNet-based parametrization $T_{\{\theta^k\}_{k=0}^K}$ produces a discrete sequence of intermediate batches $\{x_\theta^{t_k}\}_{k=0}^K$, converting the continuous optimal control problem into a nonlinear programming problem. This problem is then solved using standard static optimization methods, with gradients computed through backpropagation. The loss function in equation 9,

$$\ell = \underbrace{d_{\mathcal{W}_2}^2(\rho_\theta^{t_K}, \rho_b)}_{\text{Terminal marginal}} + \underbrace{\sum_{m=1}^M d_{\mathcal{W}_2}^2(\rho_\theta^{t_{k_m}}, \rho_m)}_{\text{Intermediate marginals}} + \underbrace{\sum_{k=1}^K \Phi^{t_k} d_{\mathcal{W}_2}^2(\rho_\theta^{t_k}, \rho_\theta^{t_{k-1}})}_{\text{Energy minimization}},$$

involving the computation of the Wasserstein distance, which can be explicitly rewritten as,

$$\ell = \underbrace{\sum_{i,l}^N \pi_{il}\|x_i^{t_K} - y_l^{t_K}\|^2}_{\text{Terminal marginal}} + \underbrace{\sum_{m=1}^M \sum_{i,l}^N \pi_{il}\|x_i^{t_{k_m}} - y_l^{t_m}\|^2}_{\text{Intermediate marginals}} + \underbrace{\sum_{k=0}^{K-1} \sum_{i,l}^N \Phi^k \pi_{il}\|x_i^{t_{k+1}} - x_i^{t_k}\|^2}_{\text{Energy minimization}}, \quad (47)$$

where $\pi_{il}$ denotes the optimal transport plan between samples $i$ and $l$ the data batches, optimized in order to minimize the transport cost and preserve the constraint, $\sum_{il}^N \pi_{il} = 1$, ensuring full mass conservation between batches. Importantly, the optimization of the transport plan $\pi_{il}$ is a

subproblem carried out for a fixed set of parameters $\{\theta^k\}_{k=0}^K$. This subproblem is a linear program whose solution depends Lipschitz-continuously on $\{\theta^k\}_{k=0}^K$, i.e.,

$$\left\| \pi\left(\{\theta^{k'}\}_{k=0}^K\right) - \pi\left(\{\theta^k\}_{k=0}^K\right) \right\| \leq L_\pi \left\| \{\theta^{k'}\}_{k=0}^K - \{\theta^k\}_{k=0}^K \right\|, \tag{48}$$

where $L_\pi$ is a scalar constant. The two differences quantify, respectively, the change in the transport plan induced by a perturbation in the parameter set, and the perturbation in the parameter set itself.

The computation of the gradient of the loss in equation (47), which is responsible for updating the set of parameters according to,

$$\{\theta_{(j+1)}^k\}_{k=0}^K = \{\theta_{(j)}^k\}_{k=0}^K - \alpha \nabla_\theta \ell, \tag{49}$$

leads to the appearance of three types of terms:

- $\nabla_\theta \pi_{il}$, which is Lipschitz in $\{\theta^k\}_{k=0}^K$ because the transport plan $\pi$ is the solution of a convex optimization problem whose dependence on the parameters is Lipschitz-continuous;

- $\nabla_\theta \|x_i^{t_k} - y_l^{t_k}\|^2$, where $x_i^{t_k}$ depends nonlinearly on $\{\theta^k\}_{k=0}^K$ due to the nonlinear ResNet parametrization $T_{\{\theta^k\}_{k=0}^K}$ used for expressivity. This implies that multiple different parameter sets may minimize this term. However, around a local optimal point the problem of minimizing the Euclidean distance is locally convex;

- $\nabla_\theta \Phi^{t_k}$, which is a nonlinear function of $\{x_\theta^{t_k}\}_{k=0}^K$. If $\Phi^{t_k}$ is at least piece-wise smooth, it is possible to identify a local minimum for $\{x_\theta^{t_k}\}_{k=0}^K$, and therefore a corresponding local minimum for $\{\theta^k\}_{k=0}^K$, depending on the initialization of the parameters. Even in the most complex scenario considered in this paper, the energy-varying SB case, the required smoothness assumptions are satisfied. Specifically, for $\Phi^{t_k} = H(\rho^{t_k}, S^{t_k}, z^{t_k}) + U(\rho^{t_k}) + I(\rho^{t_k}) - \mathcal{B}(\rho^{t_k})$, $H$ is a smooth function depending on a hyperparameter, $U$ is implemented either via at least piecewise smooth learning models (Apps. G.4, G.5, G.7) or via a LAND metric (Arvanitidis et al., 2018) with a smooth kernel (Apps. G.2, G.3), and both $I$ and $\mathcal{B}$ are smooth functions.

Therefore, each update $\{\theta_{(j)}^k\}_{k=0}^K \longrightarrow \{\theta_{(j+1)}^k\}_{k=0}^K$ does not destabilize the training and converges toward a local minimum, which geometrically corresponds to one of the possible coordinate charts that can describe the tangent bundle $\mathcal{TP}^+(\mathcal{M})$ (see Appendix E.1 for the geometrical interpretation). The stability of the gradient for long time horizons is moreover guaranteed by the ResNet architecture itself, where the skip connections prevent detrimental effects such as exploding or vanishing gradients (Zaeemzadeh et al., 2020).

### E.4 GEODESIC PARAMETERIZATION: TIME AND ARC-LENGTH PERSPECTIVES

The proposed ResNet architecture $T_{\{\theta^k\}_{k=0}^K}$ discretizes the geodesic $\rho^t$ on $\mathcal{P}^+(\mathcal{M})$ into a piecewise-linear path segmented at nodes $\{\rho_\theta^{t_k}\}_{k=0}^K$, thereby defining a specific sampling of points along the geodesic. By default, the discretization obtained by minimizing the curve energy,

$$d_{\mathcal{W}_2}^2(\rho_a, \rho_b) = \sum_{k=1}^K d_{\mathcal{W}_2}^2(\rho_\theta^{t_k}, \rho_\theta^{t_{k-1}}), \tag{50}$$

corresponds to a uniform arc-length discretization $s$, where each segment has approximately the same length, i.e., $d_{\mathcal{W}_2}^2(\rho_\theta^{t_k}, \rho_\theta^{t_{k-1}}) \approx C$. The relationship between the arc-length discretization $s$ and the time discretization $t$, which describes the progression along the geodesic with respect to the time variable, is given by $ds = \sqrt{\mathcal{K}_{\text{phy}}}\, dt$, where $\mathcal{K}_{\text{phy}}$ denotes the kinetic energy of the system whose dynamics is interpreted as a geodesic. In the energy-conserving case, $\mathcal{K}_{\text{phy}}$ remains constant, and the time and arc-length discretizations differ only by a constant rescaling of the geodesic length. Nevertheless, the discretization nodes $\{\rho_\theta^{t_k}\}_{k=0}^K$ remain unchanged, uniformly distributed in space and in time. In contrast, when modeling an energy-varying system whose associated trajectory

projects onto a geodesic, the curve energy takes the form,

$$d^2_{\mathcal{W}_2}(\rho_a, \rho_b) = \sum_{k=1}^{K} \Phi^{t_k} d^2_{\mathcal{W}_2}(\rho_\theta^{t_k}, \rho_\theta^{t_{k-1}}), \tag{51}$$

where the scaling factor $\Phi^{t_k} = H(\rho^{t_k}, S^{t_k}, z^{t_k}) - \mathcal{F}(\rho^{t_k}) - \mathcal{B}(\rho^{t_k})$ accounts for the use of the Jacobi metric. This scaling factor also determines the length of each segment, which is inversely proportional to it, i.e., $d^2_{\mathcal{W}_2}(\rho_\theta^{t_k}, \rho_\theta^{t_{k-1}}) \approx \frac{C}{\Phi^{t_k}}$. Consequently, the arc-length discretization $d_{\mathcal{W}_2}(\rho_\theta^{t_k}, \rho_\theta^{t_{k-1}}) = \Delta s$ is non-uniform, while the time discretization $\Delta t = \sqrt{\frac{C}{\kappa}}$ remains uniform, as the nodes are assumed to be sampled at constant time intervals. The origin of this discrepancy lies in the kinetic energy term relating the arc-length and time increments, $\Delta s = \sqrt{\mathcal{K}_{\text{phy}}}\Delta$, which is no longer constant in the energy-varying case.

This reasoning provides insight into the relationship between the energy term $\Phi^{t_k}$ used in the geodesic minimization and the physical kinetic energy of the modeled system. Specifically, since the geodesic minimization problem yields,

$$d^2_{\mathcal{W}_2}(\rho_\theta^{t_k}, \rho_\theta^{t_{k-1}}) = \frac{C}{\Phi^{t_k}}, \tag{52}$$

while, as established above,

$$d^2_{\mathcal{W}_2}(\rho_\theta^{t_k}, \rho_\theta^{t_{k-1}}) = \Delta s^2 = \mathcal{K}_{\text{phy}} \Delta t^2 = \mathcal{K}_{\text{phy}} \frac{C}{\kappa}, \tag{53}$$

we obtain $\Phi^{t_k} = \frac{\kappa}{\mathcal{K}_{\text{phy}}}$. This result shows that the energy employed in the geodesic computation is inversely proportional to the physical kinetic energy, scaled by the constant $\kappa$. This relationship arises from the coupling between the arc-length and time discretization of the system's trajectory.

### E.5 TIME COMPLEXITY AND PRACTICAL CONSIDERATIONS

At each iteration of the geodesic optimization of Alg. 1, we sample $N$ points $x^s \in \mathcal{M}$ of dimension $D$ from the reference distribution $\lambda$ and pass them through the ResNet. Assuming its architecture consists of $K$ blocks, each being an MLP of $L$ layers and hidden dimension $W$, the computational cost of this forward pass is $\mathcal{O}(N D K L W)$. The loss function $\ell$ in equation (9) requires $M + K$ evaluations of the Wasserstein distance between sample batches, and $K$ evaluations of the factor $\Phi^{t_k}$. For the Wasserstein distance, we employ the `geomloss` library (Feydy, 2020), which uses the Sinkhorn algorithm with time complexity $\mathcal{O}(N(D + T_{\text{sh}}))$, where $T_{\text{sh}}$ is the number of Sinkhorn iterations until convergence (Feydy, 2020). To evaluate $\Phi^{t_k}$, we apply its complete definition,

$$\Phi^{t_k} = H^{t_k} + \int_{\mathcal{M}} \left[ U(x^{t_k}) - 2\varepsilon\left(\log\left(2\rho_\theta^{t_k}\right) - 1\right) \right] \rho_\theta^{t_k}\, dx + \tfrac{1}{2}\varepsilon^2 I(\rho_\theta^{t_k}), \tag{54}$$

to a batch of samples. As discussed in Sec. 3, the Fisher information $I(\rho_\theta^{t_k})$ in the potential energy originates from the entropy regularization in the SB formulation and does not require explicit computation. Its effect is implicitly captured by the entropy-regularized Wasserstein distance. Therefore, evaluating $\Phi^{t_k}$ reduces to computing the scalar functions $H$ and $U$, along with estimating the entropy term $\log(2\rho_\theta^{t_k})\rho_\theta^{t_k}$, which is the computational bottleneck. This term can be estimated using a $k$-NN entropy estimator with time complexity $\mathcal{O}(D N \log N)$ (Borelli et al., 2022). Considering all components, and given that $K \geq M$, the overall time complexity becomes,

$$\mathcal{O}(N d K L W) + \mathcal{O}\left(N(K+M)(D+T_{\text{sh}})\right) + \mathcal{O}(K d N \log N) \approx \mathcal{O}\left(N K\left(T_{\text{sh}} + D(L W + \log N)\right)\right). \tag{55}$$

Our method demonstrates highly favorable scaling properties, offering a significant advantage over existing approaches. Notably, its computational complexity scales linearly with data dimensionality $D$ and nearly linearly with the batch size $N$. Furthermore, our model's performance is only weakly influenced by the number of marginals and it circumvents the expensive outer iteration loops. This stands in stark contrast to existing methods, as elaborated next:

- Hong et al. (2025) scales quadratically in $N$ and exponentially in $D$, with time complexity,

$$\mathcal{O}\big(T_{\text{iter}} DKN^2 \exp\big(\alpha(D+1)\big)\big), \tag{56}$$

where $T_{\text{iter}}$ denotes the number of Belief Propagation iterations required for convergence, and $\alpha$ is a parameter determined by the order of the Gaussian process and structural assumptions on their representation basis.

- Chen et al. (2023) proposed a method with time complexity,

$$\mathcal{O}\big(T_{\text{iter}}KN(2D)^2\big), \tag{57}$$

where $T_{\text{iter}}$ denotes the number of Bregman iterations needed for convergence. The quadratic dependence on $2D$ arises from the state space being doubled by incorporating velocity, which comes at the cost of computational efficiency.

- Shen et al. (2025) employed a score-matching SB solver (Vargas et al., 2021) for the mmSB problem with computational complexity,

$$\mathcal{O}(T_{\text{iter}} MN(2D)), \tag{58}$$

where $T_{\text{iter}}$ again denotes the number of Bregman iterations needed for convergence, with reported values between 2000 and 4000. Although the method is nearly linear in both $N$ and $D$, it depends explicitly on the number of intermediate marginals $M$ and suffers from additional outer-loop iterations.

As highlighted in this analysis, our CWG method advances the state of the art as a remarkably efficient computational approach: it scales linearly in all relevant terms and does not require iterations. The baselines (Shi et al., 2023; Bortoli et al., 2024; Liu et al., 2024) do not explicitly report their computational efficiency, so the comparison with our method is addressed empirically in the experiments.

## F  IMPLEMENTATION DETAILS

### F.1  EMPIRICAL APPROXIMATION OF THE WASSERSTEIN-2 DISTANCE

Let $\rho_c$ and $\rho_d$ be two probability distributions from which we draw batches of samples $\{x_{c,i}\}_{i=1}^N \sim \rho_c$ and $\{x_{d,j}\}_{j=1}^M \sim \rho_d$, respectively. The Wasserstein-2 distance, denoted by $d_{\mathcal{W}_2}(\rho_c, \rho_d)$, measures the minimal cost of transporting mass between these two distributions. To approximate this distance empirically, we first construct a cost matrix $C \in \mathbb{R}^{N \times M}$, where each entry,

$$C_{ij} = \|x_{c,i} - x_{d,j}\|^2, \tag{59}$$

represents the squared Euclidean distance between sample $x_{c,i}$ and sample $x_{d,j}$. A transport plan is then defined as a matrix $\pi \in \mathbb{R}_+^{N \times M}$ that assigns how much mass to move from each $x_{c,i}$ to each $x_{d,j}$, minimizing the total transport cost weighted by $C$. To avoid degenerate solutions where all mass is concentrated on a few points, entropy regularization is introduced, encouraging smoother and more distributed transport plans. For this computation, we employ the `SamplesLoss` function from the `geomloss` library (Feydy, 2020), with parameters resumed in Table 9.

In the conditional generation setting, the probability flow $\rho^t(x \mid y)$ is conditioned on the feature $y = f(x^{t_s})$, with $x^{t_s} \sim \rho_s$, to ensure that the generated samples satisfy $y$ at time step $t_s$. Therefore, when comparing the prescribed distribution $\rho_s$ with a marginal $\rho_m$, a modified Wasserstein-2 distance $d'_{\mathcal{W}_2}(\rho_s, \rho_m)$ incorporating the feature penalty is used. This distance is defined via the cost matrix

$$C_{ij} = \|x_i^{t_s} - x_{m,j}\|^2 + \|y - f(x_{m,j})\|^2, \tag{60}$$

which penalizes transport plans assigning mass to samples $x_{m,j}$ inconsistent with the conditioning feature $y$.

### F.2  TEASER IMAGE

The system illustrated in Figure 1 depicts the time parameterization of three different Schrödinger bridges connecting two 1D Gaussian distributions with identical variance but different means. The

energy-conserving bridge (—) corresponds to the solution of the mmGSB problem (2) in the specific case of $U = 0$ and without intermediate marginals $\rho_m$. The computed bridge can be interpreted as a piecewise geodesic on the Wasserstein manifold $\Sigma$ endowed with the pullback Wasserstein metric $T^* g^{\mathcal{W}}$.

In contrast, the two energy-varying curves (—, —) instead correspond to the solutions of the NCGSB problem (4) in the case of no intermediate marginals and $U = 0$. These trajectories are piecewise geodesics with respect to the pullback Jacobi metric $T_\theta^* \tilde{g}_{\mathrm{J}} = \left( H(\rho_\theta^{t_k}) - \mathcal{F}(\rho_\theta^{t_k}) - \mathcal{B}(\rho_\theta^{t_k}) \right) T^* g^{\mathcal{W}_2}$. Here, the potential energy term $\mathcal{F} = -U - I$, reduces to the Fisher information contribution, yielding the equivalent expression $T_\theta^* \tilde{g}_{\mathrm{J}} = \left( H(\rho_\theta^{t_k}) + I(\rho_\theta^{t_k}) - \mathcal{B}(\rho_\theta^{t_k}) \right) g^{\mathcal{W}_2}$. Minimizing the geodesic energy with respect to this metric yields trajectories that tend to reduce the scaling factor $\left( H(\rho_\theta^{t_k}) + I(\rho_\theta^{t_k}) - \mathcal{B}(\rho_\theta^{t_k}) \right)$. As a result, these paths preferentially traverse low-energy regions of $\Sigma$ while concentrating the discretization nodes $\{\rho_\theta^{t_k}\}_{k=0}^K$ in the higher-energy portions of the curve, since long segments between nodes in such regions are penalized by the large value of the scaling factor. Specifically, the geodesic computation exhibits the following behaviors:

- The minimization of the Fisher information $I(\rho_\theta^{t_k})$, promoting smoother and more regular trajectories by penalizing rapid variations in the density. This ensures stable and coherent dynamics along the path.
- The maximization of the entropy term $\mathcal{B}(\rho_\theta^{t_k})$, upper-bounded by the Hamiltonian $H(\rho_\theta^{t_k})$, since the metric $T_\theta^* \tilde{g}_{\mathrm{J}}$ must remain positive. Consequently, higher Hamiltonian values correspond to higher-entropy paths.
- The concentration of the discretization nodes in regions of higher energy (i.e., where $H(\rho_\theta^{t_k})$ is larger). As the evolution of the Hamiltonian can be prescribed, selecting a decreasing or increasing law results in a stochastic trajectory that slows down or speeds up over time, respectively, as discussed in Appendix E.4.

These behaviors can be observed in the two energy-varying probability paths illustrated in Figure 1. In the red curve (—), the Hamiltonian function, ranging from $1.5$ to $0.5$, is defined as,

$$H^{t_k} = 0.5 + \frac{\mu(x^{t_K}) - \mu(x^{t_k})}{\mu(x^{t_K}) - \mu(x^{t_0})}, \tag{61}$$

where $\mu(x^{t_k})$ denotes the mean of the samples $x^{t_k} \sim \rho_\theta^{t_k}$. The Hamiltonian is thus decreasing, producing a dynamics that evolves slowly at the beginning and accelerates toward the end of the time interval. The opposite behavior is observed in the purple curve (—), where the Hamiltonian increases from $0.01$ to $1.01$,

$$H^{t_k} = 1.01 - \frac{\mu(x^{t_K}) - \mu(x^{t_k})}{\mu(x^{t_K}) - \mu(x^{t_0})}. \tag{62}$$

In this case, the dynamics gradually slows down over time. Moreover, the lower overall energy level results in reduced entropy, as evidenced by the narrower Gaussian distribution observed at the midpoint of the bridge in the purple curve compared to the red one.

# G    EXTENDED RESULTS

## G.1    EXPERIMENTAL SETUP

The LiDAR Manifold Navigation and Cell Sequencing experiments were conducted on a machine equipped with 13th Gen Intel® Core™ i7-13850HX CPUs. The Image Generation experiment was run on a system with an NVIDIA GeForce RTX5090 GPU (32GB VRAM, CUDA12.9, driver version 575.64.03). Results for the LiDAR Manifold Navigation (Table 2), Single Cell Sequencing (Table 3), Sea Temperature Prediction (Table 4 and Appendix G.4), Robot Task Reconstruction (Table 8), FFHQ Transfer (Table 7), and MNIST-to-EMNIST (Table 38) experiments are based on a total of ten evaluations obtained from training runs with different initial conditions. The tables report the mean and standard deviation of these distributions.

The ResNet architecture varies depending on the task. For the LiDAR Manifold Navigation, the Cell Sequencing and the Unpaired Transfer experiments, each block is an MLP that processes the

output of the previous block and generates an update, which is added to the input with a step size $\tau$: $x \leftarrow x + \tau$ block$(x)$. Details of this architecture are provided in Table 11. In contrast, for the Image Generation experiment, the input consists of images, and each block is implemented as a 2D U-Net. Details of this architecture are provided in Table 12. The total number of CWG parameters used across the different experiments, alongside the baselines, is summarized in Table 13.

| Parameter | Value |
|---|---|
| Entropy | 0.05 |
| Euclidean norm order | 2 |
| Scaling | 0.7 |

Table 9: `SampleLoss` parameters.

| Parameter | Lidar | Cell | Sea | Robot | FFHQ | Mnist |
|---|---|---|---|---|---|---|
| # samples $N$ | 1000 | 1000 | 200 | 100 | 1000 | 300 |
| Weight $w_b$ | 10 | 10 | 100 | 10 | 1 | 1 |
| Weight $w_m$ | – | 10 | 100 | – | – | – |
| Weight $w_g$ | 1 | 1 | 1 | 1 | 1 | 1 |

Table 10: Loss function (9) parameters for the experiments.

| Component | ResNet | | |
|---|---|---|---|
| | Lidar | Cell | FFHQ |
| Number of blocks $K$ | 20 | 5 | 6 |
| Layers per block | 3 | 3 | 3 |
| Layer hidden size | 30 | 128 | 1024 |
| Step size $\tau$ | 0.1 | 0.1 | 0.1 |
| Input dimension $d$ | 3 | 5 | 512 |

Table 11: Configuration of the ResNet

| Component | Image ResNet | | |
|---|---|---|---|
| | Sea | Robot | Mnist |
| Number of blocks $K$ | 5 | 8 | 6 |
| Input channels | 1 | 3 | 1 |
| Output channels | 1 | 3 | 1 |
| Layers per block | 1 | 2 | 2 |
| Downsampling blocks | 2 (32 and 64 channels) | | |
| Upsampling blocks | 2 (32 and 64 channels) | | |
| Step size $\tau$ | 1 | 1 | 1 |
| Input dimension $d$ | 4096 | 12288 | 784 |

Table 12: Configuration of the Image ResNet.

During training, the weights $\{w_b, w_m, w_g\}$ balancing the loss terms in equation (9),

$$\ell = w_b \, d^2_{\mathcal{W}_2}(\rho_\theta^{t_K}, \rho_b) + w_m \sum_{m=1}^{M} d^2_{\mathcal{W}_2}(\rho_\theta^{t_{k_m}}, \rho_m) + w_g \sum_{k=1}^{K} \Phi^{t_k} \, d^2_{\mathcal{W}_2}(\rho_\theta^{t_k}, \rho_\theta^{t_{k-1}}),$$

along with the number of samples $N$ used for Wasserstein distance estimation, are experiment-specific and summarized in Table 10. The GPU memory consumption for the two image generation experiments, comparing our CWG method with the baseline approaches, is reported in Table 14. Due to the explicit handling of the full probability distribution (albeit in discretized form) within the ResNet architecture, CWG exhibits particularly high memory requirements. In contrast, methods such as GSBM and DSBM model only the drift component and subsequently integrate the dynamics. While this makes them significantly more demanding in terms of computation time, they are more memory-efficient than CWG.

| Experiment | DSBM | SB-Flow | GSBM | SBIRR | DM-SB | CWG |
|---|---|---|---|---|---|---|
| LiDAR Navigation | 0.14 M | – | 0.33 M | – | – | 0.017 M |
| Cell Sequencing | – | – | – | 0.013 M | 2.5 M | 0.017 M |
| Sea Temperature | 79.2 M | 11 M | 271 M | – | – | 5.2 M |
| Robot Task | 79.2 M | 11 M | 271 M | – | – | 7.3 M |
| Unpaired Transfer | 5.7 M | 6.6 M | 5.7 M | – | – | 15.7 M |
| Mnist-to-Emnist | 4.6 M | 3.3 M | 4.6 M | – | – | 5.2 M |

Table 13: Number of parameters (in millions, M) of the models reported in Table 1

| Methodology | CWG | GSBM | DSBM |
|---|---|---|---|
| Sea Temperature (MB) | $25200 \pm 200$ | $9600 \pm 300$ | $9000 \pm 200$ |
| Robot Task (MB) | $16100 \pm 200$ | $12500 \pm 300$ | $10200 \pm 200$ |

Table 14: Comparison of GPU memory consumption across the methods evaluated in the Image Generation experiment. CWG shows a decrease in memory usage for the Robotic Task Reconstruction experiments, whereas the other methods exhibit an increase due to the reduced batch size used in this training (Table 10).

## G.2  LiDAR Manifold Navigation

The LiDAR dataset (OpenTopography, 2025) consists of point clouds contained in the domain $[-5, 5]^3 \subset \mathbb{R}^3$. The objective of the experiment is to construct a bridge across the data manifold for connecting two distributions while avoiding regions of high elevation and remaining closely aligned with the manifold structure. The initial distribution $\rho_a$ is composed by a mixture of 4 Gaussian distributions, the target distribution $\rho_b$ is composed of 2 Gaussians on the two sides of the mountain. The manifold shape is incorporated in the problem through the potential function $U$, inherited from the baseline (Liu et al., 2024),

$$\int_{\mathcal{M}} U(x)\rho^t(x)\, dx = \int_{\mathcal{M}} \left(U_{\text{manifold}}(x) + U_{\text{height}}(x)\right)\rho^t(x)\, dx, \tag{63}$$
$$U_{\text{manifold}}(x) = w_{\text{manifold}}\|\psi(x) - x\|^2, \quad U_{\text{height}}(x) = w_{\text{height}}\|\psi^{(z)}(x)\|^2.$$

Here, $\psi(x)$ denotes the projection of a point $x$ onto an approximate tangent plane, estimated from its $p$ nearest neighbors on the data manifold. The notation $\psi^{(z)}(x)$ refers to the $z$-coordinate of $\psi(x)$, i.e., the height of the fitted plane. The weights $w_{\text{manifold}}$ and $w_{\text{height}}$ control the relative importance of the two terms in the potential function. We now detail the construction of $\psi(x)$. Let $N_p(x) = \{x_1^l, \ldots, x_p^l\}$ denote the set of $p$ nearest neighbors of $x \in \mathbb{R}^3$ in the dataset. To approximate the local tangent plane, we employ a moving least-squares (MLS) procedure (Levin, 1998). Specifically, the plane parameters $(a, b, c)$ are obtained by solving,

$$\min_{a,b,c} \frac{1}{p} \sum_{i=1}^{p} w(x, x_i^l) \left(ax_i^{l(x)} + bx_i^{l(y)} + c - x_i^{l(z)}\right)^2, \tag{64}$$

where the superscripts indicate coordinates and the weights are defined as,

$$w(x, x_i^l) = \exp\left(-\frac{\|x - x_i^l\|}{\gamma}\right), \tag{65}$$

with $\gamma$ being a scaling parameter. Given the fitted plane, the projection operator $\psi(x)$ is defined as,

$$\psi(x) = x - \frac{x^\top n + c}{\|n\|^2}\, n, \quad n = [a \ \ b \ \ -1]^\top, \tag{66}$$

where $n$ denotes the plane's normal vector. Differentiation through $\psi$ naturally restricts gradients to this tangent plane, thereby ensuring that optimization of the state cost $U$ evolves within the geometry of the data manifold. The values of the parameters used in the computation of the projection operator $\psi(x)$ are provided in Table 15.

The quantitative results reported in Table 2 aim to characterize the two key aspects of the Schrödinger Bridge problem, as formulated in the stochastic optimization problems (2) and (4). The Optimality metric reports the length of the stochastic bridge, computed as the square root of the cost functional $J$. It quantifies the ability of the approach to find low transport-cost solutions connecting the marginals $\rho_a$ and $\rho_b$. In contrast, the Feasibility metric is computed as the Wasserstein distance between the marginals and the nodes of the discretized bridge approximating them. This metric measures how well the boundary constraints in the optimization problems are satisfied. Formally, these metrics are defined as,

$$\text{Optimality} = \sqrt{J}; \tag{67a}$$
$$\text{Feasibility} = d_{\mathcal{W}_2}(\rho_\theta^{t_0}, \rho_a) + d_{\mathcal{W}_2}(\rho_\theta^{t_K}, \rho_b). \tag{67b}$$

In the guided generation setting, the feature function $f$, defines as follows,

$$f(x_b) = \text{ReLU}(w_x x_b^{(x)} - b_x) + \text{ReLU}(w_y x_b^{(y)} - b_y), \quad x_b \sim \rho_b \tag{68}$$

is used to penalize samples from the terminal marginal distribution $\rho_b$ on the left side of the mountain. The parameters used in this experiment are listed in Table 16, and the results of guidance fine-tuning are reported in Table 17. The training time (tt) metric shows that guiding the generation requires only 10.7% of the original training time.

Table 15: Potential function $U$ parameters for the LiDAR Manifold Navigation experiment.

| Parameter | Value |
|---|---|
| Weight manifold $w_{\text{manifold}}$ | 5 |
| Weight height $w_{\text{height}}$ | 1 |
| Spatial scaling $\gamma$ | 0.1 |
| # neighbor points $p$ | 20 |

Table 16: Parameters for the feature function $f$ (68), used for guided generation in the LiDAR Manifold Navigation experiment.

| Parameter | Value |
|---|---|
| Weight $(x)$ $w_x$ | $-1$ |
| Bias $(x)$ $b_x$ | 0 |
| Weight $(y)$ $w_y$ | 1 |
| Bias $(y)$ $b_y$ | 0 |

Table 17: Bridge energy $J$ ($\downarrow$) with penalty $f$ in the LiDAR Manifold Navigation task, reported for our CWG method before and after guidance fine-tuning.

| Metric | before | after |
|---|---|---|
| $J$ | $12.31_{\pm 0.18}$ | $\mathbf{2.49_{\pm 0.43}}$ |
| tt (s) | $\mathbf{280_{\pm 20}}$ | $310_{\pm 25}$ |

### G.3 SINGLE CELL SEQUENCING

The Embryoid Body (EB) stem cell differentiation dataset (Moon et al., 2019) captures cell state progression across five developmental stages $[t_0, t_1, t_2, t_3, t_4]$ over a 27-day period. Snapshots were collected at five discrete time intervals: $t_0 \in [0, 3]$, $t_1 \in [6, 9]$, $t_2 \in [12, 15]$, $t_3 \in [18, 21]$, and $t_4 \in [24, 27]$. These stages involve significant structural changes, with cells moving and reorganizing within increasingly stiff tissue while consuming and releasing mechanical energy (Zeevaert et al., 2020; Kinney et al., 2014). Consequently, the resulting dynamics exhibit energy dissipation and are better described by the NCGSB framework than by energy-conserving models. In this experiment, we evaluate the framework's ability to generalize to regions with no available data by dividing the dataset into a training set $[t_0, t_2, t_4]$ and a validation set $[t_1, t_3]$. The geometry of the data manifold is incorporated into the NCGSB problem through a potential function $U$, defined as,

$$U(x^t) = \frac{1}{N_1} \sum_{i=1}^{N_1} \left[ \frac{1}{N_2} \sum_{j=1}^{N_2} e^{\frac{1}{\gamma} \|x_j^t - x_i\|^2} (x_j^t - x_i)^2 \right]^{-1}, \tag{69}$$

where $x_j^t \sim \rho^t$ denotes a sample from the posterior distribution, and $x_i \sim \{\rho_a, \rho_m, \rho_b\}$ are samples from the marginal distributions. $N_1$ and $N_2$ indicate the number of samples taken from $\rho^t$ and $\{\rho_a, \rho_m, \rho_b\}$, respectively, while $\gamma$ represents a spatial scaling parameter. The exponential term acts as a kernel-like weight that reduces the influence in the inverse summation $\sum_{i=1}^{N_1} [\cdot]^{-1}$ of the data points far from the bridge. Globally, the potential function $U$ measures the distance of the bridge $\rho^t$ from the available data $\{\rho_a, \rho_m, \rho_b\}$ (the training set) and its minimization guides the construction of a bridge that stays close to the known manifold while generalizing effectively to regions without data (the validation set). While other approaches leveraging this dataset (Tong et al., 2020; Shen et al., 2025) first embed the data into a 100-dimensional feature space using principal component analysis (PCA) and then restrict the analysis to the first five dimensions, this procedure excessively linearizes and flattens the data manifold, making navigation trivial and eliminating the need for intermediate marginals (Shen et al., 2025). To better preserve the manifold's geometry, we instead apply the PHATE algorithm (Moon et al., 2019) to the 100-dimensional representation, producing a 5-dimensional nonlinear embedding that more faithfully captures the original structure.

In the results reported in Tables 18 and 19, corresponding to the cell dynamics reconstruction shown in Fig. 12, the scalar Hamiltonian function $H^{t_k}$ in the pullback Jacobi metric, $T_\theta^* \tilde{g}_{\text{J}} = \left( H^{t_k} - \right.$

$\mathcal{F}(\rho^t) - \mathcal{B}(\rho^t)) T_\theta^* g^{\mathcal{W}_2}$ used in the CWG approach, is linearly varied from an initial value $H^{t_0} = 0.82$ to a final value $H^{t_K} = 1$, along the trajectory defined by the first principal dimension $D_1$, i.e.,

$$H^{t_k} = \frac{\mu(x_{D_1}^{t_K}) - \mu(x_{D_1}^{t_k})}{\mu(x_{D_1}^{t_K}) - \mu(x_{D_1}^{t_0})}(H^{t_0} - H^{t_K}) + H^{t_0}, \quad x^{t_k} \sim \rho^{t_k}. \tag{70}$$

This law is treated as a hyperparameter of the methodology. All ten experiments reported are conducted under this Hamiltonian behavior. This increasing trend, associated with the reduction of physical energy (see Appendix E.4), aligns with the underlying biological process and has been shown to produce better results than the energy-conserving case, where $H^{t_k} = 1$ is kept constant (see Table 21 and Figure 13a). As detailed in Appendix F.2, the behavior of $H^{t_k}$ plays a crucial role in determining how quickly the nodes of the stochastic path depart from the initial reference distribution and approach the subsequent reference marginal. In Figure 13b, the dataset exhibits a decreasing Wasserstein distance $d_{\mathcal{W}_2}$ between successive cell snapshots, reflecting that cell differentiation progresses more rapidly at early stages (Zeevaert et al., 2020). Training with the Hamiltonian behavior defined in equation (70) enables the model to capture this trend more accurately.

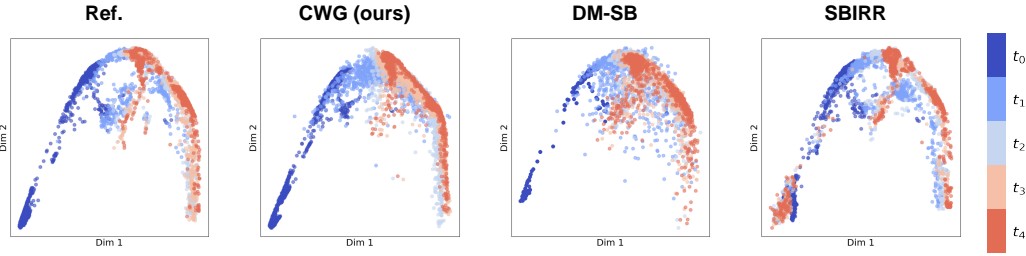

Figure 12: Reconstructions from the models in the Single Cell Sequencing experiment.

Table 18: Wasserstein error ($\downarrow$) for all the time points in Single Cell Sequencing.

| Metric | CWG (ours) | DM-SB | SBIRR |
|---|---|---|---|
| $d_{\mathcal{W}_2}(x^{t_0})$ | $\mathbf{0.10_{\pm 0.01}}$ | $0.59_{\pm 0.01}$ | $0.86_{\pm 0.02}$ |
| $d_{\mathcal{W}_2}(x^{t_1})$ | $\mathbf{1.11_{\pm 0.06}}$ | $2.25_{\pm 0.01}$ | $1.92_{\pm 0.02}$ |
| $d_{\mathcal{W}_2}(x^{t_2})$ | $\mathbf{0.16_{\pm 0.01}}$ | $1.17_{\pm 0.01}$ | $1.05_{\pm 0.02}$ |
| $d_{\mathcal{W}_2}(x^{t_3})$ | $\mathbf{0.33_{\pm 0.02}}$ | $1.64_{\pm 0.03}$ | $1.86_{\pm 0.02}$ |
| $d_{\mathcal{W}_2}(x^{t_4})$ | $\mathbf{0.11_{\pm 0.02}}$ | $1.03_{\pm 0.01}$ | $1.38_{\pm 0.02}$ |

Table 19: Maximum Mean Discrepancy ($\times 10^{-2}$) ($\downarrow$) for the time points in Single Cell Sequencing.

| Metric | CWG (ours) | DM-SB | SBIRR |
|---|---|---|---|
| $\text{MMD}(x^{t_0})$ | $0.9_{\pm 0.08}$ | $\mathbf{0.3_{\pm 0.02}}$ | $1.0_{\pm 0.11}$ |
| $\text{MMD}(x^{t_1})$ | $\mathbf{7.0_{\pm 0.21}}$ | $10.1_{\pm 0.33}$ | $9.9_{\pm 0.48}$ |
| $\text{MMD}(x^{t_2})$ | $\mathbf{1.2_{\pm 0.12}}$ | $2.9_{\pm 0.18}$ | $2.5_{\pm 0.17}$ |
| $\text{MMD}(x^{t_3})$ | $\mathbf{3.5_{\pm 0.22}}$ | $4.3_{\pm 0.20}$ | $6.7_{\pm 0.44}$ |
| $\text{MMD}(x^{t_4})$ | $\mathbf{0.2_{\pm 0.05}}$ | $1.5_{\pm 0.09}$ | $5.9_{\pm 0.36}$ |

Table 20: Single Cell Sequencing Parameters.

| Parameter | Value |
|---|---|
| Spatial scaling $\gamma$ | 0.3 |
| # bridge samples $N_1$ | 1000 |
| # marginal samples $N_2$ | 3000 |

Table 21: Wasserstein error at validation ($\downarrow$) and training time (tt) ($\downarrow$) in the ablation study comparing the energy-varying (e-v) and energy-conserving (e-c) versions of CWG on the Single Cell Sequencing task.

| Metric | CWG (e-v) | CWG (e-c) |
|---|---|---|
| $d_{\mathcal{W}_2}(x^{t_1})$ | $\mathbf{1.11_{\pm 0.06}}$ | $1.31_{\pm 0.06}$ |
| $d_{\mathcal{W}_2}(x^{t_3})$ | $\mathbf{0.33_{\pm 0.02}}$ | $0.49_{\pm 0.03}$ |
| tt (s) | $710_{\pm 30}$ | $710_{\pm 30}$ |

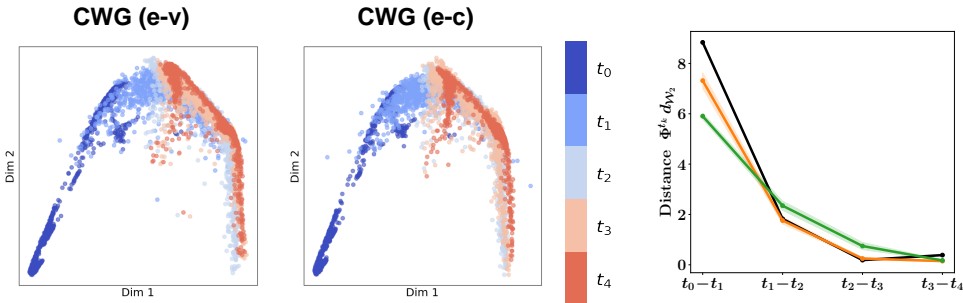

(a) Reconstruction provided by the CWG method, in the energy-varying (e-v) and energy-conserving (e-c) cases.

(b) Geodesic distances.

Figure 13: Visual reconstruction (left) and geodesic distances of cell snapshots for the reference dataset (——), CWG (e-v) (——), and CWG (e-c) (——) (right). Snapshots are equally spaced in time, but cell differentiation progresses more rapidly at the beginning, as indicated by the larger early distances in the reference. The energy profile in CWG (e-v) more accurately captures this temporal dynamics.

## G.4 SEA TEMPERATURE PREDICTION

The NOAA OISST v2 High Resolution Dataset (Huang et al., 2021) is a long-term Climate Data Record that integrates observations from multiple platforms (satellites, ships, buoys, and Argo floats) into a global gridded product. For this experiment, we use daily averages of sea surface temperature in the Gulf of Mexico between $1981$ and $2024$, represented as $64 \times 64$ single-channel images. We cluster the measurements over five-year periods and select five representative months to define five time frames: January ($t_0$), March ($t_1$), May ($t_2$), July ($t_3$), and September ($t_4$). Each month corresponds to a distribution of images, denoted as $\{\rho_a$ (January), $\rho_{m_1}$ (March), $\rho_{m_2}$ (May), $\rho_{m_3}$ (July), $\rho_b$ (September)$\}$. A sample from one of these distributions is a heatmap of the Gulf's temperature for a specific day in the corresponding month of the specified five-year period. The goal of this test is to evaluate our method's ability to interpolate across missing time frames, generating realistic temperature maps for months without data. To this end, we partition the dataset into a training set $\{t_0, t_2, t_4\}$ and a validation set $\{t_1, t_3\}$, and assess the quality of predictions on the held-out months. To encourage generalization beyond the training data, we introduce a potential function $U$ that penalizes deviations from the learned data manifold. Building on the approach of Song & Itti (2025), where generative models are evaluated by measuring the distance between their outputs and a geometric manifold of real images learned by a VAE, we adopt a similar strategy. Specifically, we use a state-of-the-art VAE architecture, with parameters listed in Table 22, to learn the manifold of the training images in our dataset. The potential function $U(x^t)$, for samples $x^t \sim \rho^t$, is then defined as the squared distance between a bridge sample $x^t$ and its VAE-projected reconstruction $\tilde{x}^t = \text{VAE}(x^t)$: $U(x^t) = \|x^t - \tilde{x}^t\|^2$. Results for each five-year periods are presented below.

In these results, the scalar Hamiltonian function $H^{t_k}$ in the pullback Jacobi metric, $T_\theta^* \tilde{g}_J = (H^{t_k} - \mathcal{F}(\rho^t) - \mathcal{B}(\rho^t)) T_\theta^* g^{\mathcal{W}_2}$ is linearly varied from an initial value $H^{t_0} = 1.36$ to a final value $H^{t_K} = 1.0$. This law is treated as a hyperparameter of the methodology, and all ten experiments reported in the following tables were conducted under this Hamiltonian behavior. In the geodesic computation, the solution path seeks to minimize $T_\theta^* \tilde{g}_J$. The functional $\mathcal{F}(\rho^t)$ measures the distance with respect to the data manifold, i.e., it evaluates whether the samples from the bridge remain coherent with the images observed during training, and it is minimized along the trajectory. Since $H^{t_k}$ is prescribed, the entropy term $\mathcal{B}(\rho^t)$ is effectively maximized up to the limit set by $H^{t_k}$, as the metric cannot become negative. Hence, the assigned Hamiltonian energy $H^{t_k}$ determines the admissible entropy level of the bridge. In this experiment, a decreasing trend of $H^{t_K}$, corresponding to an increase in physical energy (see Appendix E.4), was observed to be beneficial for modeling the warmer months, which appear to be more distant from the rest of the dataset in terms of Wasserstein distances compared to the colder months. A higher final energy $H^{t_K}$, and thus a higher final entropy $\mathcal{B}$, was observed to be beneficial for modeling the warmer months. This effect can be associated with the increased thermodynamic entropy of such cases, leading to more diverse samples and larger

variability across the data manifold. By prescribing a higher energy level, the model is encouraged to capture this diversification, exploring regions of the data manifold that in other contexts (Arvanitidis et al., 2018) are regarded as uncertain and are typically avoided.

Table 22: Architecture of the ConvVAE used in the Sea Temperature Prediction experiment. All Conv2D and ConvTranspose2D layers use kernel size 4, stride 2, and padding 1, followed by ReLU activations (except the last layer, which uses Sigmoid).

| Stage | Layer (channels) | Output size |
|---|---|---|
| Input | Single-channel image | $1 \times 64 \times 64$ |
| Encoder | Conv2D ($1 \rightarrow 32$) | $32 \times 32 \times 32$ |
| | Conv2D ($32 \rightarrow 64$) | $64 \times 16 \times 16$ |
| | Conv2D ($64 \rightarrow 128$) | $128 \times 8 \times 8$ |
| | Flatten | 8192 |
| Latent space | Linear $\rightarrow \mu$ | 5 |
| | Linear $\rightarrow \log \sigma^2$ | 5 |
| Decoder | Linear $\rightarrow$ reshape | $128 \times 8 \times 8$ |
| | ConvT2D ($128 \rightarrow 64$) | $64 \times 16 \times 16$ |
| | ConvT2D ($64 \rightarrow 32$) | $32 \times 32 \times 32$ |
| | ConvT2D ($32 \rightarrow 1$), Sigmoid | $1 \times 64 \times 64$ |

Table 23: FID scores at training and validation steps ($\downarrow$), and training time (tt) ($\downarrow$) in Sea (2020–2024).

| Metric | CWG | GSBM | DSBM | SB-Flow |
|---|---|---|---|---|
| FID($x^{t_0}$) | $41.56_{\pm 1.89}$ | — | — | — |
| FID($x^{t_1}$) | $\mathbf{121.47_{\pm 5.61}}$ | $160.68_{\pm 4.54}$ | $242.26_{\pm 9.94}$ | $176.76_{\pm 4.41}$ |
| FID($x^{t_2}$) | $51.51_{\pm 5.52}$ | $54.47_{\pm 5.88}$ | $56.51_{\pm 5.78}$ | $49.08_{\pm 4.27}$ |
| FID($x^{t_3}$) | $\mathbf{159.53_{\pm 7.38}}$ | $185.54_{\pm 7.11}$ | $235.83_{\pm 10.44}$ | $189.53_{\pm 7.42}$ |
| FID($x^{t_4}$) | $61.48_{\pm 6.79}$ | $59.14_{\pm 5.52}$ | $58.39_{\pm 6.13}$ | $59.96_{\pm 5.85}$ |
| tt (s) | $\mathbf{1030_{\pm 50}}$ | $73600_{\pm 3200}$ | $19100_{\pm 900}$ | $4950_{\pm 320}$ |

Table 24: FID scores at training and validation steps ($\downarrow$) in Sea Temperature (2015–2019).

| Metric | CWG | GSBM | DSBM | SB-Flow |
|---|---|---|---|---|
| FID($x^{t_0}$) | $42.96_{\pm 2.07}$ | — | — | — |
| FID($x^{t_1}$) | $\mathbf{130.33_{\pm 5.94}}$ | $145.97_{\pm 6.12}$ | $220.76_{\pm 8.61}$ | $174.84_{\pm 5.47}$ |
| FID($x^{t_2}$) | $58.72_{\pm 5.46}$ | $62.13_{\pm 5.69}$ | $65.47_{\pm 6.16}$ | $60.88_{\pm 5.11}$ |
| FID($x^{t_3}$) | $135.25_{\pm 6.73}$ | $142.82_{\pm 7.27}$ | $228.14_{\pm 9.26}$ | $163.53_{\pm 5.92}$ |
| FID($x^{t_4}$) | $63.02_{\pm 6.14}$ | $59.73_{\pm 5.86}$ | $61.29_{\pm 6.05}$ | $59.04_{\pm 5.76}$ |

Table 25: FID scores at training and validation steps ($\downarrow$) in Sea Temperature (2010–2014).

| Metric | CWG | GSBM | DSBM | SB-Flow |
|---|---|---|---|---|
| FID($x^{t_0}$) | $45.19_{\pm 2.27}$ | — | — | — |
| FID($x^{t_1}$) | $\mathbf{132.47_{\pm 6.58}}$ | $168.93_{\pm 6.24}$ | $255.36_{\pm 10.19}$ | $171.03_{\pm 6.38}$ |
| FID($x^{t_2}$) | $60.57_{\pm 6.08}$ | $59.79_{\pm 5.98}$ | $61.89_{\pm 6.23}$ | $58.26_{\pm 5.72}$ |
| FID($x^{t_3}$) | $140.84_{\pm 6.01}$ | $144.60_{\pm 6.83}$ | $235.08_{\pm 9.07}$ | $179.81_{\pm 7.74}$ |
| FID($x^{t_4}$) | $54.92_{\pm 5.23}$ | $51.63_{\pm 5.55}$ | $50.27_{\pm 5.94}$ | $51.38_{\pm 5.15}$ |

Table 26: FID scores at training and validation steps (tt) (↓) Sea Temperature (2005–2009).

| Metric | CWG | GSBM | DSBM | SB-Flow |
|---|---|---|---|---|
| FID($x^{t_0}$) | $45.19_{\pm 2.27}$ | – | – | – |
| FID($x^{t_1}$) | $\mathbf{172.69_{\pm 7.62}}$ | $195.83_{\pm 8.04}$ | $260.97_{\pm 10.92}$ | $194.62_{\pm 6.93}$ |
| FID($x^{t_2}$) | $56.08_{\pm 5.52}$ | $59.57_{\pm 5.87}$ | $62.03_{\pm 6.29}$ | $60.87_{\pm 5.44}$ |
| FID($x^{t_3}$) | $\mathbf{126.87_{\pm 6.94}}$ | $152.26_{\pm 6.63}$ | $243.58_{\pm 10.39}$ | $176.32_{\pm 6.21}$ |
| FID($x^{t_4}$) | $66.43_{\pm 6.58}$ | $63.17_{\pm 6.34}$ | $64.86_{\pm 6.63}$ | $64.37_{\pm 5.59}$ |

Table 27: FID scores at training and validation steps (↓) in Sea Temperature (2000–2004).

| Metric | CWG | GSBM | DSBM | SB-Flow |
|---|---|---|---|---|
| FID($x^{t_0}$) | $45.19_{\pm 2.27}$ | – | – | – |
| FID($x^{t_1}$) | $\mathbf{140.48_{\pm 7.93}}$ | $172.96_{\pm 8.34}$ | $250.86_{\pm 11.08}$ | $184.58_{\pm 7.49}$ |
| FID($x^{t_2}$) | $56.73_{\pm 5.36}$ | $56.97_{\pm 5.79}$ | $59.64_{\pm 6.10}$ | $59.53_{\pm 5.77}$ |
| FID($x^{t_3}$) | $\mathbf{142.18_{\pm 7.01}}$ | $179.42_{\pm 6.98}$ | $265.78_{\pm 10.67}$ | $195.34_{\pm 7.62}$ |
| FID($x^{t_4}$) | $68.29_{\pm 6.97}$ | $65.91_{\pm 6.68}$ | $64.73_{\pm 6.91}$ | $64.23_{\pm 5.42}$ |

Table 28: FID scores at training and validation steps (↓) in Sea Temperature (1995–1999).

| Metric | CWG | GSBM | DSBM | SB-Flow |
|---|---|---|---|---|
| FID($x^{t_0}$) | $45.19_{\pm 2.27}$ | – | – | – |
| FID($x^{t_1}$) | $\mathbf{138.92_{\pm 7.35}}$ | $176.14_{\pm 7.03}$ | $257.26_{\pm 10.93}$ | $174.53_{\pm 7.12}$ |
| FID($x^{t_2}$) | $60.19_{\pm 6.01}$ | $63.48_{\pm 6.12}$ | $66.07_{\pm 6.57}$ | $62.95_{\pm 6.06}$ |
| FID($x^{t_3}$) | $\mathbf{154.37_{\pm 8.14}}$ | $193.28_{\pm 8.47}$ | $266.86_{\pm 11.34}$ | $191.46_{\pm 7.93}$ |
| FID($x^{t_4}$) | $69.57_{\pm 7.02}$ | $66.83_{\pm 6.69}$ | $66.18_{\pm 6.94}$ | $65.62_{\pm 7.50}$ |

Table 29: FID scores at training and validation steps (↓) in Sea Temperature (1990–1994).

| Metric | CWG | GSBM | DSBM | SB-Flow |
|---|---|---|---|---|
| FID($x^{t_0}$) | $45.19_{\pm 2.27}$ | – | – | – |
| FID($x^{t_1}$) | $\mathbf{145.59_{\pm 7.82}}$ | $184.37_{\pm 7.46}$ | $258.97_{\pm 11.26}$ | $179.43_{\pm 7.83}$ |
| FID($x^{t_2}$) | $72.38_{\pm 6.42}$ | $78.69_{\pm 6.71}$ | $82.13_{\pm 7.04}$ | $75.75_{\pm 6.27}$ |
| FID($x^{t_3}$) | $\mathbf{160.08_{\pm 7.97}}$ | $181.67_{\pm 8.19}$ | $236.46_{\pm 11.59}$ | $187.26_{\pm 8.24}$ |
| FID($x^{t_4}$) | $73.42_{\pm 7.19}$ | $70.68_{\pm 6.87}$ | $69.83_{\pm 7.09}$ | $69.47_{\pm 6.98}$ |

Table 30: FID scores at training and validation steps (↓) in Sea Temperature (1985–1989).

| Metric | CWG | GSBM | DSBM | SB-Flow |
|---|---|---|---|---|
| FID($x^{t_0}$) | $45.19_{\pm 2.27}$ | – | – | – |
| FID($x^{t_1}$) | $\mathbf{151.27_{\pm 8.13}}$ | $188.79_{\pm 7.68}$ | $262.47_{\pm 11.64}$ | $194.87_{\pm 7.61}$ |
| FID($x^{t_2}$) | $66.89_{\pm 6.31}$ | $69.19_{\pm 6.47}$ | $71.23_{\pm 7.08}$ | $67.83_{\pm 6.20}$ |
| FID($x^{t_3}$) | $157.58_{\pm 8.73}$ | $160.37_{\pm 8.91}$ | $258.96_{\pm 11.78}$ | $179.36_{\pm 8.42}$ |
| FID($x^{t_4}$) | $66.02_{\pm 7.41}$ | $62.87_{\pm 7.16}$ | $61.59_{\pm 7.34}$ | $63.37_{\pm 7.19}$ |

Table 31: FID scores at training and validation steps (↓) in Sea Temperature (1981–1983).

| Metric | CWG | GSBM | DSBM | SB-Flow |
|---|---|---|---|---|
| FID($x^{t_0}$) | $45.19_{\pm 2.27}$ | – | – | – |
| FID($x^{t_1}$) | $\mathbf{166.08_{\pm 8.91}}$ | $188.79_{\pm 9.02}$ | $252.59_{\pm 12.03}$ | $194.22_{\pm 8.59}$ |
| FID($x^{t_2}$) | $69.29_{\pm 6.65}$ | $71.68_{\pm 6.74}$ | $73.87_{\pm 7.26}$ | $70.75_{\pm 7.36}$ |
| FID($x^{t_3}$) | $\mathbf{168.73_{\pm 8.46}}$ | $185.57_{\pm 8.08}$ | $260.47_{\pm 11.92}$ | $198.63_{\pm 8.45}$ |
| FID($x^{t_4}$) | $77.83_{\pm 7.68}$ | $74.97_{\pm 7.34}$ | $74.29_{\pm 7.57}$ | $74.87_{\pm 7.09}$ |

## G.5 ROBOTIC TASK RECONSTRUCTION

BridgeData V2 (Walke et al., 2023) is a large and diverse dataset of robotic manipulation behaviors, designed to advance research in scalable robot learning. In this experiment, our goal is to reconstruct the full video of a robot performing manipulation tasks while training the network only on images from the beginning and end of the sequence, interpreted as samples from the endpoint distributions $\rho_a$ and $\rho_b$. No intermediate marginals $\rho_m$ are used. Following the Sea Temperature Prediction experiment, we introduce a potential function $U$ that penalizes deviations from the learned data manifold. This manifold is learned from the initial and final frames of the videos, and the penalty encourages plausible intermediate frames consistent with these distributions. We employ a state-of-the-art VAE architecture, with parameters listed in Table 33, to model the image manifold. The potential function $U(x^t)$ for samples $x^t \sim \rho^t$ is defined as the squared distance between a bridge sample $x^t$ and its VAE reconstruction $\tilde{x}^t = \text{VAE}(x^t)$: $U(x^t) = \|x^t - \tilde{x}^t\|^2$. Fig. 7 presents snapshots of the reconstructions produced by our CWG method compared to the baselines. In this experiment, the Hamiltonian function $H^{t_k}$ is held constant, as varying it yields no apparent benefit.

In the guided generation setting, we define the feature function $f$ as

$$f(x_b) = \text{ReLU}(w_c c_b^{(x)} - b_c), \tag{71}$$

where $c_b^{(x)}$ denotes the $(x)$-coordinate pixel position of the centroid corresponding to the target location of the item placed by the robot, extracted from the image $x_b$ sampled from the reference marginal $\rho_b$. We impose a penalty $f$ on this image (parameters of this function are available in Table 34) so that samples corresponding to placements in undesired locations are discouraged, while those leading to desirable targets are favored.

The centroid extraction is performed by applying morphological opening and closing operations from the OpenCV library to remove noise and refine object boundaries, followed by color-based masking to isolate the object of interest.

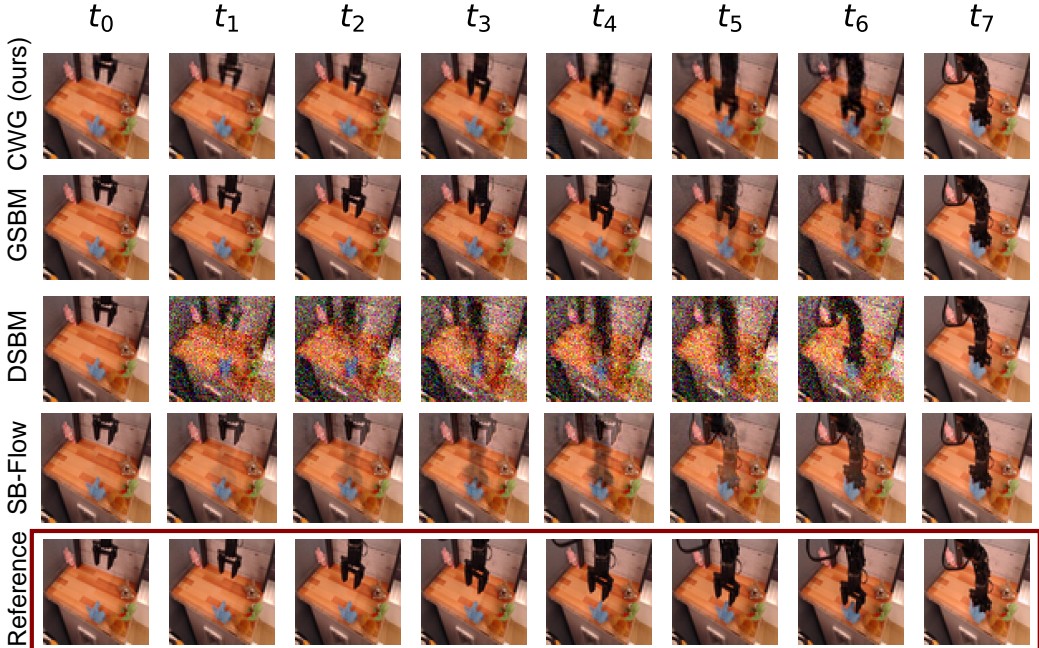

Figure 14: Reconstructions from CWG (top), GSBM (second), DSBM (third), and SB-Flow (bottom) in the Robot Task Reconstruction experiment. Red row shows the reference.

Table 32: FID score ($\downarrow$) and training time (tt) ($\downarrow$) in Robotic Task Reconstruction.

| Metric | CWG (ours) | GSBM | DSBM | SB-Flow |
|---|---|---|---|---|
| FID | $\mathbf{18.83}_{\pm\mathbf{0.66}}$ | $40.23_{\pm1.95}$ | $149.78_{\pm0.81}$ | $73.46_{\pm0.52}$ |
| tt (s) | $\mathbf{1090}_{\pm\mathbf{40}}$ | $91100_{\pm8000}$ | $27400_{\pm2500}$ | $4900_{\pm300}$ |

Table 33: Architecture of the ConvVAE used in the Robot Task Reconstruction experiment. All Conv2D and ConvTranspose2D layers use kernel size 4, stride 2, and padding 1, followed by ReLU activations (except the last layer, which uses Sigmoid).

| Stage | Layer (channels) | Output size |
|---|---|---|
| Input | Single-channel image | $3 \times 64 \times 64$ |
| Encoder | Conv2D (3→32) | $32 \times 32 \times 32$ |
| | Conv2D (32→64) | $64 \times 16 \times 16$ |
| | Conv2D (64→128) | $128 \times 8 \times 8$ |
| | Flatten | 8192 |
| Latent space | Linear $\to \mu$ | 2 |
| | Linear $\to \log \sigma^2$ | 2 |
| Decoder | Linear $\to$ reshape | $128 \times 8 \times 8$ |
| | ConvT2D (128→64) | $64 \times 16 \times 16$ |
| | ConvT2D (64→32) | $32 \times 32 \times 32$ |
| | ConvT2D (32→3), Sigmoid | $3 \times 64 \times 64$ |

Table 34: Parameters for the feature function $f$ (71), used for guided generation in the Robot Task Reconstruction experiment.

| Parameter | Value |
|---|---|
| Weight $w_c$ | 1 |
| Bias $b_c$ | 30 |

## G.6 FFHQ Transfer

The Flickr-Faces-HQ (FFHQ) dataset is a high-resolution ($1024 \times 1024$) collection of human faces that exhibits a remarkably wide range of visual variations (Karras et al., 2019). We closely follow the experimental protocol adopted by Gushchin et al. (2024), using their publicly available datasets from `https://github.com/ngushchin/LightSB`. We split the dataset into training (first $60K$ images) and testing (last $10K$ images) subsets. Each subset is further partitioned into age groups, specifically *adults* and *children*, corresponding to the source distribution $\rho_a$ and the target distribution $\rho_b$, respectively. The objective of this experiment is to generate a realistic child image from a given adult image, representing the same individual at a younger age. For each image, we employ the pre-trained ALAE encoder (Pidhorskyi et al., 2020) to extract a 512-dimensional latent vector. The SB solvers are trained directly on these latent representations. During inference, we encode a given image into the latent space, apply the learned bridge using the SB models, and then decode its outcome to obtain the transformed image. Because the probabilistic bridge is already defined within the ALAE latent space, which captures the FFHQ data manifold, the CWG method does not rely on the potential function $U$ to quantify deviations from image manifold. In contrast, the baseline GSBM approach retains a potential energy term $U$ that penalizes excessive entropy and overly concentrated probability densities $\rho^t$, distinguishing it from DSBM, which does not include any potential term.

Qualitative examples of the images generated by our CWG method and the baseline models are presented in Figures 15–17. A more compact, side-by-side qualitative comparison is provided in Figure 9, while quantitative results are reported in Table 7. The Feasibility metric indicates whether the computed bridges respect the imposed constraints, that is, whether they match the target marginal distribution. It is computed by evaluating the FID score between the terminal distribution generated by the models, $\rho_\theta^{t_K}$, and the target distribution, $\rho_b$, as defined in equation (72b). The starting distribution $\rho_a$ corresponds to the source domain from which the input samples are drawn, and therefore it is naturally satisfied and does not require matching. The Optimality metric measures the geodesic length of the Schrödinger bridges computed by the models, in order to assess the transport cost that characterizes each solution. It is computed by sampling from the probabilistic path $z^{t_k} \sim \rho^{t_k}$ in the latent space. The pre-trained ALAE decoder is then used to map these latent vectors to images $x^{t_k}$, which are subsequently encoded into feature representations $x_{\text{fea}}^{t_k}$ using a pre-trained Inception-v3

network (the same used to compute the FID score). The curve length is then obtained from equation (72a).

$$\text{Optimality} = \sum_{k=1}^{K} d_{\mathcal{W}_2}(x_{\text{fea}}^{t_k}, x_{\text{fea}}^{t_{k-1}}); \tag{72a}$$

$$\text{Feasibility} = \text{FID}(\rho_\theta^{t_K}, \rho_b). \tag{72b}$$

Hence, the length of the geodesic computed from the bridge in the latent space is evaluated according to the Wasserstein metric, pulled back from the feature space to the image space, and from the image space to the latent space. Although the baseline models achieve better Optimality scores (lower values), this improvement comes at the cost of failing to satisfy the marginal constraints, as reflected by their poorer Feasibility scores. Since constraint satisfaction is of critical importance, our CWG method exhibits superior overall behavior. This conclusion is further supported by Figure 19, where three facial age estimation networks, $h_1$ (prithivMLmods/facial-age-detection), $h_2$ (nateraw/vit-age-classifier), and $h_3$ (abhilash88/age-gender-prediction), were used to estimate the ages of the generated faces at each generation step. The different curves in the plot represent the predicted ages for the images shown in Figures 15–17. While the CWG model exhibits a more decisive transformation across time steps, the baseline curves remain comparatively flat, often producing outputs that fail to convincingly resemble children images. The age prediction models ($h_1$, $h_2$, $h_3$) provide stochastic metrics that quantify their predictive uncertainty, which we use to estimate their variance $\sigma^2$ under the assumption of a uniform error distribution. The average predicted age and the corresponding variance for the plots represented in Figure 19 are computed as,

$$\mu_{\text{age}} = \frac{\frac{h_1(x^{t_k})}{\sigma_1^2} + \frac{h_2(x^{t_k})}{\sigma_2^2} + \frac{h_3(x^{t_k})}{\sigma_3^2}}{\frac{1}{\sigma_1^2} + \frac{1}{\sigma_2^2} + \frac{1}{\sigma_3^2}}, \quad \sigma_{\text{age}}^2 = \frac{1}{\frac{1}{\sigma_1^2} + \frac{1}{\sigma_2^2} + \frac{1}{\sigma_3^2}}. \tag{73}$$

As shown in Figure 10, the energy of the bridge fitted in the latent space affects the age trends of the predictions. Decreasing the energy encourages smaller distances between the discretized nodes of the bridge at its beginning, resulting in images that are closer in age at early steps and gradually diverge toward later steps. Conversely, increasing the energy produces the opposite behavior. Importantly, in all cases, the Feasibility is preserved, as the target distribution and mean remain well below the 18-year-old threshold. This contrasts with the baseline models, which in average does not respect this constraint.

Table 35: FID scores (↓) measuring the distance to the initial adult distribution $\rho_a$ and the target child distribution $\rho_b$ for samples generated at different time steps using the CWG method in the FFHQ transfer.

| Metric | $\rho_a$ | $\rho_b$ |
|---|---|---|
| FID($x^{t_1}$) | $\mathbf{3.143_{\pm 0.447}}$ | $33.953_{\pm 0.936}$ |
| FID($x^{t_2}$) | $7.596_{\pm 0.743}$ | $27.413_{\pm 0.879}$ |
| FID($x^{t_3}$) | $25.735_{\pm 0.893}$ | $7.164_{\pm 0.806}$ |
| FID($x^{t_4}$) | $30.636_{\pm 0.843}$ | $4.375_{\pm 0.604}$ |
| FID($x^{t_5}$) | $36.001_{\pm 0.856}$ | $\mathbf{4.3316_{\pm 0.526}}$ |

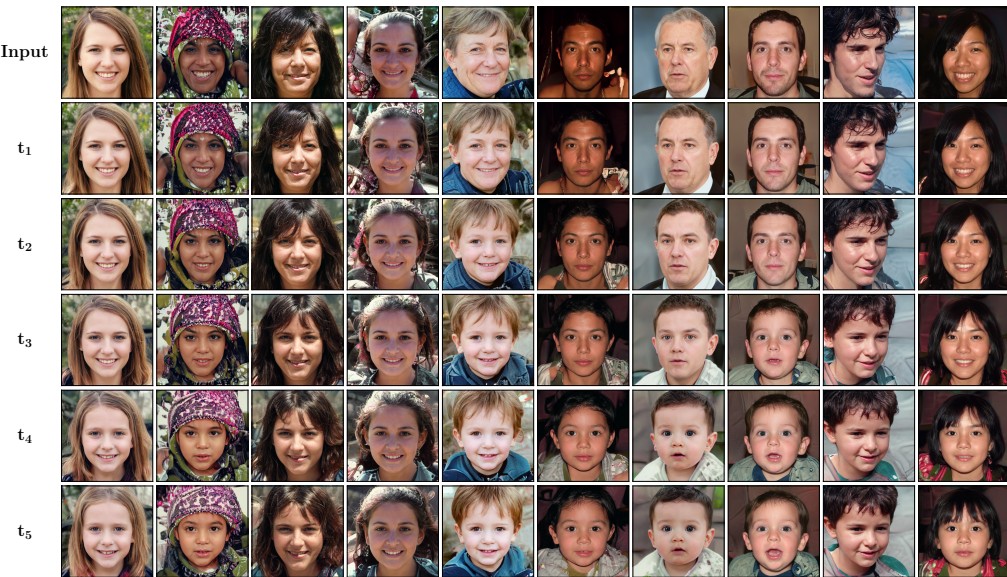

Figure 15: Adult → Child image generation in the FFHQ transfer using the CWG method.

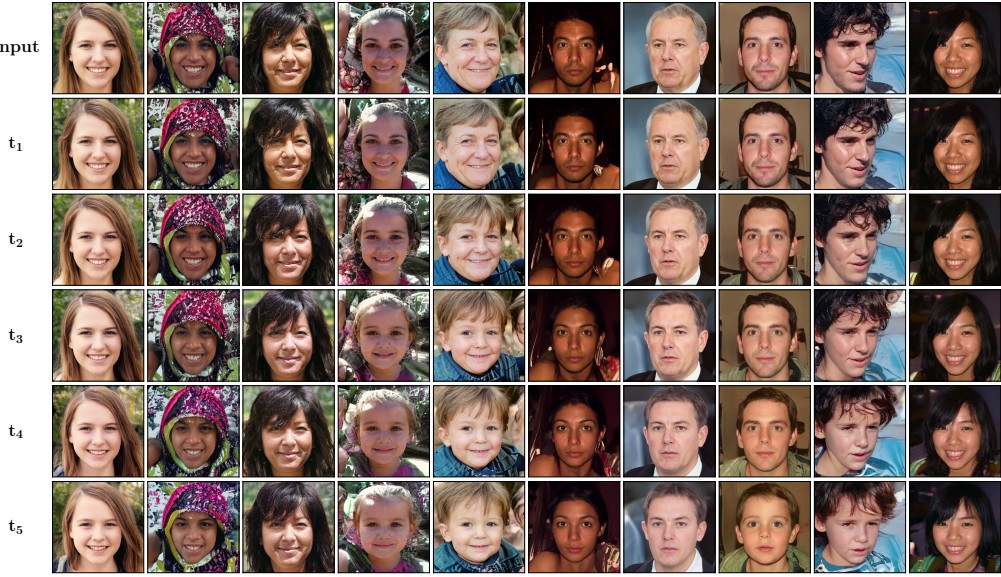

Figure 16: Adult → Child image generation FFHQ transfer using the GSBM method.

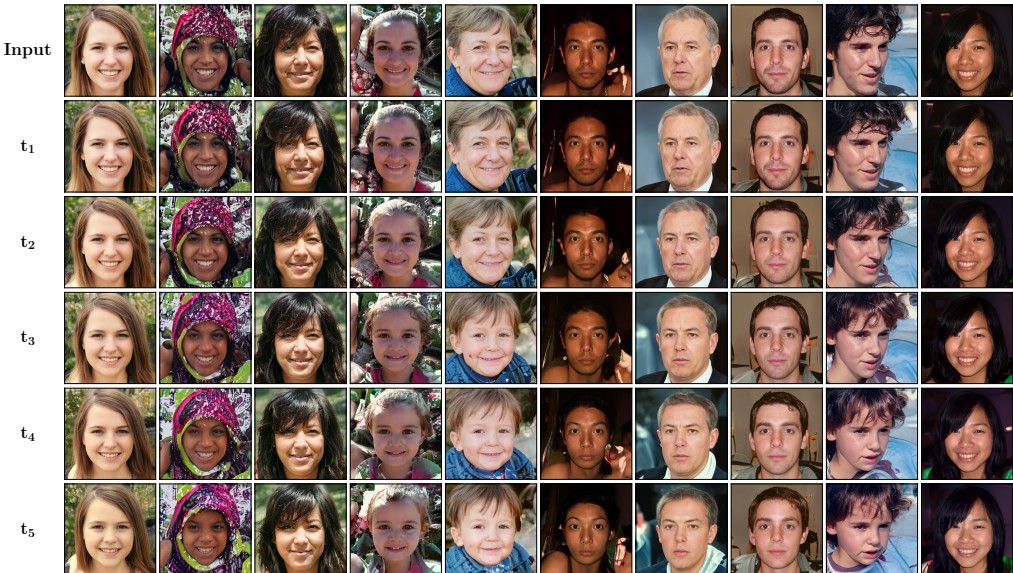

Figure 17: Adult → Child image generation FFHQ transfer using the DSBM method.

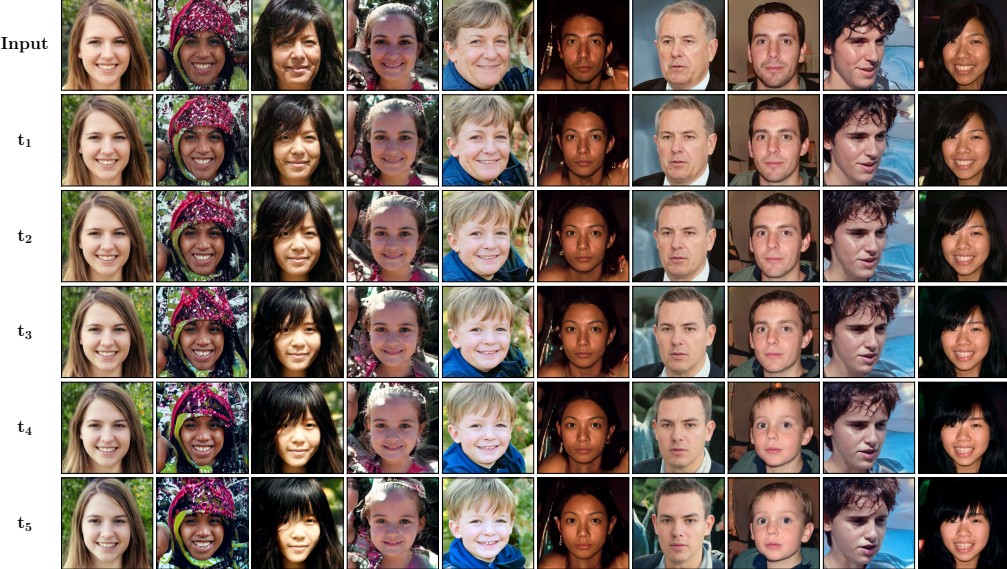

Figure 18: Adult → Child image generation FFHQ transfer using the SB-Flow method.

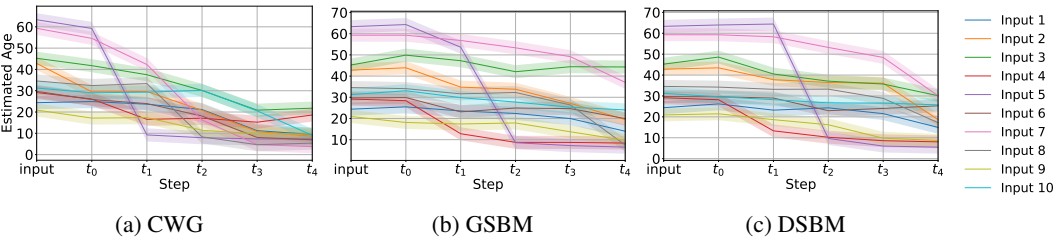

(a) CWG        (b) GSBM        (c) DSBM

Figure 19: Age prediction for the samples generated in the FFHQ transfer experiment.

## G.7 MNIST-TO-EMNIST

The MNIST and EMNIST datasets are low-resolution ($28 \times 28$) handwritten character datasets that serve as long-standing benchmarks for generative modeling, representation learning, and probabilistic transport (Cohen et al., 2017). In this work, we adopt MNIST as the source distribution $\rho_a$ and EMNIST (letters) as the target $\rho_b$ to study unpaired image-to-image translation. We restrict the analysis to ten digits and ten letters. The two datasets are visually related but statistically distinct, making the MNIST-to-EMNIST transport problem sufficiently non-trivial while still permitting clear qualitative and quantitative evaluation. Our goal is to produce terminal samples that match the EMNIST distribution while ensuring that the Schrödinger Bridge generates meaningful intermediate states that remain on the data manifold rather than collapsing into noise or artifacts.

For this purpose, we introduce a potential function $U$ that penalizes deviations from the learned data manifold. Building on the approach of Song & Itti (2025), where generative models are evaluated by measuring the distance between their outputs and a geometric manifold of real images learned by a VAE, we adopt a similar strategy. Specifically, we use a state-of-the-art VAE architecture, with parameters listed in Table 37, to learn a manifold composed of images from both marginal distributions $\rho_a$ and $\rho_b$. The potential function $U(x^t)$, for samples $x^t \sim \rho^t$, is then defined as the squared distance between a bridge sample $x^t$ and its VAE-projected reconstruction $\tilde{x}^t = \mathrm{VAE}(x^t)$: $U(x^t) = \|x^t - \tilde{x}^t\|^2$. No intermediate samples $\rho_m$ are used in this experiment.

The results are shown visually in Figure 20 and quantitatively in Table 38. As observed in the other experiments, DSBM fails to regularize the stochastic path between the two distributions, and SB-Flow searches for an interpolation that first removes features of the source digit not present in the target letter, rather than producing intermediate characters that remain close to the data manifold at every time step. GSBM produces only a deterministic trajectory along the data manifold, which it then uses as the mean of a transient Gaussian distribution. However, because the stochastic component is not properly constrained, these Gaussian samples can drift off the manifold, leading to unrealistic intermediate states. This drift appears as increased noise in the midpoint samples and results in higher FID scores. In contrast, the CWG method produces intermediate characters that remain closest to the source and target distributions at every discretized time step. Notably, our method also demonstrates strong computational efficiency, achieving training times significantly lower than those of the baselines.

To evaluate the guided version of the CWG method, we propose a Gaussian deblurring test, a well-established inverse problem commonly used in image restoration (Feng et al., 2025; Wang et al., 2026). In this setup, a reference image from the EMNIST dataset is processed with a Gaussian kernel and provided as the target for restoration. In this test, the guided loss function, $\|f(x^{t_K}) - y\|$, uses the blurry image as the target $y$, and the feature function $f$ is defined as,

$$f(x^{t_K}) = \mathbf{K}_{\mathrm{gaussian}}(\sigma, r) \odot x^{t_K}, \tag{74}$$

where $\mathbf{K}_{\mathrm{gaussian}}(\sigma, r)$ is a Gaussian kernel operator of variance $\sigma$ and size $r$, applied to the sample $x^{t_K}$ generated at the last time step of the bridge. The specific kernel parameters are detailed in Table 36. This loss measures the similarity between the blurred version of the generated image and the blurred reference, ensuring compatibility between the two. The restoration results are shown visually in Figure 21 and quantitatively in Table 39.

Table 36: Parameters for the feature function $f$ (74), used for guided generation in the Mnist-to-Emnist experiment.

| Parameter | Value |
|---|---|
| Kernel variance $\sigma$ | 3 |
| Kernel size $r$ | 9 |

Table 37: Architecture of the ConvVAE used in the MNIST-to-EMNIST experiment. All Conv2D and ConvTranspose2D layers use ReLU activations unless otherwise specified.

| Stage | Layer (channels) | Output size |
|---|---|---|
| Input | Single-channel image | $1 \times 28 \times 28$ |
| Encoder | Conv2D ($1 \rightarrow 32$, k4, s2, p1) | $32 \times 14 \times 14$ |
|  | Conv2D ($32 \rightarrow 64$, k4, s2, p1) | $64 \times 7 \times 7$ |
|  | Conv2D ($64 \rightarrow 128$, k3, s1, p1) | $128 \times 7 \times 7$ |
|  | Flatten | $128 \cdot 7 \cdot 7 = 6272$ |
| Latent space | Linear $\rightarrow \mu$ | 2 |
|  | Linear $\rightarrow \log \sigma^2$ | 2 |
| Decoder | Linear $\rightarrow$ reshape | $128 \times 7 \times 7$ |
|  | ConvT2D ($128 \rightarrow 64$, k4, s2, p1) | $64 \times 14 \times 14$ |
|  | ConvT2D ($64 \rightarrow 32$, k4, s2, p1) | $32 \times 28 \times 28$ |
|  | ConvT2D ($32 \rightarrow 1$, k3, s1, p1) | $1 \times 28 \times 28$ |

Table 38: FID scores computed w.r.t. the EMNIST distribution ($\downarrow$) and training time (tt) in the MNIST-to-EMNIST experiment.

| Metric | CWG | GSBM | DSBM | SB-Flow |
|---|---|---|---|---|
| FID($x^{t_0}$) | $77.25_{\pm 5.37}$ | $86.80_{\pm 6.11}$ | $102.94_{\pm 7.04}$ | $79.35_{\pm 4.86}$ |
| FID($x^{t_1}$) | $\mathbf{83.33_{\pm 3.74}}$ | $114.53_{\pm 8.48}$ | $228.36_{\pm 13.35}$ | $106.28_{\pm 5.46}$ |
| FID($x^{t_2}$) | $\mathbf{95.29_{\pm 6.39}}$ | $128.69_{\pm 7.80}$ | $195.38_{\pm 12.24}$ | $121.80_{\pm 6.32}$ |
| FID($x^{t_3}$) | $\mathbf{56.42_{\pm 5.63}}$ | $89.75_{\pm 6.25}$ | $163.69_{\pm 11.34}$ | $107.29_{\pm 4.74}$ |
| FID($x^{t_4}$) | $\mathbf{29.73_{\pm 3.54}}$ | $58.28_{\pm 7.17}$ | $108.53_{\pm 11.99}$ | $34.37_{\pm 2.17}$ |
| FID($x^{t_5}$) | $\mathbf{11.42_{\pm 0.32}}$ | $11.75_{\pm 0.36}$ | $11.69_{\pm 0.31}$ | $\mathbf{11.29_{\pm 0.27}}$ |
| tt (s) | $\mathbf{570_{\pm 30}}$ | $30100_{\pm 1000}$ | $8750_{\pm 300}$ | $2200_{\pm 60}$ |

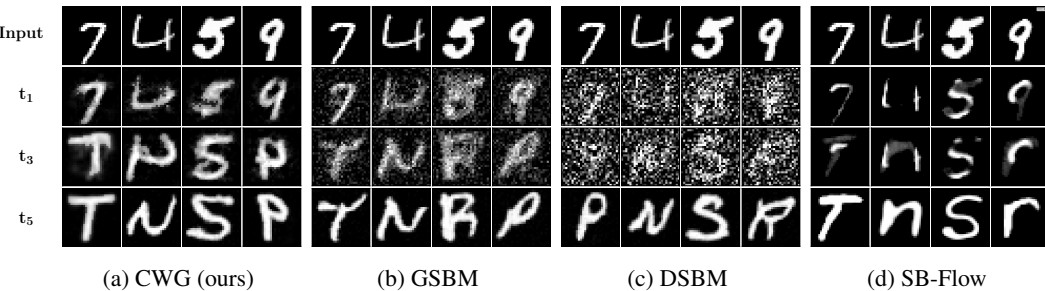

(a) CWG (ours)    (b) GSBM    (c) DSBM    (d) SB-Flow

Figure 20: Snapshots from the bridges computed in the MNIST-to-EMNIST experiment.

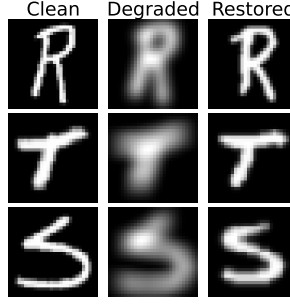

Figure 21: Gaussian deblurring test for the Mnist-to-Emnist experiment. Results for the guided CWG method.

| Metric | Score |
|---|---|
| FID | $12.72_{\pm 0.36}$ |
| LPIPS | $0.28_{\pm 0.02}$ |
| PSNR | $12.48_{\pm 1.11}$ |
| SSIM | $0.37_{\pm 0.04}$ |
| tt (s) | $320_{\pm 20}$ |

Table 39: Quantitative metrics for the Gaussian deblurring experiment: FID score ($\downarrow$) between the clean and reconstructed samples, LPIPS ($\downarrow$), PSNR ($\uparrow$), SSIM $\uparrow$), training time (tt) ($\downarrow$).

