# OpenReview forum: "Contact Wasserstein Geodesics for Non-Conservative Schrödinger Bridges"
_ICLR.cc/2026/Conference — ICLR 2026 Poster_

### Official Review · Reviewer_TgbM · 2025-10-29

**Soundness:** 3
**Presentation:** 2
**Contribution:** 3
**Rating:** 6
**Confidence:** 3

**Summary:**

This paper proposes the nonconservative generalized Schrödinger bridge (NCGSB), which covers a mathematical generalization of SB for nonconservative systems, increasing model flexibility. The noticeable changes include applying decaying energy with a time-varying state, representing the (stochastic) Lagrangian action. Furthermore, this Lagrangian formulation can also be applied to guided generation using a guidance function. The authors used a ResNet to model successive pushforwards for the discretized geodesic, enabling high-dimensional applicability, including image generation.

**Strengths:**

* The manuscript is clearly written and straightforward.
* The authors introduce a new generalization with purposeful intent and physical motivation, which could draw interest from audiences in many fields.
* The overall approach is sound and appropriate.
* The proposed method can be applied to various domains and is scalable.

**Weaknesses:**

* Lacking comparison results in the image domain. The overall volume and benchmark comparison for image-to-image translation can be considered insufficient. The authors are encouraged to review scalable SB methods for images from 2024 and 2025.
* I could not precisely follow how the proposed system ensures the uniqueness of the solution. Unlike in conservative systems, the rate of energy decay can be varied in nonconservative systems, so should there be a hyperparameter (or an assumption of 1) to ensure uniqueness? Also, the guidance function is another degree of freedom and seems to require careful design by practitioners based on the specific problem.

**Questions:**

* Could you please provide the FID scores for MNIST-to-EMNIST SB training?

---

> ### Author Response · Authors · 2025-11-23
> **Response to Reviewer TgbM (1/2)**
>
> ## Weaknesses
>
> > The overall volume and benchmark comparison for image-to-image translation can be considered insufficient. The authors are encouraged to review scalable SB methods for images from 2024 and 2025.
>
> > Could you please provide the FID scores for MNIST-to-EMNIST SB training?
>
> To address the reviewer’s comments, we have added two new experiments on image-to-image translation using widely adopted benchmarks: the suggested MNIST-to-EMNIST translation, and the high-resolution Flickr-Faces-HQ (FFHQ, 1024 × 1024) dataset, which presents a more challenging task and is commonly used in the literature [1], [2], [3]. Additionally, we have included a recent state-of-the-art method capable of fitting images directly in the ambient space, the SB-Flow method [4].
>
> We would like to emphasize that the scope of our method goes beyond unpaired image matching. Our theoretical framework provides important guarantees for fitting the dynamics of physical systems, which also allows us to achieve strong results on computational biology datasets of different nature [5]. Moreover, the ability to incorporate intermediate information, a feature highly valuable for predicting physical systems [6], [7] but rarely used in unpaired image matching, further motivates us to report this variety of experiments.
>
> In Section 6, the results of the image generation experiments have been updated. The sea temperature experiment (Appendix G.4) and the robot manipulation experiment (Appendix G.5) have been revised to include the SB-Flow method. This method generalizes the path between two marginals through an interpolation that ignores the geometry of the data manifold, which leads to unsatisfactory snapshots for both the sea temperature evolution (Table 4, Figure 5) and the robot dynamics (Table 5, Figure 7).
>
> The new MNIST-to-EMNIST translation and FFHQ translation experiments, which complete Section 6, are detailed in Appendix G.6 and G.7, respectively. In the MNIST experiment, we obtain results comparable to the best-performing baselines in terms of FID score, while producing a generation process that remains closer to the data manifold. This comes with a 74.1% improvement in computational time (Table 38). In the FFHQ translation experiment, we improve upon the second-best baseline by 36.7% in terms of target-generation success, while simultaneously achieving a 64.9% computational improvement (Table 7).
>
> > I could not precisely follow how the proposed system ensures the uniqueness of the solution. Unlike in conservative systems, the rate of energy decay can be varied in nonconservative systems, so should there be a hyperparameter (or an assumption of 1) to ensure uniqueness?
>
> The geodesic computation problem indeed takes place on the space $(\mathcal{P}^+(\mathcal{M}) \times \mathbb{R})$, which implies that the uniqueness of the geodesic on the probability manifold $\mathcal{P}^+(\mathcal{M})$ depends on the specification of an energy trajectory, represented by the scalar component in $\mathbb{R}$. In our framework, this dependence reduces to choosing a single hyperparameter that sets either the initial or the final energy.
>
> As shown in the ablation study reported in Appendix G.3, satisfactory results are achieved by varying the hyperparameter within the range (between 0.5 and 1.5) (Table 21). Nevertheless, allowing this degree of freedom is important for improving generalization: it enables the model to more closely follow the underlying geometry of the dataset (Figure 13). This is particularly beneficial in scenarios where the data exhibit non-constant evolution, with variations that become more pronounced either near the beginning or near the end of the process. In such settings, adapting the energy schedule allows the model to track these trends faithfully.

---

> > ### Author Response · Authors · 2025-11-23
> > **Response to Reviewer TgbM (2/2)**
> >
> > ## Weaknesses
> >
> > > Also, the guidance function is another degree of freedom and seems to require careful design by practitioners based on the specific problem.
> >
> > Our guidance formulation follows the same principle as classifier- or loss-based guided generative methods [8, 9, 10], in which an auxiliary loss term steers the learned distribution. This loss steers the generation toward specific classes of samples or toward samples exhibiting desired task-specific properties.
> >
> > In the results section, we present an example of spatial guidance for LiDAR manifold navigation, where the target is the right side of a mountain (Figure 4; Appendix G.2). The loss function (Eq. 68) is a simple linear penalty applied to samples falling outside the desired region, identical to the one used in our proof-of-concept experiment (Figure 3). In the robot manipulation task (Appendix G.5), the guidance term (Eq. 71) linearly penalizes final placement locations outside a designated goal region.
> > In the MNIST-to-EMNIST experiment (Appendix G.7), the guidance is used for Gaussian deblurring, with the loss (Eq. 74) matching those commonly used in the literature for similar restoration problems [11].
> >
> > Because the bridge is defined as a geodesic in probability space, this penalty effectively modifies the distance metric under which the geodesic is computed, providing a geometric interpretation of distribution steering. Crucially, this metric adjustment is not ad hoc but arises naturally from the geometry of the problem.
> >
> > Thus, while the guidance function is task-dependent, it can be any well-defined measure of the desired property one wishes to impose within the framework.
> >
> > ## References
> >
> > [1] Light and Optimal Schrödinger Bridge Matching, Gushchin et al., 2024
> >
> > [2] Implicit Image-to-Image Schrödinger Bridge, Wang et al., 2025
> >
> > [3] Feedback Schrödinger Bridge Matching, Theodoropoulos et al., 2025
> >
> > [4] Schrodinger Bridge Flow for Unpaired Data Translation, De Bortoli et al, NeurIPS 2024
> >
> > [5] Visualizing structure and transitions in high-dimensional biological data exploration, Moon et al., Nature 2019
> >
> > [6] Multi-marginal Schrödinger bridges with iterative reference refinement, Shen et al, AISTATS 2025
> >
> > [7] Deep momentum multi-marginal Schrödinger bridge, Chen at al., NeurIPS 2023
> >
> > [8] Gradient Guidance for Diffusion Models: An Optimization Perspective, Guo et al., NeurIPS 2024
> >
> > [9] Loss-Guided Diffusion Models for Plug-and-Play Controllable Generation, Song at al., ICML 2023
> >
> > [10] FlowGrad: Controlling the Output of Generative ODEs with Gradients, Liu et al., CVPR 2023
> >
> > [11] On the Guidance of Flow Matching, Feng at al., ICML 2025

---

### Official Review · Reviewer_GjBx · 2025-10-30

**Soundness:** 3
**Presentation:** 3
**Contribution:** 3
**Rating:** 8
**Confidence:** 3

**Summary:**

This paper proposes the Non-Conservative Generalized Schrödinger Bridge (NCGSB) and the Contact Wasserstein Geodesic (CWG) framework to model stochastic processes with energy variation, overcoming the energy-conservation limitation of classical Schrödinger Bridges. By leveraging contact Hamiltonian mechanics, the authors extend the Wasserstein geometry to handle non-conservative dynamics and introduce a ResNet-based non-iterative solver with near-linear complexity. The approach also supports guided generation through metric modulation. Experiments on LiDAR manifold navigation, single-cell sequencing, and image generation demonstrate efficiency and accuracy improvements over strong baselines.

**Strengths:**

- The extension of SBs to non-conservative systems via contact geometry is quite interesting.
- The proposed method demonstrates faster convergence and better accuracy across diverse tasks.
- The paper is well-organized presentation with clear motivation and theoretical grounding.

**Weaknesses:**

- Limited large-scale validation:
Although diverse, most experiments remain moderate in scale. Evaluation on large-scale datasets or high-dimensional physical systems (e.g., long protein trajectories or large image benchmarks) would better demonstrate scalability claims.
- Ablation studies and interpretability:
The role of the contact energy term, and its impact on stability or generalization, is not extensively analyzed. A detailed ablation on the energy variation factor and metric modulation would strengthen the empirical evidence.

**Questions:**

- Could you clarify the numerical stability of the proposed solver, especially under long time horizons or many intermediate marginals?
- How sensitive is the method to the choice of potential function $U(x)$?
- Is the contact Hamiltonian augmentation equivalent to introducing an auxiliary energy channel in the latent space? If so, can this be viewed as a generalization of underdamped diffusion bridges?

- Could the authors provide an intuitive geometric visualization of how the energy-varying geodesic differs from a standard Wasserstein geodesic?

---

> ### Author Response · Authors · 2025-11-23
> **Response to Reviewer GjBx (1/3)**
>
> ## Weaknesses
>
> > Limited large-scale validation: Although diverse, most experiments remain moderate in scale. Evaluation on large-scale datasets or high-dimensional physical systems (e.g., long protein trajectories or large image benchmarks) would better demonstrate scalability claims.
>
> We acknowledge that the first two experiments, conducted in 3D and 5D Euclidean spaces, are indeed low-dimensional. However, as reported in Table 12, the image-based experiments are substantially higher-dimensional: **4,096** for the sea-surface temperature task and **12,288** for the robotic reconstruction task, both operating directly in image space without  dimensionality reduction. Most datasets used in computational biology [1, 16, 17, 18] involve only a few thousand dimensions after filtering for meaningful features from the raw measurements, and existing Schrödinger Bridge methods in this field are typically limited to fewer than 100 principal dimensions [2, 13, 19]. Nevertheless, to address the reviewer’s suggestion, we have included an experiment on a large-scale image benchmark using the Flickr-Faces-HQ (FFHQ) high-resolution dataset (1024 × 1024), a standard benchmark in recent related works [3], [4], [5]. We would be happy to further clarify or expand the experimental scope if our experiments still do not meet the reviewer’s definition of large-scale evaluation.
>
> > Ablation studies and interpretability: The role of the contact energy term, and its impact on stability or generalization, is not extensively analyzed. A detailed ablation on the energy variation factor and metric modulation would strengthen the empirical evidence.
>
> To further analyze the role of the contact energy term, we have added a new proof of concept example (Figure 1, detailed in Appendix F.2) and expanded the ablation study presented in Appendix G.3.
>
> As explained in Appendix C.1, the contact Hamiltonian dynamics cause the energy of the probability path to vary between the source and target distributions. In our geometric interpretation, the probability path is modeled as a geodesic flow, and this energy variation along the path translates into a modulation of the metric structure of the Wasserstein manifold connecting the distributions. This modulation, which defines a new Jacobi metric $\tilde{g}_{\mathrm{J}}$ on the Wasserstein manifold, is controlled by a hyperparameter. During the geodesic computation, this metric remains fixed and does not dynamically change, so it does not affect the stability of the optimization. Indeed, using a varying energy does not substantially alter the training time (see Table 21, Appendix G.3).
>
> While the position of the geodesic nodes anchored to the available data for intermediate or terminal marginals remain largely stable across geodesic computations using either the Wasserstein or the Jacobi metric , the choice of the metric has a stronger influence on the floating nodes, which adjust their positions to minimize relative distances accordingly (Appendix F.2). Increasing or decreasing the energy influences whether the geodesic nodes cluster closer to the source or the target, allowing the model to better fit datasets that exhibit faster evolution at the beginning or the end. The extended ablation study (Appendix G.3, Fig. 13) quantifies how distance variations across time steps impact performance, highlighting the benefits of the energy-varying CWG formulation.
>
> ## Questions
>
> > Could you clarify the numerical stability of the proposed solver, especially under long time horizons or many intermediate marginals?
>
> We have added  Appendix E.3. for the stability considerations of our approach. The discrete ResNet parameterization converts the continuous geodesic optimization problem into a tractable nonlinear programming problem. Under piece-wise smooth assumptions of the modulation term $\Phi^k$ of the Jacobi metric that are always preserved in our implementation, we guarantee the convergence toward an optimal set of parameters that geometrically corresponds to one of the possible coordinates sets for the charts parameterizing the probability space (Appendix E.1).
>
> Long time horizons in our framework are represented by multiple blocks of the ResNet architecture, and due to its skipping connections [14], this architecture can reliably accommodate thousands of time steps [15]. The presence of several marginals does not compromise stability, as each marginal can be associated with the distribution corresponding to a specific geodesic node, effectively anchoring that node in probability space using the available data. In our experiments, we often adjust the weights  (Table 10) to prioritize this anchoring, allowing the remaining floating nodes to realign according to length-minimization criteria (i.e., the loss (9) ). As discussed in the computational complexity study (Appendix E.3), the presence of multiple marginals does not change the optimization problem, aside from adding the corresponding loss term.

---

> ### Author Response · Authors · 2025-11-23
> **Response to Reviewer GjBx (2/3)**
>
> ## Questions
>
> > How sensitive is the method to the choice of potential function $U(x)$?
>
> The potential function $U(x)$ introduces the action of conservative forces to guide the probability flow between the marginals of the Schrödinger Bridge problem. Its inclusion and the specific form it takes may depend on the problem. The choices made for our experiments are standard in the literature and shared by the baseline method [7]. Alternative formulations consistent with the task objectives are, of course, possible, provided they are at least piecewise smooth, as required for stable convergence (Appendix E.3).
>
> 1. In the experiments on LiDAR navigation (Appendix G.2), stem cell differentiation (Appendix G.3), sea temperature prediction (Appendix G.4), robot task reconstruction (Appendix G.5), and Mnist-to-Emnist translation (Appendix G.7), the potential function $U(x)$ primarily serves to penalize deviations from the data manifold.
>
>     1a. For stem cell differentiation (Equation 69), we adopt the LAND metric [6, 8] as $U(x)$, a Riemannian metric that takes lower values near regions of high data density. In the LiDAR navigation experiment, this same metric is augmented with an additional term that penalizes the altitude of the stochastic distribution (Equation 68), ensuring that the bridge remains on the manifold while circumventing the mountain. This formulation matches the definition of potential function used by the GSBM baseline [7], ensuring a fair comparison.
>
>     1b. In the image domain, we adopt a Variational Autoencoder (VAE) to learn the data manifold associated with the available images. We then evaluate an image's distance from this manifold by computing the difference between the image and its projection onto the manifold (i.e., its encoded and reconstructed version), as suggested by [10]. Notably, the GSBM baseline [7] employs the same projection-based strategy.
>
>
> 2. In the proof of concept (section 5) and in the FFHQ translation (Appendix G.6), the tasks do not require the modeling of conservative forces or to remain close to a data manifold, so we consider $U(x)=0$ in this case.
>
> As shown in the experiments where CWG and GSBM share the same potential $U(x)$, the ability to compute the geodesic bridge induced by this metric depends on the specific Schrödinger Bridge formulation. While GSBM fits a deterministic path on the data manifold and then samples around this path (so its samples may not lie on the manifold), CWG succeeds in fitting a stochastic bridge that remains on the data manifold.
>
> Without a suitable manifold encoding, setting $U=0$ the Schrödinger Bridge framework may not generalize toward the data manifold. In a vector space (case 1a), a geometry-unaware bridge may generalize along straight lines, for example, crossing directly through the mountain in the LiDAR experiment. In image space (case 1b), using any feature extractor to measure distances in feature space rather than pixel space can already lead to better generalization, even without a VAE trained on the task-specific data. Nevertheless, employing such a VAE remains convenient, as its compact feature representation makes distance computation lightweight and easy to backpropagate through.

---

> ### Author Response · Authors · 2025-11-23
> **Response to Reviewer GjBx (3/3)**
>
> ## Questions
>
> > Is the contact Hamiltonian augmentation equivalent to introducing an auxiliary energy channel in the latent space? If so, can this be viewed as a generalization of underdamped diffusion bridges?
>
> First, we would like to highlight that in most of our image-generation experiments, our framework does not use a latent space but fits the Schrödinger bridge directly in the ambient space. Regarding the contact Hamiltonian dynamics, the additional energy channel arises from the explicit introduction of the auxiliary variable $z^t$ (Equation 4), which makes  the dynamics dependent on the path already taken. This yields a non-Markovian effect at the cost of adding only a single scalar variable.
>
> In contrast, other state-of-the-art energy-based generative models [10] use energy $E(x)$ solely in the definition of the target marginals $e^{-E(x)}$. This energy is tied to the target distribution and is minimized during optimization, but it is not the energy governing the dynamics of the process itself, and is therefore fundamentally different from our formulation.
>
> The underdamped diffusion bridges [11, 12] do actually focus on the energy of the stochastic process. They introduce dissipation by adopting Langevin dynamics as the evolution equation, similar in spirit to the non-conservative momentum-based Schrödinger Bridge frameworks [13] described in Section 2. However, these methods **require doubling the dimensionality of the system by explicitly including velocity as part of the state**. For these reasons, we view our approach not as a generalization of underdamped diffusion bridges, but rather as a more efficient alternative that emerges naturally from the underlying geometric interpretation. This geometric view, in turn, unlocks the formulation of the SB solver as a geodesic optimizer, thus avoiding expensive iterative schemes.
>
> > Could the authors provide an intuitive geometric visualization of how the energy-varying geodesic differs from a standard Wasserstein geodesic?
>
> The newly added teaser image (Figure 1) provides an intuitive geometric illustration of how the energy-varying geodesic differs from a standard Wasserstein geodesic.
>
> ## References
>
> [1] Visualizing structure and transitions in high-dimensional biological data exploration, Moon et al., Nature 2019
>
> [2] Multi-marginal Schrödinger bridges with iterative reference refinement, Shen et al, AISTATS 2025
>
> [3] Light and Optimal Schrödinger Bridge Matching, Gushchin et al., ICML 2024
>
> [4] Implicit Image-to-Image Schrödinger Bridge for Image Restoration, Wang et al., 2025
>
> [5] Feedback Schrödinger Bridge Matching, Theodoropoulos et al., ICLR 2025
>
> [6] A Locally Adaptive Normal Distribution,  Arvanitidis et al., NeurIPS 2016
>
> [7] Generalized Schrödinger Bridge Matching, Liu at al., ICLR 2024
>
> [8] Flow Matching on General Geometries, Chen at al., ICLR 2024
>
> [9] Riemannian-geometric fingerprints of generative models., Song at al., 2025
>
> [10] Generalized Energy Based Models, Arbel et al., ICLR 2021
>
> [11] Underdamped Diffusion Bridges with Applications to Sampling, Blessing et al., ICLR 2025
>
> [12] Score-based Generative Modeling with Critically-damped Langevin Diffusion, Dockhorn et al., ICLR 2022
>
> [13] Momentum Multi-Marginal Schrödinger Bridge Matching, Theodoropoulos et al., NeurIPS 2025
>
> [14] Norm-preservation: Why residual networks can become extremely deep?, Zaeemzadeh et al, 2020
>
> [15] Deep Residual Learning for Image Recognition, He et al, 2015
>
> [16] A Single-Cell Transcriptome Atlas of the Human Pancreas, Muraro et al., 2016
>
> [17] Transcriptional Heterogeneity and Lineage Commitment in Myeloid Progenitors, Paul et al., 2015
>
> [18] Brain structure. Cell types in the mouse cortex and hippocampus revealed by single-cell RNA-seq, Zeisel et al., 2015
>
> [19] Trajectory Inference with Smooth Schrödinger Bridges, Hong et al., ICML 2025

---

### Official Review · Reviewer_7xEZ · 2025-11-01

**Soundness:** 3
**Presentation:** 2
**Contribution:** 3
**Rating:** 4
**Confidence:** 3

**Summary:**

This paper addresses the Generalized Schrödinger Bridge (GSB) problem by relaxing certain energy-conserving constraints and instead applying a Hamiltonian mechanism to enable efficient training and inference. Furthermore, by adjusting the time-varying energy term, it allows for guided generation. The proposed method, called CWG, is implemented in the form of a ResNet, which effectively reduces training time. The authors also validate guided generation using interpolation when intermediate marginals are provided partially.

**Strengths:**

- Powerful Theoretical Basis: This study generalizes the conventional GSB framework from the perspective of Hamiltonian dynamics by introducing an energy-varying term. The theoretical analysis presented in the appendix provides a strong justification for the proposed methodology and effectively supports its validity.

- Compact Model Design: The proposed method is implemented using a ResNet architecture, which represents the most essential structure for expressing the Schrödinger bridge. This choice is appropriate, as additional modules can be incorporated when the task complexity increases.

- Efficient Training Speed: Compared to existing methods such as GSBM and DSBM, the proposed approach demonstrates superior performance and faster training. In particular, its advantage in training efficiency further enhances the novelty of this work.

**Weaknesses:**

#### Major Weaknesses
- Weak Presentation: Some important details necessary to fully understand the study may be missed unless the appendix is read carefully. Moreover, the core ideas of the proposed method appear too late in the manuscript. In my opinion, it would be better to move the Related Work section toward the later part of the paper and highlight the main contributions more prominently.

- Insufficient Emphasis on Mathematical Notation: The manuscript includes all key equations and attempts to distinguish important terms using various colors. However, the chosen colors are not sufficiently distinguishable, and the equations are interspersed throughout the text, which somewhat reduces readability. To emphasize key terms more effectively, it would be helpful to use a separate table or structured summary highlighting the core notations.

- Inconsistent Explanation of Intermediate Marginals: In Equation (9), the loss function appears to require intermediate marginal distributions. However, in the actual experiments, only a subset of them—or none at all—seems to be used. This inconsistency is not clearly emphasized. It would strengthen the manuscript to consolidate and highlight these explanations right after Equation (9), rather than scattering them across different sections.

- Novelty of Guided Generation: The authors emphasize guided generation as the main contribution of their work. In my view, guidance-based generation should ideally enable generation through indirect guidance rather than direct guidance toward an intended mode. In the current formulation and experiments, however, it seems that only interpolation between intermediate distributions during the optimal transport process is evaluated.

#### Minor Weaknesses

- In Appendix D.2, the notation for the intermediate distribution is inconsistent. It should be corrected from $\rho_n$ to $\rho_m$.
- In Appendix D.3, the time complexity analysis could be strengthened by comparing it with other existing methods. Such a comparison would help to better emphasize the novelty of the proposed approach.

**Questions:**

See Weaknesses section.

---

> ### Author Response · Authors · 2025-11-23
> **Response to Reviewer 7xEZ (1/3)**
>
> ## Summary
>
> > This paper addresses the Generalized Schrödinger Bridge (GSB) problem by relaxing certain energy-conserving constraints and instead applying a Hamiltonian mechanism to enable efficient training and inference.
>
> For clarity, we would like to emphasize that the paper addresses both the Generalized Schrödinger Bridge (GSB) problem and the multi-marginal Schrödinger Bridge (mmSB) problem, which are often tackled separately in the literature. Furthermore, the Hamiltonian formulation is indeed energy-conserving and provides  an alternative interpretation of the standard Schrödinger Bridge optimality conditions. In our work, we extend this framework to allow energy-varying solutions through a novel contact Hamiltonian formulation.
>
> ## Weaknesses
> >  The core ideas of the proposed method appear too late in the manuscript. In my opinion, it would be better to move the Related Work section toward the later part of the paper and highlight the main contributions more prominently.
>
> We thank the reviewer for the suggestion. Due to the constraints of the rebuttal phase, we may not be able to restructure the paper at this stage, but we will carefully consider this possibility if the paper is accepted. Nevertheless, we are happy to further clarify our contributions, which are highlighted in colored boxes at the end of Section 1, with additional explanations:
>
> 1. Our first contribution is a novel contact Hamiltonian formulation of the Schrödinger Bridge, previously considered only from the traditional Hamiltonian perspective. This new formulation, that we call the Non-Conservative Generalized Schrödinger Bridge (NCGSB), allows for solutions that leverage the  energy variation provided by the contact Hamiltonian, thus providing more flexible probability paths between the marginal distributions (see Figure 1). The flexibility is controlled via hyperparameters that determine the energy dynamics.
> 2. We then introduce the Contact Wasserstein Geodesic (CWG) framework, a general solver compatible with all Schrödinger Bridge variants (GSB, mmSB, NCGSB). By leveraging the geometric structure of the SB problem, our CWG method avoids the iterative computations between the bridge and reference refinements that are required by the other baselines [8, 9, 10, 11], and it scales only linearly with both the number of samples and the problem dimensions, in sharp contrast with iterative methods. The accuracy and efficiency of our CWG solver are demonstrated in the results shown in Section 6.
> 3. Furthermore, we propose a guided generation methodology based on the modulation of the geometric structure learned by the CWG framework. This modification is introduced through a task-dependent loss function (e.g., generating samples with prescribed properties or solving inverse generation problems).
>
> In the revised paper, we have clarified these contributions by improving the introduction section (Section 1, paragraph beginning with ‘Our paper’) and the accompanying colored box. Thanks to the reviewer’s feedback, we have provided clearer explanations of concepts that were previously perceived as confusing.
>
> > The authors emphasize guided generation as the main contribution of their work.
>
> > By adjusting the time-varying energy term, it allows for guided generation.
>
> We would like to clarify that the main contribution of our work, summarized in our answer above, is the formulation of the energy-varying Schrödinger Bridge and a general non-iterative solver for the SB problem. Guided generation is coupled to these contributions as an additional feature to control and steer the generative process. Note that energy variation and guidance are distinct features: energy variation determines the overall behavior of the learned bridge $\rho^t(x)$ between the available data distributions (e.g., how the bridge transitions from distribution $\rho_a(x)$ to $\rho_b(x)$), whereas guidance is an additional extension that can steer the learned bridge according to specific task $y$ based on a loss function: $\rho^t(x|y)$. This guidance can either modify the learned bridge between $\rho_a(x)$ and $\rho_b(x)$ or adjust the shape of the target distribution $\rho_b(x)$.
>
> We have included this clarification in Section 1 of the revised paper.
>
> > To emphasize key terms more effectively, it would be helpful to use a separate table or structured summary highlighting the core notations.
>
> We agree with the reviewer and have added a one-page table summarizing all the mathematical notation in Appendix B, which is referenced at the beginning of Section 3.

---

> ### Author Response · Authors · 2025-11-23
> **Response to Reviewer 7xEZ (2/3)**
>
> ## Weaknesses
>
> > In Equation (9), the loss function appears to require intermediate marginal distributions. However, in the actual experiments, only a subset of them—or none at all—seems to be used.
>
> These intermediate marginal distributions are optional, and the loss can be applied without them. We have clarified this in the loss description provided in Section 5. In the Experiments, we further specify whether intermediate marginals are available for each problem.
>
> > In my view, guidance-based generation should ideally enable generation through indirect guidance rather than direct guidance toward an intended mode. In the current formulation and experiments, however, it seems that only interpolation between intermediate distributions during the optimal transport process is evaluated.
>
> We understand the reviewer’s comment as twofold: (1) guided generation should enable sampling beyond modes already learned during training, and (2) this generative capability should be quantitatively evaluated. If we have misunderstood the reviewer’s request, we kindly invite the reviewer to clarify.
>
> Regarding point 2, we emphasize that all guidance experiments in the results section (LiDAR manifold navigation and image generation) are accompanied by quantitative evaluations (Tables 6 and 17), which measure how well the guided model aligns with the target labels or desired properties.
>
> Regarding point 1, we first clarify that our guidance formulation follows the same principle as classifier- or loss-based guided generative methods [1], [2], [3], where a loss term steers the learned distribution. In our geometric framework, we adopt this idea by introducing a penalty $\Vert y - f({x}^{t_s}) \Vert^2$ (Eq. 5) applied at a chosen time step $t_s$, encouraging the feature function $f$ to match the target value $y$. Because the bridge is defined as a geodesic in probability space, this penalty effectively modifies the distance metric under which the geodesic is computed, providing a geometric interpretation of distribution steering. Importantly, this metric adjustment is not arbitrary but emerges naturally from the contact geometry of the non-conservative SB problem.
>
> We acknowledge that our initial experiments primarily focused on guidance toward specific modes already present in the training set, either through class-based generation or property maximization. We made  this choice deliberately, since these experiments are the most natural use-cases for the fields investigated by our approach, and the literature provides many examples of such application for guided models [4], [5], [6].
>
> Nevertheless, to further illustrate the capabilities of our approach, we have extended our evaluation with a Gaussian deblurring experiment, where the guidance function measures the discrepancy between the blurred generated sample and the noisy input to be reconstructed. This represents an example of an indirect problem, a widely used benchmark for evaluating guided methods [7]. The results are referenced in Section 6, and are further detailed in Appendix G.7. We provide visual reconstructions in Figure 21, demonstrating our framework’s ability to recover faithful images from highly blurred samples. Table 39 reports four quantitative metrics that evaluate the reconstruction quality, which are satisfactory given the significant noise applied to the reference image.
>
> ### Minor Weaknesses
>
> > In Appendix D.3, the time complexity analysis could be strengthened by comparing it with other existing methods.
>
> Following the reviewer’s suggestion,  we have incorporated three state-of-the-art methods, prioritizing the baselines used in Section 6. Our method stands out over existing SB frameworks due to: its linear scaling with both dimensionality and number of samples, no explicit dependence on the number of intermediate marginals to fit, and the complete absence of outer-loop iterations. This analysis is elaborated in Appendix E.5.
>
> > In Appendix D.2, the notation for the intermediate distribution is inconsistent. It should be corrected from $\rho_n$ to $\rho_m$.
>
> We thank the reviewer for their careful analysis and for catching this typo. We have corrected this.

---

> > ### Author Response · Authors · 2025-11-23
> > **Response to Reviewer 7xEZ (3/3)**
> >
> > ## References
> >
> > [1] Gradient Guidance for Diffusion Models: An Optimization Perspective, Guo et al., NeurIPS 2024
> >
> > [2] Loss-Guided Diffusion Models for Plug-and-Play Controllable Generation, Song at al., ICML 2023
> >
> > [3] FlowGrad: Controlling the Output of Generative ODEs with Gradients, Liu et al., CVPR 2023
> >
> > [4] Training Free Guided Flow Matching with Optimal Control, Wang et al., ICLR 2025
> >
> > [5] Dirichlet Flow Matching with Applications to DNA Sequence Design, Stark et al., ICLR 2024
> >
> > [6] Simple Guidance Mechanisms for Discrete Diffusion models, Schiff et al., ICLR 2025
> >
> > [7] On the Guidance of Flow Matching, Feng at al., ICML 2025
> >
> > [8] Generalized Schrödinger Bridge Matching, Liu at al., ICLR 2024.
> >
> > [9] Multi-marginal Schrödinger bridges with iterative reference refinement, Shen et al., AISTATS 2025
> >
> > [10] Diffusion Schrödinger bridge matching, Shi et al., NeurIPS 2023
> >
> > [11] Deep momentum multi-marginal Schrödinger bridge, Chen et al., NeurIPS 2023

---

### Author Response · Authors · 2025-12-01
**Summary**

Dear Area Chair,

Unfortunately, we were not able to continue the discussion with the reviewers. We therefore summarize our main contributions, as stated in our paper, and clarifications below.

### Summary of Contributions:

1. **Non-Conservative Generalized Schrödinger Bridge (NCGSB)**:
We introduce a novel contact Hamiltonian formulation of the Schrödinger Bridge, generalizing prior work restricted to traditional Hamiltonian dynamics. The contact Hamiltonian formulation unlocks probability paths between marginals that are no longer confined to a single energy level. The energy hyperparameter serves as an inductive bias, enabling ascending or descending energy profiles along the probability path. This flexibility is essential for modeling real-world processes that naturally exhibit varying physical energy. For example, in weather forecasting, it can capture dissipative processes, such as storms gradually losing intensity, as well as energy-increasing processes, such as monsoon winds and rains strengthening as solar heating rises from winter to summer.

2. **Contact Wasserstein Geodesic (CWG) framework**:
We propose a general solver applicable to all Schrödinger Bridge (SB) variants (Generalized SB, multi marginal SB, Non-Conservative Generalized SB). By exploiting the contact geometric structure of the problem, CWG avoids iterative bridge or reference refinements and scales linearly with sample size and dimension, unlike existing methods.

3. **Guided generation methodology**:
We introduce a task-dependent guidance mechanism that modulates the bridge learned by CWG. This allows the stochastic path to be adapted to downstream tasks, such as generating samples with prescribed properties or addressing inverse generation problems.

### Summary of improvements during the rebuttal:

- **Reviewer 7xEZ**: We addressed the misinterpretation regarding our main contributions, as our guided-generation method was misleadingly understood as  the general energy-varying formulation. Specifically, we clarified that the energy hyperparameter controls the bridge’s generalization, whereas the guidance mechanism reshapes it for new tasks through an explicit loss. This feedback helped us further refine the presentation of our contributions in Section 1. We also added a notation table in Appendix B to improve readability, strengthened the time-complexity analysis in Appendix E.5 by including comparisons with state-of-the-art methods [1,2,3], and provided additional Gaussian deblurring tests for the guidance experiments in Appendix G.7.
- **Reviewer GjBx** suggested including  larger scale experiments, additional ablation studies, and clarifications on the loss components and training stability. We addressed these suggestions by adding a new high-resolution 1024x1024 image transfer experiment on the FFHQ dataset, complementing the existing image-generation experiments on sea-surface temperature forecasting and video robot-manipulation reconstruction (dimensions of 4,096 and 12,288). We also added the requested ablations in Appendices F.2 and G.3, showing a 15% improvement in reconstruction performance enabled by the energy-varying formulation, and included a new stability study in Appendix E.3. All questions were addressed with detailed answers. Finally, we clarified the method’s sensitivity to the potential function used in the loss by discussing our design choices and reporting results across different scenarios (Appendix G.2-G.7), and we clarified the position of our approach in the literature, highlighting its differences from the methods raised by the reviewers [4, 5].
- **Reviewer TgbM** suggested including additional benchmarks and baselines for the image transfer experiments, as well as clarifications on (1) how energy variation affects solution uniqueness, and (2) on the choice of guidance functions. We addressed these points by adding the requested MNIST-to-EMNIST translation experiment in Appendix G.7, a new FFHQ experiment in Appendix G.6, and by including the recent SB-Flow method as an additional baseline for all image-based tasks. While SB-Flow removes the alternating optimization procedure and improves computational efficiency, it still requires higher training time than our method across all benchmarks. We also clarified the geometric conditions and design choices ensuring solution uniqueness, and explained that our guidance functions follow the same straightforward criteria used in other loss-based guidance methods in the literature.

We consider that the added clarifications, new experiments, and extra ablation studies strengthen the presentation of our approach and its contributions to the state of the art in Schrödinger bridge-based generative models.

---

> ### Author Response · Authors · 2025-12-01
> **References**
>
> ### References
> [1] Trajectory inference with smooth Schrödinger bridges, Hong at al., ICML 2025.
>
> [2] Deep momentum multi-marginal Schrödinger bridge, Chen at al., NeurIPS 2023.
>
> [3] Multi-marginal Schrödinger bridges with iterative reference refinement, Shen et al., AISTATS 2025.
>
> [4] Score-based Generative Modeling with Critically-damped Langevin Diffusion, Dockhorn et al., ICLR 2022.
>
> [5] Momentum Multi-Marginal Schrödinger Bridge Matching, Theodoropoulos et al., NeurIPS 2025.
>
> [6] Schrodinger Bridge Flow for Unpaired Data Translation, De Bortoli et al, NeurIPS 2024.

---

### Meta-Review · Area_Chair_mTK5 · 2026-01-08

**Summary:**

The paper introduces a novel framework for generative modeling and stochastic process estimation called the Non-Conservative Generalized Schrödinger Bridge (NCGSB). The authors use contact Hamiltonian mechanics to allow for energy-varying probability paths. They propose the Contact Wasserstein Geodesic (CWG) framework, which uses a ResNet architecture to solve the bridge problem without the expensive iterative refinements required by previous methods.

**Reviewer Concerns:**

Reviewer 7xEZ stated that the core ideas appeared too late in the manuscript, and the mathematical notation was difficult to follow.
Reviewers GjBx, TgbM think that most experiments were low-dimensional (3D/5D) or moderate. Reviewers asked for larger-scale validation and more competitive baselines (like SB-Flow).
The reviewer 7xEZ argued that "guidance" seemed like simple interpolation rather than steering toward a new mode.

**Reviewer Scores:**

The author have addressed most of the concerns raised by reviewers, the overall score should have increased if they had been able to participate fully in the discussion.

---

### Decision · Program_Chairs · 2026-01-26

Accept (Poster)